# On the theoretical limit of gradient descent for Simple Recurrent Neural Networks with finite precision[*]

**Volodimir Mitarchuk**                                                      *volodimir.mitarchuk@univ-st-etienne.fr*
**Rémi Emonet**[†]                                                             *remi.emonet@univ-st-etienne.fr*
**Rémi Eyraud**                                                               *remi.eyraud@univ-st-etienne.fr*
**Amaury Habrard**[†]                                                        *amaury.habrard@univ-st-etienne.fr*
*Université Jean Monnet Saint-Etienne, CNRS, Institut d'Optique Graduate School, Inria, Laboratoire Hubert Curien*
*UMR 5516, F-42023, Saint-Etienne, France*
[†]*and Institut Universitaire de France (IUF)*

**Reviewed on OpenReview:** *https://openreview.net/revisions?id=4c9UzDhg49*

## Abstract

Despite their great practical successes, the understanding of neural network behavior is still a topical research issue. In particular, the class of functions learnable in the context of a finite precision configuration is an open question. In this paper, we propose to study the limits of gradient descent when such a configuration is set for the class of Simple Recurrent Networks (SRNs). We exhibit conditions under which the gradient descent will provably fail. We also design a class of SRN based on Deterministic finite State Automata (DFA) that fulfills the failure requirements. The definition of this class is constructive: we propose an algorithm that, from any DFA, constructs an SRN that computes exactly the same function, a result of interest on its own.

## 1 Introduction

One of the main challenges Machine Learning is facing is the lack of theoretical understanding of the reasons of the practical successes of deep learning. Indeed, whole research fields have been revolutionized by the boom of deep neural networks, from signal (Mehrish et al., 2023) to image processing (Li et al., 2024) including finance (Zhang et al., 2021), for instance, but much remains to be done to fully understand their capabilities and limits.

In this paper, we propose to study this issue in the context of Recurrent Neural Networks (RNN). As with neural networks in general, we can identify at least two ways of interpreting capabilities and limits. The first one is to analyze the expressivity, that is, the class of functions a class of models can compute. Within this direction of research a notorious theoretical result states that vanilla RNN, the SRN (Elman, 1990), are able to compute any Turing machine (Siegelmann & Sontag, 1992). Though of great interest, this theorem says little about the models used nowadays: its proof relies on the use of unbounded time and requires neurons of infinite precision. It has been shown (Chung & Siegelmann, 2021) that it is possible to maintain the Turing completeness for an SRN in finite precision if the algorithm has access to a potentially unbounded amount of neurons. A similar result is proposed by Stogin et al. (2024) with a particular emphasis on the provable stability of the proposed architecture. When it comes to conventional RNN architectures, those with a fixed number of neurons, Weiss et al. (2018) practically shows that the expressivity drastically decreases for all types of RNNs, most of them fall in classes equivalent to the expressivity of finite state machines. Merrill et al. (2020) theorize the phenomenon of saturation and derive a hierarchy of RNN architectures. The saturation in RNN occurs when the bounded activation function composing the RNN (the sigmoidal function of tanh) are pushed to their boundaries and thus produce binary outputs. Despite the interest of such a result, the saturation formalized by Merrill et al. (2020) is unrealistic because it is based on the notion of limit to be performed at every iteration.

---

[*]This work is supported in part by the ANR TAUDoS (ANR-20-CE23-0020).

Another way of establishing RNN limitations and capabilities is to investigate what can be achieved in the context of statistical learning. In this area of research the most notorious phenomenon to be studied is the one of vanishing gradient. Bengio et al. (1994) and Hochreiter (1998) demonstrate that long dependencies in the SRN computations are very difficult to be learned by gradient descent. Issues with the vanishing/exploding gradient were so common that Hochreiter & Schmidhuber (1997) proposed the LSTM, a RNN architecture designed to bypass these issues. Algorithmic solutions such as gradient clipping or smoothing were also considered in the literature (Pascanu et al., 2013) as well as avoiding particular regions of the parameter space (Ribeiro et al., 2020).

Our work fills a theoretical gap at the intersection of the above-mentioned fields of research. We explore the limitations of training RNNs by Gradient Descent (GD) in the context of finite precision. To the best of our knowledge, this is the first study of its kind for RNNs. We built on the saturation assumption and the finite precision arithmetic's to explicitly delimit regions of parameter space that cannot be reached by classical GD with bounded learning rate. We thus theoretically prove that it is impossible to learn saturated SRNs. We also demonstrate that in finite precision the saturation in RNNs (based on sigmoidal or hyperbolic tangent function) can be defined without any notion of limit, hence RNNs can exhibit binary behavior with bounded parameters. And finally, leveraging the saturation in finite precision we exhibit a way of accurately defining SRN parameters, allowing the SRN to simulate a finite state machine. Therefore, in this way we demonstrate that the non-learnable region by GD for SRNs contains an interesting and expressive class of functions, the one of Deterministic Finite State Automaton (DFA). More formally our contributions are:

- We prove a theorem stating that saturated SRNs cannot be learned by gradient descent. The theorem shows that a subset of gradient parameters will be too small to have an effect during the gradient descent and thus prevent the algorithm to reach the optimum.

- We conduct experiments that validate the theorem, and that show that the task of learning saturated SRN is even harder than predicted by the theory.

- We provide an algorithm that, given a Deterministic Finite State Automaton (DFA), constructs a saturated SRN that will simulate the DFA. The algorithm is an extension of the algorithm provided by Minsky (1967) to SRN in finite precision.

The article is structured as follows: Section 2 of this paper is devoted for notations and the introduction of different necessary notions for the proofs. Section 3 contains the secondary but necessary result on the stability to noise in saturated SRNs. In Section 4, we develop the arguments towards the gradient descent failure in finite precision, as well as a detailed example of the application of the theorem on a small SRN. Section 5 details the algorithm that for a given DFA outputs a saturated SRN capable of simulating the DFA. Section 6 is the experimental section where we provide empirical evidence that our theory is valid. And the final Section 7 contains the conclusive discussion with possible extensions of our result, future works and related works on the limitation of gradient descent in finite precision.

## 2 Preliminaries

In this section are enumerated all the notations and elementary operations that are used in this paper.

### 2.1 Elements of Linear Algebra

Numbers, besides properly defined constants, are noted by a lowercase italic letter, *e.g.*, $x \in \mathbb{R}$; the vectors of dimension higher than 1 by a bold lowercase letter, *e.g.*, $\mathbf{x} \in \mathbb{R}^d$ for $d > 1$; the matrices by a bold uppercase letter, *e.g.*, $\mathbf{M} \in \mathbb{R}^{d_1 \times d_2}$. For $\mathbf{x} \in \mathbb{R}^d$, $\mathbf{x}[j]$ represents the $j^{th}$ coordinate of the vector $\mathbf{x}$. $\mathbf{I}_d = Diag(1, \ldots, 1) \in \mathbb{R}^{d \times d}$ is the identity matrix of dimension $d$. Given $x \in \mathbb{R}$ and $\mathbf{M} \in \mathbb{R}^{d_1 \times d_2}$, $x \cdot \mathbf{M}$ denotes multiplication of all values of $\mathbf{M}$ by $x$.

Given two vectors $\mathbf{u}, \mathbf{v} \in \mathbb{R}^d$ we note $[\mathbf{u} \ \mathbf{v}]$ the matrix of $\mathbb{R}^{d \times 2}$ whose columns are the original vectors. We extend this definition to matrices in an obvious way.

For a matrix $\mathbf{M} \in \mathbb{R}^{d_1 \times d_2}$ we will denote the row vectors of $\mathbf{M}$ by $\mathbf{m}_i$ for $1 \leq i \leq d_1$, and we denote the column vectors of $\mathbf{M}$ by $\mathbf{m}^j$ for $1 \leq j \leq d_2$.

For $\mathbf{x}, \mathbf{y} \in \mathbb{R}^d$, $\langle \mathbf{x}, \mathbf{y} \rangle$ is the usual scalar product in $\mathbb{R}^d$. If we fix $\mathbf{m} \in \mathbb{R}^d$ the function $\langle \mathbf{m}, \cdot \rangle : \mathbb{R}^d \ni \mathbf{x} \mapsto \langle \mathbf{m}, \mathbf{x} \rangle \in \mathbb{R}$ is a linear form.

For $\mathbf{M} \in \mathbb{R}^{d_1 \times d_2}$ we define the transposition operation by $\mathbf{M}^T \in \mathbb{R}^{d_2 \times d_1}$ where the rows of $\mathbf{M}$ are columns of $\mathbf{M}^T$. For $\mathbf{x} \in \mathbb{R}^d$, in this paper $\|\mathbf{x}\| = \sqrt{\sum_{s=1}^{d} \mathbf{x}[s]^2}$.

$\odot$ is the Hadamard product between vectors: if $\mathbf{x} \in \mathbb{R}^d$ and $\mathbf{y} \in \mathbb{R}^d$ we have $\mathbf{x} \odot \mathbf{y} = \mathbf{z}$, with $\mathbf{z}[j] = \mathbf{x}[j] \cdot \mathbf{y}[j]$ for $1 \le j \le d$.

$\oplus$ is the concatenation operation on vectors. For $\mathbf{x} \in \mathbb{R}^{d_1}$ and $\mathbf{y} \in \mathbb{R}^{d_2}$ we have $\mathbf{x} \oplus \mathbf{y} \in \mathbb{R}^{d_1 + d_2}$ (where $d_1$ and $d_2$ might be different): if $\mathbf{x} = (x_1, \ldots, x_{d_1})$ and $\mathbf{y} = (y_1, \ldots, y_{d_2})$ then $\mathbf{x} \oplus \mathbf{y} = (x_1, \ldots, x_{d_1}, y_1, \ldots, y_{d_2})$.

### 2.2 Elements of arithmetic

In this part, we introduce the definitions considered for modeling the finite precision configuration. We begin with the concept of centered integer representation related to a finite basis representation.

**Definition 1** *(Centered integer representation)*
*Let $B, X > 1$ two positive integers. We define*

$$\llbracket X \rrbracket^B = \{\lambda_1 \cdots \lambda_X \ : \ \lambda_j \in \{0, \ldots, B - 1\}\},$$

*the set of integers in basis $B$ encoded on $X$ digits. We define a centered integer representation as the image of the set $\llbracket X \rrbracket^B$, by a bijective function $shift$ defined as follows:*

$$\llbracket X \rrbracket^B \ni x \mapsto shift(x) = x - \left\lfloor \frac{B^X}{2} \right\rfloor$$

*where $\lfloor \cdot \rfloor$ is the floor function. We represent a centered integer representation by the couple of integers $(B, X)$.*

The definition above will help us to define in a rigorous way the floating numbers and a finite precision configuration.

**Definition 2 (Finite precision configuration)** *A finite precision configuration is defined by a tuple of positive integers $(B, M, X)$, $B$ being the basis, $M$ and $X$ representing the number of digits allocated for the "mantissa" and the "exponent", respectively. We define two sets:*

$$\llbracket M \rrbracket^B := \{x_1 \cdots x_M \ : \ x_i \in \{0, \ldots, B - 1\}\}$$
$$\llbracket X \rrbracket^B := \{\lambda_1 \cdots \lambda_X \ : \ \lambda_i \in \{0, \ldots, B - 1\}\}$$

*In this configuration a float number $x$ is represented as a triplet:*

$$float_x = (sign, mantissa, exponent)$$

*where $sign \in \{-1, 1\}$, $mantissa \in \llbracket M \rrbracket^B$ and $exponent \in \llbracket X \rrbracket^B$. In order to go from the representation to the number itself, one needs to:*

$$x = (sign) \times (mantissa) \times B^{shift(exponent)}$$

*where $shift$ is the function of Definition 1.*

For example in the configuration $(B, M, X) = (10, 5, 4)$ the number $x := -12.35$ is represented by the triplet $(-1, 12350, 4997)$ and when we switch to the actual number we get

$$x = -1 \times 12350 \times 10^{shift(4997)}$$
$$= -1 \times 12350 \times 10^{-3}$$

where in this case the function $shift$ is defined by $shift(n) = n - \frac{10^4}{2} = n - 5000$.

One can remark that this notation allows multiple representations for the same number, indeed both representations $(-1, 12350, 4997)$ and $(-1, 01235, 4998)$ will produce the same float number $-12.35$ in the finite precision configuration $(10, 5, 4)$. Therefore to avoid all ambiguities, we say that a non zero float number is in its underline{normal form} $(sign, mantissa, exponent)$ if the $mantissa = x_1 \cdots x_M$ is such that $x_1 \neq 0$.

The setting $(B, M, X)$ generates the set $\mathbb{G}_{(B,M,X)}$ composed of all numbers representable following these restrictions. We will abusively use the notation $\mathbb{G}$ instead of $\mathbb{G}_{(B,M,X)}$ for the sake of simplicity when there is no ambiguity.

As an example, in the IEEE 754 norm[1], the simple floats are encoded in base $B = 2$ on 32 bits, with 1 bit for the sign, $M = 23$ for the mantissa and $X = 8$ for the exponent.

Arithmetic's operations in a finite precision configuration might sometimes be counter intuitive. For instance, suppose one wants to compute the addition of two numbers $x$ and $s$. Let $x = (sign_x) \times x_1 \cdots x_M \times B^{exponent_x}$ and $s = (sign_s) \times s_1 \cdots s_M \times B^{exponent_s}$ such that $exponent_s < exponent_x$. In order to compute $x + s$, one first needs to match the exponents of $x$ and $s$. Without loss of generality, we can fix the highest exponent and modify the magnitude of the other number, however since we can represent the mantissa of $s$ with only $M$ digits we have to compromise and replace the $s$ mantissa $s_1 \cdots s_M$ by an approximation $\underbrace{0 \cdots 0}_{D \ zeros} s_1 \cdots s_{M-D}$

and therefore obtaining an approximation of $s$

$$\tilde{s} = \underbrace{0 \cdots 0}_{D \ zeros} s_1 \cdots s_{M-D} \times B^{exponent_x}$$

with $D = exponent_x - exponent_s$. We obtain that the operation $x + s$ is in fact $x + \tilde{s}$.

This illustrates the consequences of rounding in finite precision, since the $D$ last digits of $s$ are not taken into account in the addition. Worse, if $D \geq M$, the operation $x + s$ will output $x$. In the following Lemma (proved in Appendix D), we discuss another phenomenon, the saturation of the bounded activation function. In infinite precision, functions as the sigmoidal function and the hyperbolic tangent, never reach their boundaries when evaluated on real numbers. However in finite precision they do reach the boundaries, and the following lemma characterizes the smallest float number to underline{saturate} the sigmoidal function.

**Lemma 3 (Underflow boundary)** *Let $(B, M, X)$ the finite precision configuration and let $\sigma : x \mapsto \frac{1}{1+e^{-x}}$ be the sigmoidal function. We define $u := \lfloor \frac{B^X}{2} \rfloor$ and $l := B^X - u$. If $M < \min\{u, l\}$ then there exists $J \in \mathbb{G}_{(B,M,X)}$ such that $\sigma(J) = 1$ and $\sigma(-J) = 0$ and for all positive float numbers $s$ such that $J - s < J$ we have $\sigma(-(J - s)) \neq 0$*

The proof of this result is in Appendix D.

## 2.3 Elements of Deep Learning

In the following we focus on a single Recurrent Neural Network architecture, and therefore we set the activation function to be $\sigma$.

In deep learning the same notation is usually used for the activation function as a one variable scalar function and for a whole activation layer. To avoid any confusion, in this paper $\sigma'$ will refer to the derivative of the function $\sigma : \mathbb{R} \to \mathbb{R}$ and $\sigma^\nabla : \mathbb{R}^d \to \mathbb{R}^d$ its element-wise counterpart defined by:

$$\sigma^\nabla(x_1, \ldots, x_d) = (\sigma'(x_1), \ldots, \sigma'(x_d)).$$

In this work we focus on just one RNN architecture: the Simple Recurrent Networks (SRNs) (Elman, 1990). However, we would like to emphasize that we are studying SRNs in finite precision, so to formalize this point, we introduce the definition FP-SRNs where FP stands for Finite Precision.

---

[1]The current version are defined according to the following reference: IEEE Computer Society (2019-07-22). IEEE Standard for Floating-Point Arithmetic. IEEE STD 754-2019. IEEE. pp. 1–84. doi:10.1109/IEEESTD.2019.8766229. ISBN 978-1-5044-5924-2. IEEE Std 754-2019.

**Definition 4 (FP-SRN)** *Let $(B, M, X)$ be a finite precision configuration, we set $\mathbb{G} = \mathbb{G}_{(B,M,X)}$. A Finite Precision Simple Recurrent Network (FP-SRN) $\mathcal{R}$ is a triplet $(\mathbf{h}_0, \mathcal{E}, \mathcal{D})$ where $\mathbf{h}_0 \in \mathbb{G}^h$ is the initial hidden state vector, and $\mathcal{E}, \mathcal{D}$ are functions called $\underline{encoder}$, $\underline{decoder}$ respectively and are defined by:*

$$\mathcal{E} : \mathbb{G}^u \times \mathbb{G}^h \ni (\mathbf{x}, \mathbf{h}) \mapsto \sigma\left(\mathbf{M}_h(\mathbf{x} \oplus \mathbf{h} \oplus 1)\right) \in \mathbb{G}^h$$

$$\mathcal{D} : \mathbb{G}^h \ni \mathbf{h} \mapsto \sigma\left(\mathbf{M}_o(\mathbf{h} \oplus 1)\right) \in \mathbb{G}^o.$$

*The values $u, h$ and $o$ are positive integers representing the input dimension, the hidden dimension and the output dimension respectively. $\mathbf{M}_h \in \mathbb{G}^h \times \mathbb{G}^{u+h+1}$ and $\mathbf{M}_o \in \mathbb{G}^o \times \mathbb{G}^{h+1}$ are matrices. And $\sigma$ is the sigmoidal logistic function.*

*For a finite sequence $\{\mathbf{x}_k\}_{k=1}^T \subset \mathbb{G}^u$ with $T \geq 1$ an integer, the execution of $\mathcal{R}$ on the sequence $\{\mathbf{x}_k\}_{k=1}^T$ is given by the sequence $\{(\mathbf{h}_k, \mathbf{y}_k)\}_{k=1}^T$ where $\mathbf{h}_k$ and $\mathbf{y}_k$ are defined recursively following the rule:*

$$\mathbf{h}_k := \mathcal{E}(\mathbf{x}_k, \mathbf{h}_{k-1}) \ ; \ \ \mathbf{y}_k := \mathcal{D}(\mathbf{h}_k).$$

In this definition, we have chosen to group the parameters together in a single matrix as far as possible. However, in this work it will also be useful to have a definition that makes a clear distinction between parameters of different types, so we give the following definition, perfectly equivalent to the one above.

**Definition 5 (FP-SRN version 2)** *A Finite Precision Simple Recurrent Network (FP-SRN) $\mathcal{R}$ is a triplet $(\mathbf{h}_0, \mathcal{E}, \mathcal{D})$ where $\mathbf{h}_0 \in \mathbb{G}^h$ is the initial hidden state vector, and $\mathcal{E}, \mathcal{D}$ are functions called $\underline{encoder}$, $\underline{decoder}$ respectively and are defined by:*

$$\mathcal{E} : \mathbb{G}^u \times \mathbb{G}^h \ni (\mathbf{x}, \mathbf{h}) \mapsto \sigma\left(\mathbf{U}\mathbf{x} + \mathbf{W}\mathbf{h} + \mathbf{b}\right) \in \mathbb{G}^h$$

$$\mathcal{D} : \mathbb{G}^h \ni \mathbf{h} \mapsto \sigma\left(\mathbf{V}\mathbf{h} + \mathbf{c}\right) \in \mathbb{G}^o$$

*where $\sigma$ is the activation function, $\mathbf{U}$ is a matrix with $h$ rows and $u$ columns, $\mathbf{W}$ is a matrix with $h$ rows and $h$ columns, $\mathbf{V}$ is a matrix with $o$ rows and $h$ columns, and $\mathbf{b}$, $\mathbf{c}$ are vectors of dimension $h$ and $o$ respectively.*

*Execution on a sequence is defined exactly as in Definition 4*

We call the matrix $\mathbf{W}$ the transition kernel, $\mathbf{U}$ the input kernel and $\mathbf{V}$ the output kernel. Clearly, the two definitions are equivalent, because

$$\mathbf{M}_h = [\mathbf{U} \ \mathbf{W} \ \mathbf{b}] \quad \mathbf{M}_o = [\mathbf{V} \ \mathbf{c}] .$$

Another handy notation is the one of the $\underline{\text{parameter vector}}$. Indeed, when we need to consider a parameters in a FP-SRN $\mathcal{R}$ independently of their function, it is convenient to group all the parameters together in a vector of parameters denoted $\theta$. So, to underline the distinction between two FP-SRN which differ by their parameters we will note them respectively $\mathcal{R}_\theta$ and $\mathcal{R}_{\tilde{\theta}}$.

Depending on the context and the problem we are treating, we will prefer one to the other definition. Notably, Definition 4 will be handy to define a neuron:

**Definition 6 (Neuron)** *Let $\mathbf{m} \in \mathbb{G}^d$ be a vector and $\sigma$ the sigmoidal function. A neuron is a function defined as:*

$$\sigma_{\mathbf{m}} : \mathbb{G}^d \ni \mathbf{x} \mapsto \sigma(\langle \mathbf{m}, \mathbf{x} \rangle) \in \mathbb{G}.$$

In this work, the definition of the gradient requires the notion of a differentiated neuron.

**Definition 7 (Differentiated Neuron)** *Let $\mathbf{m} \in \mathbf{G}^d$ be a vector and $\sigma$ the sigmoidal function. We define a differentiated neuron as:*

$$\sigma'_{\mathbf{m}} : \mathbb{G}^d \ni \mathbf{x} \mapsto \sigma'(\langle \mathbf{m}, \mathbf{x} \rangle) \in \mathbb{G}.$$

We intentionally defined a neuron as composition of a linear form and a non linear activation function (in our case the sigmoidal function). If we refer to Definition 4, then by looking at the encoder or decoder coordinate

by coordinate we find the definition of a neuron. If we take an FP-SRN $\mathcal{R}$ and a decoder coordinate $1 \leq j \leq h$, we have that:

$$\sigma\Big(\mathbf{M}_h(\mathbf{x} \oplus \mathbf{h} \oplus 1)\Big)[j] = \sigma\big(\langle \mathbf{m}_j, \mathbf{x} \oplus \mathbf{h} \oplus 1 \rangle\big)$$

In the following, the integer $u, h$ and $o$ will represent the input dimension, the hidden dimension and the output dimension, respectively.

**Definition 8 ($\beta$-saturated SRN)** *Let $\mathcal{R}$ be a FP-SRN and $\mathcal{X} = \big\{ \{\mathbf{x}_k\}_{k=1}^T : T \geq 1, \, \mathbf{x}_k \in \mathbb{R}^u \big\}$ a set of input sequences, we say that $\mathcal{R}$ is $\beta$-saturated for $0 \leq \beta \leq 1$, if for all input sequence $\{\mathbf{x}_k\}_{k=1}^T$, we have:*

$$\min_{1 \leq i \leq h} \Big| \big(\mathbf{U}\mathbf{x}_k + \mathbf{W}\mathbf{h}_{k-1} + \mathbf{b}\big)[i] \Big| > \beta \; \forall \; k \in \{1, \ldots, T\}.$$

In other words, a $\beta$-saturated FP-SRN linear part of the encoder will always produce a vector that is in $\mathbb{R}^h \setminus [-\beta, \beta]^h$, *i.e.* whose coordinates are always at a distance $\beta$ from zero. An interesting fact about $\beta$-saturated SRNs is that for all input sequences $\{\mathbf{x}_k\}_{k=1}^T$, for all $1 \leq k \leq T$, we have:

$$\Big\| \sigma^\nabla \Big( \mathbf{U}\mathbf{x}_k + \mathbf{W}\mathbf{h}_{k-1} + \mathbf{b} \Big) \Big\|_\infty \leq \sigma'(\beta).$$

This inequality derives from the properties of $\sigma$, and the $\beta$-saturation hypothesis. Indeed, by hypothesis we are sure to have $\big(\mathbf{U}\mathbf{x}_k + \mathbf{W}\mathbf{h}_{k-1} + \mathbf{b}\big)[j] \geq \beta$ for all coordinates $1 \leq j \leq h$, on the other side $\sigma'$ is a decreasing function on the interval $[0, +\infty[$ and symmetric. Hence for all $1 \leq j \leq h$ we have:

$$\Big| \sigma^\nabla \Big( \mathbf{U}\mathbf{x}_k + \mathbf{W}\mathbf{h}_{k-1} + \mathbf{b} \Big)[j] \Big| \leq \sigma'(\beta)$$

This inequality is discussed more in detail in the beginning of Appendix A. In supervised machine learning, the training of a neural network requires the use of a loss function defined as follows.

**Definition 9** *A loss function is a function $\mathcal{L} : \mathbb{G}^o \times \mathbb{G}^o \to \mathbb{G}$ that compares computationally the output of a model on a given training data and the expected target value. The Binary Cross entropy is an example for $p, q \in\, ]0, 1[$:*

$$\mathcal{L}(p, q) = -q \cdot \ln(p) - (1 - q) \cdot \ln(1 - p).$$

In a practical context of deep learning one could, during early phases of the training, encounter a situation where the prediction of a neural network is the opposite of the target label *i.e.* $p \approx 0$ and $q = 1$. In this kind of situation we obtain $\mathcal{L}(p, q) = -q \cdot \ln(p) = -\ln(p)$, a value unstable and potentially undefined. Moreover, the gradient of such a calculation is $\frac{1}{p} >> 1$ in the case of $p \approx 0$. Therefore in order to prevent instability, a common practice is to add a small constant $\epsilon$ and define a numerically stable loss function:

$$\mathcal{L}_\epsilon(p, q) = -q \ln(p + \epsilon) - (1 - q) \ln(1 - p + \epsilon). \tag{1}$$

We will assume $\epsilon = 10^{-7}$, corresponding to usual practice in the `tensorflow/keras` platform[2]. This small change will have a profound impact in this work when we discuss the back propagation of the gradient, because the function $\frac{\partial}{\partial p}\mathcal{L}_\epsilon(p, q)$ becomes bounded on $[0, 1] \times \{0, 1\}$.

**Definition 10 (Fixed step Gradient Descent)** *Let $f : \mathbb{R}^d \to \mathbb{R}$ be a differentiable function, $\mathbf{u}_0 \in \mathbb{R}^d$ an initial point and $\alpha > 0$ a positive real number. The fixed step gradient descent of $f$ starting at $\mathbf{x}_0$ is the sequence $\{\mathbf{g}_l\}_{l \geq 0} \subset \mathbb{R}^d$ defined as follows:*

$$\mathbf{g}_0 = \mathbf{x}_0$$
$$\mathbf{g}_l = \mathbf{g}_{l-1} - \alpha \nabla f(\mathbf{g}_{l-1}) \; for \; l > 0$$

*where $\nabla f(\mathbf{g}_{l-1})$ is the vector of partial derivatives of $f$ at the point $\mathbf{g}_{l-1}$.*

This definition naturally extends to a variable learning rate $\alpha_l$.

---

[2]`https://www.tensorflow.org/api_docs/python/tf/keras/backend/epsilon`

## 2.4 Elements of Language Theory

Let $\Sigma$ be a finite set of symbols called an alphabet. $\Sigma^*$ represents all the finite sequences on $\Sigma$ and $\epsilon$ is the empty sequence. Any subset $L$ of $\Sigma^*$ is called a language.

For $\omega \in \Sigma^*$, $|\omega|$ represents its length, that is, its number of symbols. The concatenation of two sequences is denoted by: $\omega \cdot \omega'$.

**Definition 11 (DFA)** *A Deterministic Finite State Automata (DFA) is entirely defined by $(\Sigma, Q, q_1, \delta, F)$ where: $\Sigma$ is a finite alphabet; $Q$ is the finite set of states; $q_1$ is the initial state; $\delta : Q \times \Sigma \rightarrow Q$ is the transition function; $F \subseteq Q$ is the set of accepting states.*

The language represented by a DFA is the set $L = \{\omega \in \Sigma^* : \delta^*(q_1, \omega) \in F\}$, where $\delta^*$ is the transitive extension of the transition function: $\delta^*(q, \epsilon) = q$, $\delta^*(q, a \cdot \omega) = \delta^*(\delta(q, a), \omega)$ for $q \in Q, a \in \Sigma, \omega \in \Sigma^*$.

**Definition 12 (One-hot encoding)** *Let $\Sigma = \{a_1, a_2, \ldots, a_u\}$ be a finite and ordered alphabet of size $u$. Let $\{e_1, e_2, \ldots e_u\}$ be the canonical basis of the vector space $\mathbb{R}^u$, defined by: $e_i[j] = 1$ if $i = j$ and $0$ otherwise. We define a one-hot encoding as a bijective map:*

$$\phi : \ \Sigma \ni a_i \mapsto e_i \in \{e_1, e_2, \ldots e_u\}.$$

*The definition of a one-hot encoding is naturally extended to the elements of $\Sigma^*$ by the bijective map $\phi^* : \Sigma^* \rightarrow \{\mathbf{e}_1, \mathbf{e}_2, \ldots, \mathbf{e}_m\}^*$ with $\phi^*(a_1 a_2 \cdots a_\kappa) \mapsto (\phi(a_1), \phi(a_2), \cdots, \phi(a_\kappa))$.*

One-hot encoding is a bijective correspondence between an ordered set of symbols $\Sigma = \{a_1, \ldots, a_u\}$ and the canonical basis of $\mathbb{R}^u$, $\{\mathbf{e}_1, \mathbf{e}_2, \ldots, \mathbf{e}_u\}$. In order to simplify notations, we take advantage of the bijective correspondence and from now on we consider that $\Sigma = \{\mathbf{e}_1, \mathbf{e}_2, \ldots, \mathbf{e}_u\}$.

## 3 Stability to Noise in Parameters

Our main goal is to prove that there are regions of the parameter space that are not accessible by gradient descent. Our strategy is to exhibit points in the parameter space such that FP-SRNs with those parameters will experience a stationary gradient. The notion of stationary gradient is formally defined later in Definition 15. Nevertheless, intuitively a FP-SRN $\mathcal{R}_\theta$ will experience a stationary gradient when there is a non empty set of coordinates in the parameter vector $\theta$ such that for any element of the training set the gradient for these parameters is negligible with respect to the parameters. We will demonstrate that some $\beta$-saturated FP-SRNs experience a stationary gradient, but in this section we exhibit another particular feature of $\beta$-saturated FP-SRNs, the one of stability to noise. Indeed, Mitarchuk et al. (2024) prove that it is possible to disturb the parameter of a $\beta$-saturated SRN (and thus of a FP-SRNs) and have guarantees that the disturbed FP-SRN is $\tilde{\beta}$-saturated, for $\tilde{\beta}$ proportional to $\beta$ and to the perturbation. We will build on this perturbation property of $\beta$-saturated SRNs to expand the stationary gradient experience around the saturated FP-SRN, and hence obtain a region of parameter space where FP-SRNs experience a stationary gradient. In our work, we have slightly reformulated their result and their definition of $\beta$-saturation, but the theorem stated below and the definition remain perfectly equivalent. Before stating the result we introduce the notion of perturbed SRN.

**Definition 13 (Perturbed FP-SRN )** *Let $\mathcal{R}_\theta$ be a FP-SRN and $\vartheta \in \mathbb{R}^{dim(\theta)}$, we define a perturbed FP-SRN $\mathcal{R}_{\theta+\vartheta}$ as follows:*

$$\mathring{\mathcal{E}} : \mathbb{G}^u \times \mathbb{G}^h \ni (\mathbf{x}, \mathbf{h}) \mapsto \sigma\left((\mathbf{U} + \vartheta_{\mathbf{U}})\mathbf{x} + (\mathbf{W} + \vartheta_{\mathbf{W}})\mathbf{h} + (\mathbf{b} + \vartheta_{\mathbf{b}})\right) \in \mathbb{G}^h$$
$$\mathring{\mathcal{D}} : \mathbb{G}^h \ni \mathbf{h} \mapsto \sigma\left((\mathbf{V}\vartheta_{\mathbf{V}})\mathbf{h} + (\mathbf{c} + \vartheta_{\mathbf{c}})\right) \in \mathbb{G}^o$$

*The hidden and the output vectors produced by the perturbed FP-SRN $\mathcal{R}_{\theta+\vartheta}$ are denoted by $\mathring{\mathbf{h}}$ and $\mathring{\mathbf{y}}$.*

The following theorem makes it possible, under certain conditions, to limit the deviation between an SRN and its perturbed version.

**Theorem 14** *Let $\mathcal{R}_\theta$ be a $\tilde{\zeta} + \tilde{\eta}$-saturated FP-SRN, with $z = \frac{1}{\|\mathbf{W}\|}$ and $\tilde{\zeta} = \ln\left(\frac{1+\sqrt{1-4z}}{1-\sqrt{1-4z}}\right)$, for some $\tilde{\eta} > 0$. We assume that $\mathcal{R}_\theta$ is operating on one-hot encoded data. We select any real number $t$ such that $0 < t < 1$,*

*then for all $\vartheta := \mathbf{vec}(\vartheta_{\mathbf{U}}, \vartheta_{\mathbf{W}}, \vartheta_{\mathbf{V}}, \vartheta_{\mathbf{b}}, \vartheta_{\mathbf{c}}) \in \mathbb{R}^{dim(\theta)}$ such that $\|\vartheta\| \leq (1-t)\tilde{\eta} \cdot \frac{(1-\tilde{\Delta}\|\mathbf{W}\|)}{(2+\sqrt{h})}$, with $\tilde{\Delta} = \sigma'(\tilde{\zeta}+t\tilde{\eta})$ and for all $\omega \in \Sigma^*$ with $|\omega| = T$ we have:*

*1. $\|\mathring{\mathbf{h}}_k - \mathbf{h}_k\|_\infty \leq \frac{(2+\sqrt{h})}{1-\Delta\|\mathbf{W}\|}\|\vartheta\|$ where $1 \leq k \leq T$.*

*2. $\mathcal{R}_{\theta+\vartheta}$ is $(\tilde{\zeta}+t\tilde{\eta})$-saturated.*

*The sequence of hidden state vectors $\mathring{\mathbf{h}}_k$ is produced by the perturbed FP-SRN $\mathcal{R}_{\mathbf{p}+\vartheta}$.*

We recall all the proofs of this theorem in Appendix B. The idea is that it is possible to express $\mathring{\mathbf{h}}_k$ as a sum of $\mathbf{h}_k$ and $G(\vartheta, k)$, where $G(\vartheta, k)$ is a function of the perturbation $\vartheta$ and the iteration $k$ and takes the form of a sum of $k$ elements. Mitarchuk et al. (2024) exploit the saturation assumption to upper-bound $G(\vartheta, k)$ with a convergent series and hence show that the perturbation injected at every step cannot be endlessly amplified. This proves the first claim, after what the second claim is deduced from the first one.

## 4 Finite Precision SRN and Learnability

In this section, we discuss gradient failure for RNNs training. First, we describe the algorithm of gradient propagation in RNNs called Back Propagation Trough Time (BPTT) and derive a representation of the gradient as a sum of products. Then, we formally define the stationary gradient, after what we state our central result. Finally we apply this theorem to a concrete case and discuss the different variables involved in the theorem.

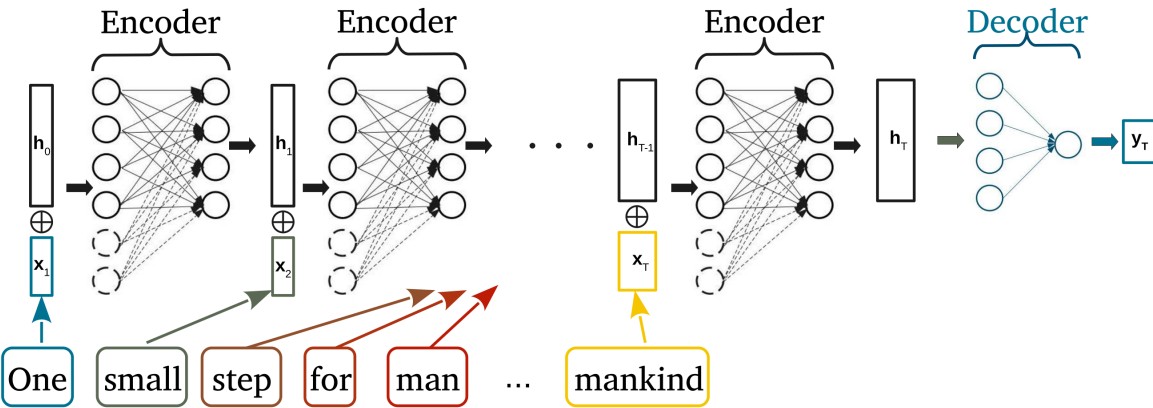

Figure 1: Illustration of the Encoder Decoder architecture.

### 4.1 Back Propagation Trough Time

To train neural networks the usual approach is the one of Gradient Descent (GD). Its classical implementation for RNNs is called Back Propagation Trough Time (BPTT) which we detail in this section. There are many ways from which one can consider neural networks and BPTT. Switching from one representation to another provides an in-depth and intuitive understanding of the mechanics of BPTT. We start by setting out the machine learning framework within which we will develop our arguments, and then use the various neural network representations to obtain a gradient representation that will enable us to prove our assertion.

We place ourselves in the context of binary classification on sequences. Let $\mathcal{R}_\theta$ be a FP-SRN and $S = \{(\omega_1, y_1), \dots (\omega_N, y_N)\}$ a labeled training set with $N$ samples and where $\omega_l = \{\mathbf{x}_k\}_{k=1}^T \subset \mathbb{G}^u$ is a finite sequence of vectors (the vectors $\mathbf{x}_k$ can represent vector embedding of words) with $1 \leq l \leq N$ and $T \geq 1$ an integer representing the length of the word $\omega_l$. We set $\mathcal{L} : \mathbb{G} \times \mathbb{G} \to \mathbb{G}$ to be a differentiable loss function.

For a $(\omega, y) \in S$ we update the parameters of $\mathcal{R}_\theta$ using $(\omega, y)$, by computing $\nabla_\theta \mathcal{L}(\mathcal{R}(\omega), y)$ as stated in Definition 10. In order to obtain the output of the FP-SRN on $\omega$, $\mathcal{R}_\theta(\omega)$, we have to compute recursively $\mathbf{h}_T$

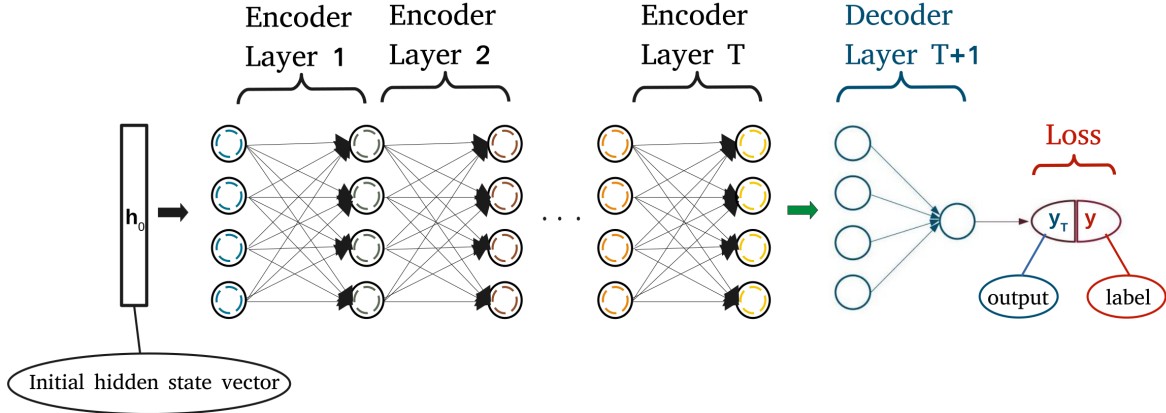

Figure 2: Illustration of the unrolling of the RNN for the BPTT.

and then $\mathcal{D}(\mathbf{h}_T)$, with $\mathbf{h}_{k+1} = \mathcal{E}(\mathbf{x}_{k+1}, \mathbf{h}_k)$ for $1 \leq k < T$. A way of seeing the recursive computation of the RNN is to unroll the calculation so that we obtain a pseudo feed-forward neural network $\mathcal{R}^\omega$ derived from the evaluation of $\mathcal{R}_\theta$ on $\omega$ with $(T+1)+1$ layers, which includes the decoder layer and the loss function layer $\mathcal{L}(\mathcal{D}(\mathbf{h}_T), y)$. Each parameter of this pseudo feed-forward network $\mathcal{R}^\omega$ is then updated by back propagating the gradient. The difference between $\mathcal{R}^\omega$ and a real feed-forward network is that the parameters do not change from one layer to the other. To give a more formal definition of what we mean by pseudo feed-forward network $\mathcal{R}^\omega$, let us consider the notations of Definition 5, allowing us to write $\mathbf{h}_T$ as the output of a $T$ layer feed-forward neural network:

$$\mathbf{h}_T = \sigma\left(\mathbf{W}\sigma\left(\mathbf{W}\sigma\left(...\sigma\left(\mathbf{W}\mathbf{h}_0 + \mathbf{U}\mathbf{x}_1 + \mathbf{b}\right)...\right) + \mathbf{U}\mathbf{x}_{T-1} + \mathbf{b}\right) + \mathbf{U}\mathbf{x}_T + \mathbf{b}\right)$$

$$= \sigma\left(\mathbf{W}\sigma\left(\mathbf{W}\sigma\left(...\sigma\left(\mathbf{W}\mathbf{h}_0 + \mathbf{b}_1\right)...\right) + \mathbf{b}_{T-1}\right) + \mathbf{b}_T\right).$$

The second equality has the form of a regular architecture of feed-forward neural network where we alternate between an affine function and the activation function. In this representation, we have encoded the sequential input $\omega = \{\mathbf{x}_1, \mathbf{x}_2, \ldots, \mathbf{x}_T\}$, in the vectors $\mathbf{b}_k$ which are defined by $\mathbf{b}_k = \mathbf{U}\mathbf{x}_k + \mathbf{b}$. Obviously, during the training the parameters of the matrix $\mathbf{U}$ and those of the bias vector $\mathbf{b}$ will be taken into account by the algorithm.

In Figure 1, we have made a distinction between the parameters of the transition kernel, which are represented by full arrows, and the parameters of the input kernel, which are represented by dotted arrows. We also represented the embedding of the sentence "One small step for man ... humanity" with a color code representing the order of the words in the sentence. Thus in Figure 2 we have represented the integration of the input kernel evaluated on the words in the pseudo feed-forward network parameters by representing the neurons by full and dotted concentric circles with the coloration of the embedding of the words. Figure 2 illustrates the unrolling procedure of an SRN of hidden dimension 4 evaluated on a sentence.

Figures 1 and 2 represent the neural network as a graph where the circles are the activation function $\sigma$ and the arrows represent the weighted connections realized by the matrix $\mathbf{M}_h$. Rojas & Rojas (1996) explain the back propagation for deep neural networks with sigmoidal activation. They build on the graph representation of the neural network to construct the back propagation algorithm. The first phase is the forward pass, *i.e.* the computation of $\mathcal{L}(\mathcal{R}^\omega, y)$. Secondly, we compute, recursively, the gradient for the parameters involved in

every layer of the network. The gradient propagation can be seen as tracking the impact of a parameter on the final result $\mathcal{L}(\mathcal{R}^\omega, y)$. In the context of gradient calculation in a directed graph, tracing the impact of a parameter is achieved by observing all the paths in the graph that link the loss to the parameter in question and assessing the impact of the parameter on each of these paths. We emphasize that every layer, except the last one, of the network $\mathcal{R}^\omega$ uses the same set of parameters, what we called the pseudo feed-forward network. This implies that every parameter of $\mathbf{M}_h$ will produce a partial derivative composed of a variety of different impacts that the parameter may have had on the computation of $\mathcal{L}(\mathcal{R}^\omega, y)$. Our goal is to derive a convenient representation of the gradient that leads us to consider the fact that the network is composed of weighted sums and differentiable scalar functions (building blocks of the neuron in Definition 6). One can deduce from the chain rule, that the partial derivative of $\mathcal{L}(\mathcal{R}^\omega, y)$ with respect to a parameter $m_{i,j}$ can be represented as a sum of products, where we sum over the set of paths linking the result $\mathcal{L}(\mathcal{R}^\omega, y)$ and the parameter $m_{i,j}$ within the network, and the products are composed of the weighted differentiated activations on a path. More formally for a parameter $m_{i,j}$ we define a path $\gamma = (\gamma_1, \ldots, \gamma_k)$ of length $k$ for $1 \le k \le T$ as a finite sequence such that for $k = 1$ we have $\gamma_1 = \mathbf{M}_o[i]$, for $k = 2$ we have $\gamma_1 = \mathbf{M}_o[v]$ and $\gamma_2 = m_{v,i}$ for some $1 \le v \le h$. For the case $k \ge 3$ we define $\gamma$ such that $\gamma_k = m_{v,i}$ for some $1 \le v \le h$, for all $1 < s < k$ $\gamma_{s-1} = m_{l,t}$, $\gamma_s = m_{t,r}$ and $\gamma_{s+1} = m_{r,w}$. In other words $\gamma$ is an ordered sequence that contains the information about a path followed within the network $\mathcal{R}^\omega$ from the loss to the neuron hosting the parameter $m_{i,j}$. We can gather all such path $\gamma$ for the parameter $m_{i,j}$ in the set $\Gamma_{i,j}$, what allows us to define a partial derivative $\frac{\partial \mathcal{L}(\mathcal{R}^\omega, y)}{\partial m_{i,j}}$ as follows:

$$\frac{\partial \mathcal{L}(\mathcal{R}^\omega, y)}{\partial m_{i,j}} = \frac{\partial \mathcal{L}(\mathcal{R}^\omega, y)}{\partial x} \cdot \sigma'_{\mathbf{M}_o}(\mathbf{h}_T) \sum_{\gamma \in \Gamma_{i,j}} \prod_{s=1}^{k} \gamma_s \cdot \sigma'_{\gamma_s}(\mathbf{x}_{T-s} \oplus \mathbf{h}_{T-s-1} \oplus 1). \tag{2}$$

where $\frac{\partial \mathcal{L}(\mathcal{R}^\omega, y)}{\partial x}$ represents the partial derivative of $\mathcal{L}$ with respect to the first variable. We abuse a little bit the notation of the neuron here because $\gamma_s$ refers to a parameter in the matrix $\mathbf{M}_h$ and not to a row vector, but since a unique row vector contains the parameter $\gamma_s$ we allow ourselves to denote a neuron in this manner. Equation 2 is not new, in fact it can be deduced from the back propagation algorithm of Rojas & Rojas (1996). We recall that our goal is to show that some regions of the parameter space of FP-SRN cannot be reached by gradient descent. The idea is to exhibit regions of the parameter space where some coordinates of the gradient $\nabla \mathcal{L}(\mathcal{R}(\omega), y)$ will have a negligible value with respect to the parameters, leading to $\theta - \alpha \cdot \nabla \mathcal{L}(\mathcal{R}(\omega), y) = \theta$ in finite precision. To achieve this we will use Equation 2 to prove an upper bound on $\left| \frac{\partial \mathcal{L}(\mathcal{R}^\omega, y)}{\partial m_{i,j}} \right|$ that we can exploit in finite precision arithmetic's. We have used a lot of formalism for the definition of the gradient, which is necessary for the rigor of our work but will not really be used in the proofs, because we will state hypotheses on $\mathcal{R}_\theta$ that will allow us to derive a bound that does not depend on $T$, vectors $\mathbf{x}_{T-s} \oplus \mathbf{h}_{T-s-1} \oplus 1$ and parameters $m_{l,t}$.

We define the notion of stationary gradient in finite precision. A condition characterized by the fact that, for some parameters, adding the gradient will not change the parameter due to rounding in finite precision.

**Definition 15 (Stationary Gradient)** *Let $(B, M, X)$ be a finite precision configuration, $\mathcal{R}_\theta$ be a RNN with parameter vector $\theta$, a loss function $\mathcal{L}$, a training set $S = \{(\omega_1, y_1), \ldots, (\omega_N, y_N)\}$ with $N \ge 1$ training samples and $\alpha > 0$ a positive real number called the learning rate. We say that $\mathcal{R}_\theta$ experiences a stationary gradient if there is a non empty subset $P$ of coordinates in $\theta$ such that for all $m_{i,j} \in P$ and for all $(\omega, y) \in S$ we have in finite precision arithmetic:*

$$m_{i,j} - \alpha \frac{\partial \mathcal{L}}{\partial m_{i,j}}(\mathcal{R}(\omega), y) = m_{i,j}.$$

## 4.2 A Condition for BPTT Failure

In this subsection we develop our argument of BPTT failure in a finite precision framework. The idea of this theorem is: if the parameters that we try to reach by GD have the properties stated by the theorem (we will show that fully saturated FP-SRNs fulfill these properties) then the gradient will fail to update the parameters before reaching the target.

**Theorem 16** *Let $\mathbb{G}$ be the set of all float numbers representable in a given finite precision configuration $(B, M, X)$. Let $S = \{(\omega_1, y_1), \ldots (\omega_N, y_N)\}$ be a training set with $N$ samples. For $1 \leq l \leq N$, $\omega_l = \{\mathbf{x}_k\}_{k=1}^{T} \subset \mathbb{G}^u$ is a finite sequence of length $T$ where $\mathbf{x}_k$ are vectors verifying $\|\mathbf{x}_k\| \leq 1$ for $1 \leq k \leq T$ and $T \geq 1$ is the integer representing the length of the word $\omega_l$. We set $\mathcal{L} : \mathbb{G} \times \mathbb{G} \to \mathbb{G}$ to be a differentiable loss function. Let $\alpha > 0$ the learning rate as used in Definition 10. Let $\mathcal{R}_\theta$ be an FP-SRN with hidden dimension $h$, input dimension $u$ and output dimension $o = 1$ i.e. $\mathcal{R}_\theta$ is shaped for binary classification on strings. We recall that the row vectors of the matrix $\mathbf{M}_h$ are denoted $\mathbf{m}_i$ for $1 \leq i \leq h$, and that since the output dimension $o = 1$ it means that the matrix $\mathbf{M}_o$ is of dimension $(1, h)$. We set*

$$m^* := \max\left\{|m| \ : \ m \in \theta\right\}$$
$$\underline{m} := \min\left\{|m| \ : \ m \in \theta \text{ and } |m| > 0\right\},$$

*that is $m^*$ is the largest parameter in absolute value and $\underline{m}$ is the smallest, in absolute value, non zero parameter. We set $\varepsilon = B^D$ with $D = \lfloor \log_B(\underline{m}) \rfloor - (M + 1)$. We suppose that:*

1. *For all $y \in \{0, 1\}$ the function $[0, 1] \ni x \mapsto \left|\frac{\partial}{\partial x}\mathcal{L}(x, y)\right|$ is bounded by a constant $\rho_L$.*

2. *$\exists \ \eta > 0$ such that $\mathcal{R}_\theta$ is $(\zeta + \eta)$-saturated where*

$$\zeta = \ln\left(\frac{1 + \sqrt{1 - 4z}}{1 - \sqrt{1 - 4z}}\right) \ , z = \frac{4}{\psi h \rho_L}\frac{\varepsilon}{1 + \varepsilon} \ , \psi := \alpha \cdot \max\{(m^* + \eta/2), \|\mathbf{W}\|\}$$

*Then with $\Delta := \sigma'(\zeta + \frac{\eta}{2})$ and with $\mathbf{W}$ (the transition kernel extracted from $\mathbf{M}_h$) for all $\vartheta \in \mathbb{R}^{dim(\theta)}$ such that $\|\vartheta_{Encoder}\| \leq \frac{\eta}{2} \cdot \frac{(1 - \Delta\|\mathbf{W}\|)}{(2 + \sqrt{h})}$ the FP-SRN $\mathcal{R}_{\theta + \vartheta}$ will experience a stationary gradient for all non zero parameters on the training data set $S$.*

Note that this result does not include the parameters in the decoder's part of the FP-SRN. The assumptions on $\zeta$ overcome the information transmitted through the decoder's gradient. The consequence of this is that one can change the decoders parameters completely, it will have no effect on the encoder parameters gradient.

A few words to comment on the assumptions and quantities used in the theorem. The idea behind this result is that a FP-SRN $\mathcal{R}$ satisfying hypotheses 1 and 2 will have at least two behaviors of interest to us: A) robustness to noise, B) stationary gradient. The robustness to noise is essential in this result, as we aim to exhibit a whole population of RNNs that have similar behavior to $\mathcal{R}$. Quite naturally, when it comes to perturbation, we have to make assumptions about the Lipschitzian constants $\rho_L$ of functions that form the networks. This explains assumption number 1, which ensures that the loss function will not explode if the networks parameters are perturbed. Assumption 2 is more central to this work. This is the assumption of saturation of the activation function, *i.e.* how close the activation function is to its bounds. This assumption is essential, as it both demonstrates the stability of the RNN with respect of the noise injection, and squashes the gradient to zero. The $\zeta$ quantity acts as a threshold below which the guarantee of behaviors A) and B) are lost. More precisely, $\zeta$ is the solution to the equation $\sigma'(\zeta) = \frac{4}{\psi h \rho_L}\frac{\varepsilon}{1 + \varepsilon}$. Since $\sigma'(x) = \sigma(x)(1 - \sigma(x))$ *i.e.* an equation of second degree in $\sigma(x)$ and that $\sigma$ is a function defined with the exponential function, one can deduce the appearance of a square root and the application of the logarithm function in the definition of $\zeta$. For a formal proof please refer to Lemma 18 in Appendix A. The desired quantity $\frac{4}{\psi h \rho_L}\frac{\varepsilon}{1 + \varepsilon}$ to achieve with $\sigma'(\zeta)$ is built such that it is possible to bound Equation 2 by $\varepsilon$. On the other hand, the constant $\eta$ has to be seen as a budget for the perturbation that $\mathcal{R}_\theta$ can handle without loosing the saturation property. All the proofs articulate around well-defining the border $\zeta$, and wisely managing the budget $\eta$.

For readability reasons, the complete proof of the theorem is in Appendix A of this paper. Nevertheless, we propose a sketch of the proof. At first, we prove that a $(\zeta + \eta)$-saturated FP-SRN is stable to perturbations with Theorem 14 proven in Appendix B. Indeed, one can disturb the parameter of $\mathcal{R}_\theta$ by adding a noise vector $\vartheta$ to the parameter vector $\theta$ and be sure that the output of $\mathcal{R}_{\theta + \vartheta}$ will remain within a controllable distance of $\mathcal{R}_\theta$. On top of that, for a well chosen $\vartheta$ we can be sure that $\mathcal{R}_{\theta + \vartheta}$ will be $(\zeta + \eta/2)$-saturated (*i.e.* for $\|\vartheta_{Encoder}\| \leq \frac{\eta}{2}\frac{1 - \Delta\|\mathbf{W}\|}{2 + \sqrt{h}}$). After this first step, we proceed to prove that the gradient will be rounded to zero (the stationary gradient property), by using the gradient expression in Equation 2. In the following subsections we provide some concrete examples based on a set of parameters.

### 4.3   An example of a non learnable FP-SRN

In this section we apply our results on a small example. DFA suit particularly well our case, because they model binary functions on sequences. Figure 3 is a representation of a DFA accepting the language of all words that contain an odd number of $a$'s on the alphabet $\Sigma = \{a, b\}$ (the initial state is 1, the sole accepting state is 2).

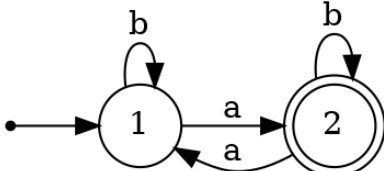

Figure 3: A DFA accepting the language composed of words containing an odd number of $a$'s. The initial state is drawn with an incoming arrow with no starting state, the finale state by a double circle, and the transitions with labelled arrows.

Let $(B, M, X)$ be the finite precision configuration. We prove in Appendix D Lemma 3 that there exists a positive float number $J$ such that $\sigma(-J) = 0$, $\sigma(J) = 1$ and for all positive float number $s$ such that $J - s < J$ we have $\sigma(-(J-s)) > 0$. The number $J$ is called the underflow barrier float.

Then we define:

$$\mathbf{h}_0 := \begin{pmatrix} 1 \\ 0 \\ 0 \\ 0 \end{pmatrix}, \mathbf{w}^1 := \begin{pmatrix} -3 \\ -1 \\ -1 \\ -3 \end{pmatrix}, \mathbf{w}^2 := \begin{pmatrix} -1 \\ -3 \\ -3 \\ -1 \end{pmatrix}, \mathbf{M}_o := J \cdot \begin{bmatrix} -1 & -1 & 1 & 1 \end{bmatrix}$$

and:

$$\mathbf{U} = 2J \cdot \begin{bmatrix} 1 & 0 \\ 0 & 1 \\ 1 & 0 \\ 0 & 1 \end{bmatrix} \qquad \mathbf{W} = -J \cdot \begin{bmatrix} 3 & 3 & 1 & 1 \\ 1 & 1 & 3 & 3 \\ 1 & 1 & 3 & 3 \\ 3 & 3 & 1 & 1 \end{bmatrix}$$

We consider the computation of the FP-SRN on the word $a$ and the word $b$. Their one-hot encoding is $\phi(a) = \begin{pmatrix} 1 \\ 0 \end{pmatrix}$ and $\phi(b) = \begin{pmatrix} 0 \\ 1 \end{pmatrix}$.

We have:

$$\mathbf{U}\phi(a) + \mathbf{W}\mathbf{h}_0 = J \cdot \left( \begin{pmatrix} 2 & 0 & 2 & 0 \end{pmatrix} - \begin{pmatrix} 3 & 1 & 1 & 3 \end{pmatrix} \right)^T$$
$$= J \cdot \begin{pmatrix} -1 & -1 & 1 & -3 \end{pmatrix}^T.$$

After applying the activation function we obtain $\mathbf{h}_1 = \begin{pmatrix} 0 & 0 & 1 & 0 \end{pmatrix}^T$ due to the overflow and underflow. The vector $\mathbf{h}_1$ is an accepting vector because $\sigma(\mathbf{M}_o\mathbf{h}_1) = \sigma(J) = 1$. If we do the same computation for $b$ we

obtain $\mathbf{h}_1' = \begin{pmatrix} 0 & 1 & 0 & 0 \end{pmatrix}^T$. The vector $\mathbf{h}_1'$ is not an accepting vector because $\sigma(\mathbf{M}_o \mathbf{h}_1') = \sigma(-J) = 0$ due to the underflow. We draw the readers attention to fact that this FP-SRN is $J$-saturated by construction. In Section 5 we propose an algorithm that for any DFA, will return a FP-SRN that is $J$-saturated and that simulates the DFA. But first we propose to study the example above from the perspective of Theorem 16.

### 4.4 On the learnability of this FP-SRN

In the following subsections we apply Theorem 16 to the FP-SRN defined just above. We will gradually show that the FP-SRN of the example respects the assumptions of Theorem 16 and explain the magnitude of each variable involved in that theorem.

At first we assume to have $(B, M, X) = (2, 23, 8)$ the usual finite precision configuration of 32 bit float numbers, that we convert to base 10, in order to have readable constants. In this configuration we have $J = 88.7228391117$. When we switch from base $B = 2$ to base $\tilde{B} = 10$, the number of digits needed to encode all the floats in $\mathbb{G}_{(B,M,X)}$ does not exceed 15, so when we switch to base $\tilde{B} = 10$ we get that $\tilde{M} = 15$. In other words, we have $\mathbb{G}_{(2,23,8)} \subset \mathbb{G}_{(10,15,6)}$. The constant $M$ is necessary for the definition of $\varepsilon$, the value that must upper bound the coordinates of the gradient in order for it to be rounded to zero. Therefore, in the following, since we switched to basis 10, we will use $\tilde{M} = 15$ to define $\varepsilon$. The FP-SRN presented above in Section 4.3 is $J$-saturated by definition. By taking a closer look to the matrix $\mathbf{M}_h = \begin{bmatrix} \mathbf{U} & \mathbf{W} \end{bmatrix}$ one can deduce that for all $1 \le i \le h$ and all $1 \le j \le u + h + 1$ we have $|\mathbf{m}_{i,j}| \le 3J$. Also the smallest non zero parameter in $\mathbf{M}_h, \mathbf{M}_o$ is $\underline{m} = J$, Therefore we obtain:

$$m^* = 3 \cdot J \approx 2.7 \cdot 10^2$$

$$\lfloor \log_{10}(\underline{m}) \rfloor = \lfloor \log_{10}(J) \rfloor = \log_{10}(10^2) = 2$$

$$\varepsilon = 10^{2-(15+1)} = 10^{-14} \ .$$

We remind that $\varepsilon$ defined above represents the threshold such that for all float numbers $m \le \varepsilon$ will have no effect when added to non zero parameters of the FP-SRN $\mathcal{R}_\theta$.

### 4.5 How $h$, $\varepsilon$, and $\rho_L$ affects $\zeta$ ?

In Theorem 16, the quantity $\zeta$ has a central role, it delimits a region where we have the guarantee that $\mathcal{R}_\theta$ is stable to noise and will experience a stationary gradient. In the case that we consider now, $\mathcal{R}_\theta$ is $J$-saturated with $\zeta + \eta = J$, knowing $\zeta$ allows us to know exactly $\eta$ (a constant that represents the budget for the perturbation) and thus the size of the Euclidean ball containing FP-SRNs with stationary gradient. The variable $\zeta$ is directly linked to $m^*$, $\varepsilon$, $\rho_L$ and $\alpha$ the learning rate. In our case $m^* = 267$. For this study we will represent the size of the hidden state vector $h = 10^n$ for $n \in [0, 11]$, because it is the magnitude of $h$ that will mostly impact the computations. We assume to use $\mathcal{L}_\epsilon$ as the loss function with, $\epsilon = 10^{-7}$, a default quantity in tensorflow.keras platform for numerical stability. It comes that we have $\rho_L = 10^7$, and for simplicity we assume to have $\alpha = 1$.

The first quantity we want to discuss is $\psi$:

$$\begin{aligned} \psi &= \alpha \cdot \max\{(m^* + \eta/2), \|\mathbf{W}\|\} \\ &= \max\{(m^* + \eta/2), \|\mathbf{W}\|\} \\ &\le m^* \cdot \sqrt{h} \text{ for } h \ge 2 \\ &= (267 \cdot 10^{\frac{n}{2}}) \le 2.7 \cdot 10^{\frac{n}{2}+2}. \end{aligned}$$

The inequality above comes from $\|\mathbf{W}\| \le \|\mathbf{W}\|_1 \le m^* \cdot \sqrt{h}$, and we assume that $m^* \ge \eta/2$. In the case of the variable $\psi$ an upper bound is enough for us because it will give us an upper bound for $\zeta$. This allows us to estimate the magnitude of the quantity $z = \frac{4}{\psi h \rho_L} \frac{\varepsilon}{1+\varepsilon}$, First we have $1 + \varepsilon = 1 + 10^{-14} \approx 1$, hence $\frac{\varepsilon}{1+\varepsilon} \approx \varepsilon$. Then we have:

$$\frac{4}{\psi h \rho_L} \ge \frac{4}{2.7 \cdot 10^{\frac{n}{2}+2} 10^n 10^7} = 1.48 \cdot 10^{-2-\frac{3n}{2}},$$

$$\frac{4}{\psi h \rho_L} \frac{\varepsilon}{1+\varepsilon} \approx \frac{4 \cdot 10^{-14}}{2.7 \cdot 10^{\frac{n}{2}+2} 10^n 10^7} = 1.48 \cdot 10^{-16-\frac{3n}{2}}.$$

By using the Taylor expansion of $\sqrt{1+x} = 1 + x/2 + o(x^2)$ for $x \approx 0$, we deduce that

$$\zeta = \ln\left(\frac{1 + \sqrt{1-4z}}{1 - \sqrt{1-4z}}\right)$$

$$\approx \ln\left(\frac{2-2z}{2z}\right)$$

$$\approx \ln\left(\frac{1}{1.48}\right) + (16 + \tfrac{3n}{2})\ln(10) + \ln\left(1 - 1.48 \cdot 10^{-16-\frac{3n}{2}}\right)$$

$$\approx \ln\left(\frac{1}{1.48}\right) + (16 + \tfrac{3n}{2})\ln(10) + 0$$

$$\leq (16 + \tfrac{3n}{2})\ln(10) \text{ because } \ln\left(\frac{1}{1.48}\right) < 0.$$

If we consider cases where $1 < h = 10^n \leq 10^{10}$ then we have that the boundary $\zeta$ is in the interval $\zeta \in [38, 72]$. We recall that $\zeta$ represents a boundary beyond which we have the guarantee that $\mathcal{R}_\theta$ is stable to noise in parameters and that this FP-SRN will experience a stationary gradient. It is important to note that even with very hard requirements on the gradient (*i.e.* the coordinates of the gradient have to be smaller than $10^{-21}$), it is sufficient for a FP-SRN to be saturated with values under 72. In other words, by Theorem 16 it is impossible to train FP-SRN by gradient descent such that the linear part will output vectors with coordinates further than 72 from zero.

### 4.6   How big is the perturbation radius $\frac{\eta}{2}\frac{1-\Delta\|\mathbf{W}\|}{2+\sqrt{h}}$?

In this part we discuss the quantities $1 - \Delta\|\mathbf{W}\|$ and $\frac{\eta}{2}\frac{1-\Delta\|\mathbf{W}\|}{2+\sqrt{h}}$. All the computations are based on the hypothesis that $\mathcal{R}_\theta$ is defined as in Section 4.3. Our goal here is to estimate how these quantities affect the radius of the Euclidean ball around the parameters $\mathbf{M}_h$ that fulfill the assumptions of Theorem 16. We recall that the matrix $\mathbf{W}$ is the transition kernel extracted from $\mathbf{M}_h$, and that $\Delta := \sigma'(\zeta + \eta/2)$, where $\zeta + \eta = J \approx 88.72$. We start by analyzing the quantity $\Delta\|\mathbf{W}\| = \sigma'(\zeta + \eta/2)\|\mathbf{W}\|$. Lemma 19 in Appendix A tells us that

$$\sigma'(\zeta + \eta/2) \leq \sigma'(\zeta) + (\eta/2)e^{-(\zeta-\eta/2)}.$$

This result is based on the Taylor expansion of $\sigma'$. From the computations above one can deduce that:

$$\zeta \approx (16 + \tfrac{3n}{2})\ln(10)$$

$$\eta/2 = \frac{1}{2}(J - \zeta)$$

$$\approx 25.68 - \tfrac{3n}{4}\ln(10).$$

We have also $\|\mathbf{W}\| \leq m^* \cdot h \approx 2.7 \cdot 10^{n+2}$, leading us to:

$$\Delta\|\mathbf{W}\| = \sigma'(\zeta + \eta/2)\|\mathbf{W}\|$$

$$\leq \sigma'(\zeta)\|\mathbf{W}\| + m^* \cdot h(\eta/2)e^{-(\zeta-\eta/2)}.$$

By Lemma 17 we have that $\sigma'(\zeta) = \frac{4}{\psi h \rho_L}\frac{\varepsilon}{1+\varepsilon}$. We recall that $\psi = \max\{(m^* + \eta/2), \|\mathbf{W}\|\}$ Therefore:

$$\sigma'(\zeta)\|\mathbf{W}\| \leq \frac{4}{\psi h \rho_L}\frac{\varepsilon}{1+\varepsilon}\|\mathbf{W}\| \leq \frac{4}{h\rho_L}\frac{\varepsilon}{1+\varepsilon} \approx \frac{4\varepsilon}{h\rho_L} = 4 \cdot 10^{-21-n}$$

The quantity $m^* \cdot h(\eta/2)e^{-(\zeta-\eta/2)}$ requires to plug into the exponential function the value of $\zeta - \eta/2$ that we have introduced above. We obtain that:

$$m^* \cdot h(\eta/2)e^{-(\zeta-\eta/2)} \approx 2.7 \cdot 10^{n+2} \cdot (25.68 - \frac{3n}{4}\ln(10)) \cdot (10^{-11.15-\frac{9n}{4}})$$

$$= 2.7 \cdot (25.68 - \frac{3n}{4}\ln(10)) \cdot 10^{-9.15-\frac{5n}{4}}.$$

Clearly the quantity $\Delta\|\mathbf{W}\|$ is upper bounded by a sum of quantities negligible with respect to 1. Therefore we can assume that $1 - \Delta\|\mathbf{W}\| \approx 1$, hence $\frac{\eta}{2}\frac{1-\Delta\|\mathbf{W}\|}{2+\sqrt{h}} \approx \frac{\eta}{4+2\sqrt{h}}$.

The previous results lead us the conclusion that the quantity that influences the most $\frac{\eta}{2}\frac{1-\Delta\|\mathbf{W}\|}{2+\sqrt{h}}$ is $h = 10^n$. From the previously established representations of $\zeta, \eta/2$ we can deduce that:

$$\frac{\eta/2}{2+\sqrt{h}} \approx \frac{25.68 - \frac{3n}{4}\ln(10)}{2 + 10^{\frac{n}{2}}}.$$

In order to visualize the evolution of the perturbation radius with respect to $h$, we define the radius function $r(n) := \frac{25.68 - \frac{3n}{4}\ln(10)}{2 + 10^{\frac{n}{2}}}$ and we plot this function with $n \in \mathbb{R}_+$ in Figure 4. We have highlighted the particular value of the radius function at 0.602 for the reason that for this value of $n$ we have $10^n = 4$ i.e. the number of neurons that the FP-SRN presented in Section 4.3 has. Hence the reader can see that with our results we obtain that the perturbation radius in this example is 6.16. We will see in a next paragraph how such perturbation can impact the network performance. Finally, note that SRNs with more than $10^3$ neurons are rarely observed, so this last constraint on the number of neurons is realistic. Indeed, based on the survey of Lara-Benitez et al. (2021), we identified 3 works published between 2012 and 2018 that employ SRNs ((Chandra & Zhang, 2012), (Rueda & Pegalajar, 2018), and (Mohammadi et al., 2018)). The information about the number of neurons is missing in (Mohammadi et al., 2018), while the number of hidden neurons is less than $10^3$ in the two other papers. The most recent use of SRNs, dating from 2023, is in the TAYSIR competition (Eyraud et al., 2023), where a benchmark of already trained models was presented to competitors. The benchmark is composed of different RNN architectures and in particular, SRNs were proposed whose number of hidden neurons did not exceed 512.

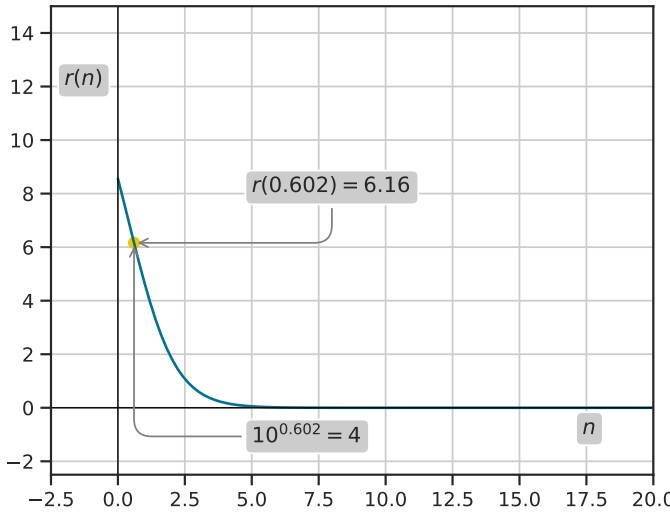

Figure 4: This figure displays the radius function $r(n)$ where $n = \ln_{10}(d)$ the logarithm in base 10 of the number of hidden neurons. The yellow dot represents the perturbation radius for the example 4.3

## 4.7 Stationary gradient even with $\mathcal{L}_\epsilon \neq 0$

Just above we established that the FP-SRN $\mathcal{R}_\theta$ from Section 4.3, within the requirement of Theorem 16, has a perturbation radius of 6.16. In other words, for all $\vartheta \in \mathbb{R}^{dim(\theta)}$ such that $\|\vartheta_{Encoder}\| \leq 6.16$ the FP-SRN $\mathcal{R}_{\theta+\vartheta}$ will experience a stationary gradient. In Table 5 we present the variables of interest for $\mathcal{R}_\theta$. Theorem 16 does not depend on how the perturbation is spread among the parameters of the FP-SRN

| $h$ | $\zeta$ | $\eta$ | $r(n)$ |
|---|---|---|---|
| 4 | $\approx 38.92$ | $\approx 49.8$ | $\approx 6.16$ |

Figure 5: Statistics of the FP-SRN from Section 4.3 in usual 32 bit float precision

and does not depend on the decoders parameters. We present here an example where the noise is spread randomly across the parameters of the transition kernel:

$$\mathring{\mathbf{U}} := \mathbf{U} = 2J \cdot \begin{bmatrix} 1 & 0 \\ 0 & 1 \\ 1 & 0 \\ 0 & 1 \end{bmatrix} \quad \mathring{\mathbf{W}} := -J \begin{bmatrix} 3.07 & 2.90 & 1.15 & 0.71 \\ 0.49 & 1.06 & 3.0 & 2.88 \\ 0.86 & 0.79 & 2.93 & 3.43 \\ 3.09 & 3.24 & 1.33 & 0.59 \end{bmatrix} \quad \mathring{\mathbf{M}}_o := 2 \cdot \begin{bmatrix} -1 & -1 & 1 & 1 \end{bmatrix}$$

By design of the perturbation we have $\|\mathring{\mathbf{W}} - \mathbf{W}\|_{\text{Fro}} = 6.16$. We execute this perturbed FP-SRN on the word $a$ that is one-hot encoded as follows $\phi(a) = \begin{pmatrix} 1 \\ 0 \end{pmatrix}$. We obtain:

$$\mathring{\mathbf{U}}\phi(a) + \mathring{\mathbf{W}}\mathbf{h}_0 = J \cdot \left( \begin{pmatrix} 2 & 0 & 2 & 0 \end{pmatrix} - \begin{pmatrix} 3.07 & 0.49 & 0.86 & 3.09 \end{pmatrix} \right)^T$$

$$= J \cdot \begin{pmatrix} -1.07 & -0.49 & 0.86 & -3.09 \end{pmatrix}^T$$

After applying the activation function $\sigma$ we obtain the hidden state vector $\mathring{\mathbf{h}}_1 = \begin{pmatrix} 0 & \alpha & 1 & 0 \end{pmatrix}^T$, with $\alpha \approx 1.316 \cdot 10^{-19}$. Now we apply the decoder on $\mathring{\mathbf{h}}_1$, and obtain:

$$\sigma(\mathring{\mathbf{M}}_o\mathring{\mathbf{h}}_1) = \sigma(2 \cdot (1 - \alpha))$$

$$\approx 0.88$$

By definition of the DFA in the example, it accepts all the words with an odd number of $a$'s, Therefore the word $a$ is labeled with 1. One can compute the loss in this particular case and obtain that:

$$\mathcal{L}_\epsilon((\mathcal{R}_{\theta+\vartheta}(\phi(a)), 1)) = -1 \cdot \log(\sigma(\mathring{\mathbf{M}}_o\mathring{\mathbf{h}}_1) + 10^{-7}) \approx -\log(0.88 + 10^{-7}) \approx 0.1269$$

This means that the FP-SRN $\mathcal{R}_{\theta+\vartheta}$ can experience a stationary gradient even though the loss is not 0. In this example, the classification is still correct if one sets a threshold of 0.5, indeed $0.88 > 0.5$ thus the classification would be correct. However, since Theorem 16 does not depend on the Encoder parameters, one could set all the Encoder's parameters to 0 while maintaining the theorem's validity. In this extreme case, the SRN outputs would be constantly 0.5, *i.e.* no classification and the gradient would be stationary.

## 5 A non learnable class

In the previous sections we have developed a series of arguments towards the fact that saturated FP-SRNs cannot be learned by gradient descent with a fixed learning rate. In order to exemplify the result we have exhibited a FP-SRN that simulates the DFA accepting all words containing an odd number of $a$'s on the alphabet $\{a, b\}$. In this section, we argue that the example of Section 4.3 is not an isolated case. We present an algorithm that, for any given DFA, outputs a saturated FP-SRN that simulates the DFA. We call this algorithm DFA2SRN, and it is an adaptation of the algorithm proposed by Minsky (1967) to FP-SRNs. Indeed, the author of (Minsky, 1967) works with RNNs based on a non differentiable activation function $H : \mathbb{R} \to \{0, 1\}$ with $H(x) = 1$ if $x \geq 0$ and 0 otherwise. This algorithm demonstrates that the non attainable region by gradient descent for FP-SRN contains a large class of interesting functions.

### 5.1 DFA2SRN

Let $\mathcal{D} = (\Sigma, Q, q_1, \delta, F)$ be a DFA. We suppose that the alphabet $\Sigma = \{\mathbf{e}_1, \ldots, \mathbf{e}_u\}$ is a set of one-hot vectors. We recall that $Q = \{q_1, \ldots, q_{|Q|}\}$ is the set of states, $q_1$ is the initial state, $\delta : \Sigma \times Q \to Q$ is the transition function, and $F \subset Q$ is the set of final states. In order to simulate the DFA by a FP-SRN we have to: 1) simulate the transition function, 2) encode every state such that it can be read by the FP-SRN, 3) simulate

the function discriminating between elements of $F$ and $Q \setminus F$. The encoding of the states of the DFA follows the spirit of one-hot encoding. The transition function is simulated by the encoder and the discrimination function will be simulated by the decoder.

The main idea is to make the hidden states $h_t$, $t \geq 1$, be an encoding of the state $q$ reached in the DFA while parsing a sequence up to its $t^{th}$ element. Thus, $\mathbf{M}_h$ has to encode the transition function, and $\mathbf{M}_o$ has to encode the $q \in F$ relation. In the rest of this section we will refer to the FP-SRN of Definition 5 and we will use formalism $\mathbf{W}$, the transition kernel $\mathbf{U}$, the input kernel and $\mathbf{v}$, the output kernel (the output kernel is a vector because the output that we need is a scalar in $\{0, 1\}$).

We set $|Q| := Card(Q)$, $|\Sigma| := Card(\Sigma)$. We assume that $Q = \{q_1, q_2, \ldots, q_{|Q|}\}$ and we define $I_F \subset \{1, \ldots, |Q||\Sigma|\}$ with:

$$(\{i, i+1, \ldots, (i+|\Sigma|) - 1\} \subset I_F \iff q_i \in F).$$

The linear parts of the SRN are defined by:

$$\mathbf{W} \in \mathbb{G}^{|Q||\Sigma| \times |Q||\Sigma|} \ ; \ \mathbf{U} \in \mathbb{G}^{|Q||\Sigma| \times |\Sigma|}$$

$$\mathbf{h}_0 \in \mathbb{G}^{|Q||\Sigma|} \text{ with } \mathbf{h}_0[1] = 1, \mathbf{h}_0[j] = 0 \text{ if } j \neq 1;$$

$$\mathbf{v} \in \mathbb{G}^{1 \times |Q||\Sigma|} \text{ with } \mathbf{v}[j] = J \text{ if } j \in I_F \text{ and } -J \text{ if } j \notin I_F.$$

$J$ is the smallest float number saturating the activation function as defined in Lemma 3.

We define $\mathbf{U} := 2J \cdot \begin{bmatrix} \mathbf{I}_{|\Sigma|} \\ \vdots \\ \mathbf{I}_{|\Sigma|} \end{bmatrix} \Big\} |Q|$ identity matrices $\mathbf{I}_{|\Sigma|}$ piled up.

For the construction of $\mathbf{W}$, we start by defining for every state $q_k$, $1 \leq k \leq |Q|$, a column vector $\mathbf{w}^k \in \mathbb{G}^{|Q||\Sigma|}$ by the following rules:

R1: For $1 \leq s \leq |\Sigma|$ if $\delta(q_k, a_s) = q_j$ then $\mathbf{w}^k[r] = -1$   with $r = (j-1)|\Sigma| + s$,

R2: All the other coordinates of $\mathbf{w}^k$ are set to $-3$.

Note that by definition of a DFA every vector $\mathbf{w}^k$ has exactly $|\Sigma|$ coordinates set to $-1$ and $(|Q|-1)|\Sigma|$ coordinates set to $-3$. The rule R1 defines an encoding for $(q_k, a_s, \delta(q_k, a_s))$. Therefore the vector $\mathbf{w}^k$ contains all the transitions starting at $q_k$.

Afterward we generate, for every $q_k \in Q$, the matrix:

$$\mathbf{W} := J \cdot \begin{bmatrix} \mathbf{W}^1 & \mathbf{W}^2 & \cdots & \mathbf{W}^{|Q|} \end{bmatrix} \in \mathbb{G}^{|Q||\Sigma| \times |Q||\Sigma|}$$

$$\text{where for } 1 \leq k \leq |Q|$$

$$\mathbf{W}^k := \underbrace{\begin{bmatrix} \mathbf{w}^k & \mathbf{w}^k & \cdots & \mathbf{w}^k \end{bmatrix}}_{|\Sigma| \text{ times}} \in \mathbb{G}^{|Q||\Sigma| \times |\Sigma|}. \tag{3}$$

In the definition of $\mathbf{W}^k$ the vector $\mathbf{w}^k$ is repeated $|\Sigma|$ times.

In Appendix C we give a formal proof that this construction simulates the given DFA on one-hot encoded symbols. The pseudo-code of the DFA2SRN algorithm is given in Algorithm 1. In this pseudo-code we used the numpy notation convention for matrix coordinates, meaning that $\mathbf{U}[k, s]$ represents the coordinate of matrix $\mathbf{U}$ at row $k$ and column $s$, and $\mathbf{U}[k, s:t]$ represents a set of coordinates that are located at row $k$ and ranging from $s$ to $t$.

## 6 Experimental validation

This section is dedicated to the experiments we carried in order to observe the theory on a practical case. This section can be broken down into two parts: the first is a series of experiments in which we attempt to train

---

**Algorithm 1** DFA2SRN

---

**Require:** $\mathcal{A} = (\Sigma, Q, q_1, \delta, F)$ a DFA, with $Q = \{q_0, \ldots, q_{|Q|-1}\}$ and $\Sigma = \{\omega_1, \ldots, \omega_{|\Sigma|}\}$ ; $J$ the saturating integer
   $\mathbf{U} \leftarrow |\Sigma| \cdot |Q|$ by $|\Sigma|$ matrix of zeros.
   $\mathbf{W} \leftarrow |\Sigma| \cdot |Q|$ by $|\Sigma| \cdot |Q|$ matrix of $-3$'s
   $\mathbf{V} \leftarrow |\Sigma| \cdot |Q|$ by 1 matrix of $-1$'s.
   **for** $j = 1 \ldots |Q|$ **do**
      **if** $q_j \in F$ **then**
         $A \leftarrow j \cdot |\Sigma|$
         $\mathbf{V}[A : A + |\Sigma|] \leftarrow 1$
      **end if**
      **for** $k = 1 \ldots |\Sigma|$ **do**
         $B \leftarrow t \cdot |\Sigma|$ where $t \in \{0, \ldots |Q|-1\}$ defined by $q_t = \delta(\omega_k, q_j)$
         $\mathbf{U}[k + A, k] \leftarrow 2$
         $\mathbf{W}[k + B, A : A + |\Sigma|] \leftarrow 1$
      **end for**
   **end for**
   $\mathbf{U} \leftarrow J \cdot \mathbf{U}$
   $\mathbf{W} \leftarrow J \cdot \mathbf{W}$
   $\mathbf{V} \leftarrow J \cdot \mathbf{V}$

---

FP-SRNs in the classical way, in the sense that we provide training data and apply the SGD algorithm. The second part is a more synthetic experiment. The reason for this dichotomy is that the classical experiments did not allow us to draw any conclusions concerning Theorem 16, so we had to imagine a different setting.

### 6.1 Learning regular languages

As we stated above, in this section we seek to learn regular languages in the classical way of deep learning practices. Our aim is to gather statistics about gradients in the hope of observing the stationary gradient phenomenon. Learning a regular language is learning a binary classification task on strings.

First we tried to learn the DFA of the example given in Section 4.3, whose language corresponds to the string of characters on the alphabet $\{a, b\}$ that have an odd number of $a$'s. To do this, we randomly generated 458 labeled strings with $50 \pm 2\%$ of positive labels. What is meant by positive labeling are the character strings that are part of the language and are labeled by 1, the other character strings are labeled by 0. The training sample is made up of character strings of lengths ranging from 2 to 10. The loss function used is the one we dealt with in the theorem (see Definition 1). The DFA2SRN algorithm was used as a guide for choosing the hidden dimension of the FP-SRN, so we randomly initialized an FP-SRN with hidden dimension $d = 4$, trained by stochastic gradient descent on 20,000 mini-batches of 32 strings. Indeed, we slightly deviated from the usual SGD, where data is split into mini-batches that will not change until the end of the training. In our setup, we randomly draw a mini-batch at each iteration $i = 1, 2, \ldots, 20000$. The statistics we tracked during training are: the value of the *Loss*, the Euclidean distance between *two consecutive parameter vectors*, the infinite **norm of the gradient** and the *distance to the target parameters* (i.e. the parameters given by the DFA2SRN algorithm) The results of this experiment are presented in Figure 6

This experiment does not allow us to validate or invalidate Theorem 16. The FP-SRN did not learn the behavior of the DFA. By observing the *distance graph between two consecutive parameter vectors*, we do not observe any convergence behavior. However, despite the apparent simplicity of the DFA that we tried to learn (two states and an alphabet composed of two symbols), this DFA belongs to the most complex subclass of regular languages. According to van der Poel et al. (2024) the subclass in question is called $\mathbb{Z}_p$. Moreover, the training set is quite small (only 458 sequences). For this reason, we conducted the same experiment on languages from lower complexity classes in order to ensure that there is no bias in the experience of SGD learning from Figure 6. In addition to the classification of sub-regular languages, van der Poel et al. (2024) provides a benchmark of 1,800 languages covering all levels of complexity. To carry out this experiment, we selected 9 languages that closely match the benchmark's complexity spectrum. The training on these 9

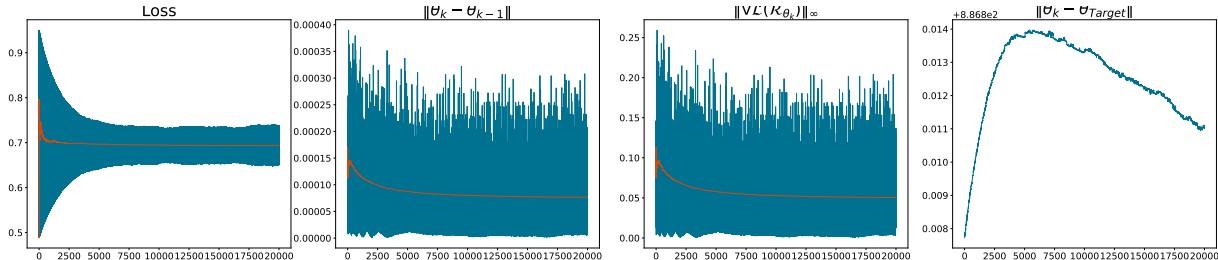

Figure 6: Regular SGD training where the four statistics of the learning experience of the odd $a$'s language. The curves in blue correspond to the statistics recorded during training, and the curves in orange are the rolling averages.

languages led us to the same conclusion than the first experiment. The details of this experiment can be found in Appendix E.

As we have already stated, these experiments do not validate Theorem 16, but neither do they invalidate it. We did not observe the stationary gradient phenomenon in these experiments, but the parameters of the FP-SRN remained far from the zone where saturation could be observed. All that can be concluded is that the task and/or the context of the learning is too hard for SGD learning. To better fit the context of our theorem, we designed another type of experiments.

## 6.2 The Synthetic experiment

The failure of the previous experiment may come from the fact that the direction of the gradient fluctuates constantly, which does not allow the parameters to converge. Some cases have been reported where this fluctuation is not too important because on average the direction of the gradient is constant or almost constant. In our case, however, it would appear that the fluctuations are too important. We have therefore built a setup in which the direction of the gradient is constant and points towards the target parameters. We recall that the target parameters are those given by the DFA2SRN algorithm. This is fairly simple to set up, as all we need to do is to define a straight line between the parameters randomly initialized and the target parameters. By moving along this segment, we obtain a vector of parameters $\theta_i$ which we can use to evaluate the gradient of the network $\mathcal{R}_{\theta_i}$. This is exactly what we did with two variants. In the first one, all the parameters were evolving while, in the second one, we froze the decoder parameters after a single iteration. The aim of the second variant was to observe the impact of the decoder on the gradient.

We used the data from the first experiment to evaluate the gradient, i.e., we used the data set of 458 words on the alphabet $\{a, b\}$ labeled to correspond to the language of strings with an odd number of $a$'s. We initialized $\theta_0$ parameters of FP-SRN at the origin, according to common practices in Deep Learning. Using these $\theta_0$ parameters and the $\theta_T$ goal parameters, we constructed the segment $[\theta_0, \theta_T]$ as well as its subdivision $\{\theta_0, \theta_1, \dots, \theta_T\}$ containing 10 points on the segment distributed equidistantly. For $i = 0, 1, \dots, T$ we evaluated the gradient of $\mathcal{R}_{\theta_i}$ over the training set, with a stopping criterion $\|\nabla(\mathcal{L}_\varepsilon(\mathcal{R}_{\theta_i}))\|_\infty < 10^{-14}$. If the stopping criterion is satisfied for $i < T$ we end up with a rough approximation of where the norm of the gradient has dropped below $10^{-14}$. So, in order to find a better approximation of where the norm of the gradient falls below this threshold, we repeat the same procedure on the segment $[\theta_{i-1}, \theta_i]$. Then, once the stopping criterion is satisfied again, we start a further refinement.

The statistics in Figure 7, are the value of the Loss, the distance to the target parameters, the infinite norm of the gradient and the proportion of the parameters affected by the stationary gradient. All this statistics (beside the pie chart) are computed for every single word in the training set. One can observe some variations in the thickness in the plots, this is due to oscillations in the statistics caused by the computation of the gradient for every word individually and that all the data are disposed on relatively small graphs. Note that the horizontal axis reflects the number of data points collected for the statistics, in the first variant over 60,000 data points for each statistic were collected, and in the second variant just above 30,000.

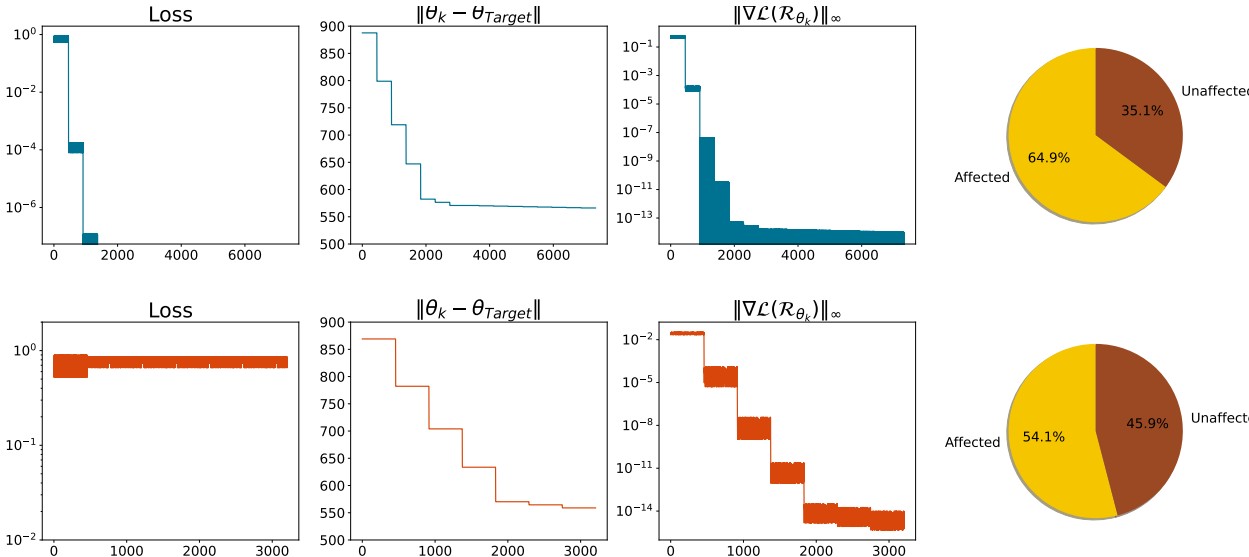

Figure 7: Two variants of the synthetic experiment, the first row represents the experiment where all the parameters are updated. The second row is the variant where the decoder's parameters are frozen after one update.

In the first version of this experiment we observe the stationary phenomenon occurring at distance to the target parameters over 500. In this configuration, the loss converges to zero even if the FP-SRN parameters have not reached the target parameters. Given the experimental setting, this cannot come from over-fitting as the impact of the data on the change of parameters is small. It can be the case that the learned FP-SRN corresponds to a finite part of the language, as already observed by Weiss et al. (2018). But it can also be the case that the learned FP-SRN corresponds to the correct language since there may be a several combinations of parameters that can be used to simulate the operation of the same DFA. Though of general interest, this FP-SRN is not saturated, which puts it out of the scope of this work: the fact that the loss converges does not invalidate our assertion about the impossibility to learn saturated FP-SRN.

In the second variant, we have frozen the decoder's parameters after just one update. In this case, we observe that the loss does not converge to zero, and this can be explained by the fact that the decoder does not dispose of suitable parameters for the required task. Nevertheless, a frozen decoder does not affect the encoders gradient, just as predicted by Theorem 16. Moreover, it seems that the stationary gradient is reached in less iterations than in the first variant, a fact that we do not explain, but which is an interesting question by its own. Finally, the proportion of affected parameters by the stationary gradient is smaller in the second variant because, despite having no effect on the encoder's gradient, the frozen decoder's parameters did not grow and thus can be affected by a gradient with coordinates smaller than $10^{-14}$.

Figure 7 reveals that the synthetic experiment validates Theorem 16 in the sense that by heading towards parameters that will saturate the FP-SRN the gradient will experience the stationary phenomenon. Moreover, this experiment shows that learning saturated FP-SRN by gradient descent is even harder than what is predicted by Theorem 16, since for this particular example the theorem predicts a stationary behavior at distance 6.16 to the target parameters, and we observe here that all the gradients parameters are below the rounding error threshold (i.e. below $10^{-14}$) at distance over 500. We illustrate this fact in Figure 8. This discrepancy with what predicts Theorem 16 can be explained by the use of loosed upper bounds in Theorem 14 (a triangle inequality involving a lot of elements) and the consideration of the worst case scenario with the Lipschitz constant $\rho_L$ of the Loss function $\mathcal{L}_\varepsilon$. We discuss this further in the Discussion (Section 7).

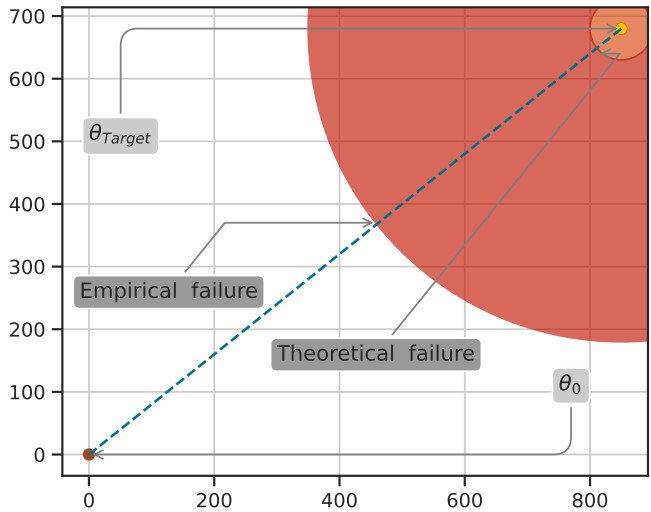

Figure 8: Representation of the synthetic experiment in 2D. The initial parameters $\theta_0$ are randomly chosen around the origin, and the $\theta_{Target}$ parameters represent the target parameters provided by the DFA2SRN algorithm. During the experiment, the parameters are selected on the blue line and the gradient of these parameters are evaluated. The orange disk represents the Euclidean ball centered at $\theta_{Target}$ whose radius is predicted by Theorem 16, and the red disk represents the zone from which the gradient is empirically stationary.

## 7 Discussion

In this work we studied theoretically and experimentally the behavior of SRNs subjected to finite precision gradient descent training. We have shown that it is not possible to obtain saturated SRNs by gradient descent, in finite precision, with a fixed training step. We discuss in the next paragraph that this result is easily transferable to the case of evolutionary but bounded learning steps. Our approach consists in: 1) exhibiting a set of parameters $\mathbf{p}$ that will induce, in finite precision, saturation in FP-SRNs and a stationary gradient, 2) showing that this behavior persists in a sufficiently close neighborhood of $\mathbf{p}$. This result shows that finite precision induces the existence of areas of the parameter space that are not accessible by gradient descent. We then show that the parameters that saturate the SRN produce functions of interest because it is possible to simulate the operation of any DFA. This is the result of the 1 algorithm.

A first element of discussion is that our result stands only for the gradient descent algorithm with fixed learning rate $\alpha$. In fact, it is automatically extendable to the case of a variable learning rate $\{\alpha_l\}_{l=1}^D$ where $D > 1$, as long as $\{\alpha_l\}_{l=1}^D$ is bounded. By setting $\alpha := \max_{1 \leq j \leq D} \alpha_j$ the result remains true for any $\alpha_j$. A second element of discussion is that our result, though stated for a classical gradient descent, is applicable to the widely used stochastic gradient descent: as all gradients are null under the assumptions of Theorem 16, so is their mean on any batch. In this work we have considered the case of binary classification which can effortlessly be extended to multilogit classification. Indeed, a Decoder with output dimension $o > 1$ can be seen as a concatenation of $o$ different Sub-Decoders of dimension 1. Since our result stands independently from the Decoders parameters none of the Sub-Decoder's gradient will influence the Encoder's parameters. Another straightforward extension of our result concerns other activation functions. Indeed, a similar argument can be made concerning any bounded function, like the hyperbolic tangent: the only difference being the value that saturates the activation, and a different expression of the boundary $\zeta$. The same goes for the algorithm DFA2SRN, the modifications to realize are in matrices $\mathbf{W}$ and $\mathbf{V}$. The saturation of the tanh will produce vectors in $\{-1, 1\}^h$ so instead of having a one-hot encoded vector the decoder will produce vectors with a unique 1, and $-1$. This family $\mathcal{F}_{\{-1,1\}}$ of vectors forms a basis in $\mathbb{R}^h$, hence one

can define an invertible matrix $\Theta$ that maps $\mathcal{F}_{\{-1,1\}}$ to the canonical basis formed of one-hot encoded vectors. By defining $\tilde{\mathbf{W}} = \mathbf{W}\Theta$ and $\tilde{\mathbf{V}} = \mathbf{V}\Theta$ we will obtain a FP-SRN with tanh activation function and that will simulate the DFA. A potential line of research would be to reduce the gap between theory and empirical results. In fact, in Section 6 we found a significant discrepancy between the prediction of Theorem 16 regarding the appearance of the stationary gradient phenomenon and its effective appearance. This difference can be explained by the use of loose upper bounds in the proofs of Theorems 14 and 16. The first one is in the application of a triangle inequality in the proof of Theorem 14. A second non-optimal upper bounding is applied when considering the Lipschitz constant of the Loss function. This constant, denotes $\rho_L$, plays an important role in Theorem 16. It represents the worst possible case in the deviations between $\mathcal{R}_\theta$ and its neighbors $\mathcal{R}_{\tilde{\theta}}$. We think it is possible to remove the constant $\rho_L$ by leveraging the $\beta$-saturation assumption. Indeed, we have guaranties on the deviation between $\mathcal{R}_\theta$ and $\mathcal{R}_{\theta+\vartheta}$, which could allow us to avoid the worst-case scenario represented by the Lipschitz constant $\rho_L$.

One can wonder the interest of the class of SRNs that we exhibit as an example of a non-learnable class. Notice first that any RNN behaves like a finite state machine in a finite precision configuration (Pozarlik, 1998): its hidden state can only store a (huge) finite number of configurations. It is thus relevant to use the simplest finite state machines, the DFA, as a starting point. Then, even if its construction is not trivial, it covers a large class of functions thus assessing the interest of the theorem. Finally, the algorithm constructing an SRN from any DFA is of interest by itself. The algorithm of Minsky (1967) is designed for RNNs with non continuous activation functions, *i.e.* RNNs that are not used nowadays. The closest paper on the subject (Siegelmann, 1996) shows that for a given DFA with $|Q|$ states operating on an alphabet of size $|\Sigma|$ there exists an SRN with $3|Q||\Sigma|$ neurons that simulates the computation of the DFA. However, its proof is not constructive and thus does not provide an algorithm to build such model.

After discussing some direct extensions to our work, we would like to discuss the perspectives. The first direction is to relax the saturation hypothesis. Indeed, as it is formulated, the $\beta$-saturation is a strong hypothesis. Mitarchuk et al. (2024) propose a relaxation to the $\beta$-saturation that they call DS-$\beta$-saturation standing for Desynchronised Sliding $\beta$-saturation. The idea of this definition is that within a sliding window of $F$ elements, every Encoder neurons has to saturate at least once. The authors demonstrate that with this hypothesis a SRN can sill be robust to perturbations. Hence, we want to explore a potential extension our work in this direction. Another direction of work would be to extend this result that stands for sequence classification corresponding to many-to-one, to problems where we need to produce sequences many-to-many situations. Finally, since our work shows that gradient descent with a bounded learning rate is not well-suited to achieve saturation in finite precision, despite the proven interest of this notion for learning long term dependencies, an advice to practitioners could be to look either for new regularization terms forcing saturation in the loss or even for other learning algorithms than gradient descent. This research direction echoes current efforts in low-precision learning and the determination of optimal hyper-parameters (Blake et al., 2024).

**Related works**

Sum et al. (2019) study the gradient descent algorithm for feed-forward networks. They study the behavior of the algorithm when the network weights are noisy. They assume that the network weights and the arithmetic are based on finite precision, and two cases are distinguished: 1) fixed-point arithmetic, 2) floating-point arithmetic. They show that even slight noise in the case of floating-point arithmetic can deviate the learning algorithm. Karner et al. (2024) consider training a feed-forward network with ReLU activation under the assumption of finite precision. Since the type of network studied in this paper is a piece-wise affine function, they tackle the established fact that network depth causes an exponential increase in affine parts. They show that it is highly unlikely that the network obtained after training exhibits an exponential amount of affine parts if the network is trained in finite precision. Colbrook et al. (2022) consider the class of inverse problems, and show that for any finite precision set up with $M$ digits of precision there exists an infinity of inverse problems that cannot be learned by neural networks. Their work is of a greater scope, because they do not focus on a specific learning algorithm. Fono et al. (2022) address the problem of the limits of artificial neural network with an even higher standpoint. The authors go back to the limits of Turing machines as digital machines, thus positioning neural networks as a subset of digital machines unable to cope with non-computable signals.

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

## A    General results on $\sigma$ and proof of Theorem 16

### A.1    General results about $\sigma$

In this section we state and prove a series of results about the sigmoidal function $\sigma$.

**Lemma 17** *Let $\sigma : \mathbb{R} \ni x \mapsto \frac{1}{1+e^{-x}}$. For all $c \geq 1/4$ we have $\sigma'(x) \leq c$ and for $1/4 \geq c > 0$ such that $|x| \geq \ln\left(\frac{1+\sqrt{1-4c}}{1-\sqrt{1-4c}}\right)$ we have: $\sigma'(x) \leq c$*

**Proof** Of the lemma

At first we recall some properties of the functions $\sigma$ and $\sigma'$. We have:

$$\forall\, x \in \mathbb{R} \;\; \sigma(x) = 1 - \sigma(-x)$$

$\sigma'$ is symmetric:

$$\forall\, x \in \mathbb{R} \;\; \sigma'(x) = \sigma'(-x)$$

and

$$\sigma'(x) = \sigma(x)(1 - \sigma(x)).$$

Moreover the maximum of $\sigma'$ is attained on $x = 0$ and is equal to $1/4$, thus the first claim of the lemme is trivial. Now we assume that $0 < c \leq 1/4$ and we are going to solve the equation:

$$\sigma(x)(1 - \sigma(x)) = c.$$

This is a quadratic equation if we substitute $\sigma(x)$ by $y$ and write the equation as:

$$y - y^2 - c = 0.$$

By hypothesis $0 < c \leq 1/4$, thus the above equation admits two solutions:

$$y_1 = \frac{1 - \sqrt{1-4c}}{2} \;\;, y_2 = \frac{1 + \sqrt{1-4c}}{2}.$$

The sign of the dominant factor is negative thus all the real numbers $y$ that satisfy:

$$y - y^2 \leq c,$$

are outside the open interval $]\frac{1-\sqrt{1-4c}}{2}, \frac{1+\sqrt{1-4c}}{2}[$. In our case we have a supplementary restriction on $y$, that is $y \in [0, 1]$, since $y = \sigma(x)$. In order to obtain a bound for $x$, it is sufficient to apply $\sigma^{-1}$ on $\frac{1+\sqrt{1-4c}}{2}$. Hence:

$$\sigma^{-1}\left(\frac{1+\sqrt{1-4c}}{2}\right) = \ln\left(\frac{\frac{1+\sqrt{1-4c}}{2}}{1 - \frac{1+\sqrt{1-4c}}{2}}\right)$$

$$= \ln\left(\frac{1+\sqrt{1-4c}}{1-\sqrt{1-4c}}\right).$$

Thus if $x > \ln\left(\frac{1+\sqrt{1-4c}}{1-\sqrt{1-4c}}\right)$ then $\sigma'(x) < c$. The symmetry of $\sigma'$ extends the result: for all $x$ such that $|x| > \ln\left(\frac{1+\sqrt{1-4c}}{1-\sqrt{1-4c}}\right)$ we have $\sigma'(x) < c$. ∎

The following lemma is a more specific version of the previous lemma.

**Lemma 18** *Let $z > 1, a \geq 4$ and $0 < \varepsilon \leq 1$ be tree real numbers. For all $|x| \geq \ln\left(\frac{1+\sqrt{1-\frac{4\varepsilon}{za(1+\varepsilon)}}}{1-\sqrt{1-\frac{4\varepsilon}{za(1+\varepsilon)}}}\right)$ we have:*

$$za\sigma'(x) \leq 1 \quad and \quad \frac{za\sigma'(x)}{1 - za\sigma'(x)} \leq \varepsilon$$

**Proof** Let $z > 1, a \geq 4$ and $0 < \varepsilon \leq 1$ be three real numbers and $x \geq \ln\left(\frac{1+\sqrt{1-\frac{4\varepsilon}{za(1+\varepsilon)}}}{1-\sqrt{1-\frac{4\varepsilon}{za(1+\varepsilon)}}}\right)$. By Lemma 17, just above, we have that:

$$\sigma'(x) = \frac{\varepsilon}{za(1+\varepsilon)}$$

because $0 < \varepsilon \leq 1$ and $za \geq 4$. And we obtain that:

$$za\sigma'(x) = \frac{\varepsilon}{(1+\varepsilon)} < 1 .$$

We set $c := a\sigma'(x)$ and from the previous equality one can deduce:

$$zc = \frac{\varepsilon}{(1+\varepsilon)} \iff (1+\varepsilon)zc = \varepsilon$$
$$\iff zc + \varepsilon zc = \varepsilon$$
$$\iff zc = \varepsilon(1 - zc)$$
$$\iff \frac{zc}{1-zc} = \varepsilon$$

∎

Lemma 19 is a result about the convergence rate and the approximation of $\sigma'$

**Lemma 19** *Let $x, a \in \mathbb{R}_+$ two positive real numbers such that $a \leq x - 1.5$, and let $\sigma : \mathbb{R} \ni x \mapsto \frac{1}{1+e^{-1}}$, the sigmoidal function. Then:*

$$\sigma'(x - a) \leq \sigma'(x) + ae^{-(x-a)}$$

**Proof** Let $x, a \in \mathbb{R}_+$ two positive real numbers such that $a \leq x - 1.5$, and let $\sigma : \mathbb{R} \ni x \mapsto \frac{1}{1+e^{-x}}$. We recall that the sigmoidal function is at least twice differentiable, with for all $t \in \mathbb{R}$

$$\sigma'(t) = \sigma(t)(1 - \sigma(t))$$
$$\sigma''(t) = \sigma'(t)(1 - 2\sigma(t)).$$

One can observe that $\sigma''(t) \leq 0$ for $t \geq 0$, because $\sigma' > 0$ and $(1 - 2\sigma(t)) \leq 0$ on positive real numbers $t$. Moreover, it is easy to see that $(1 - 2\sigma(t)) \in [-1, 1]$ for all real numbers $t$. Hence $0 \leq -(1 - 2\sigma(t)) \leq 1$. On the other side, from the definition of $\sigma(t)$ we have another expression of its derivative :

$$\sigma'(t) = \frac{e^{-t}}{(1 + e^{-t})^2} \leq e^{-t}.$$

From the mean value theorem we obtain that there exists $c \in ]x - a, x[$ such that:

$$\sigma'(x - a) = \sigma'(x) - a \cdot \sigma''(c).$$

The function $\sigma''$ reaches its global minimum in the interval $]1, 1.5[$, and by assumption we have $x - a \geq 1.5$, Therefore by summoning the fact that $\sigma''$ is a strictly increasing function on the interval $]1.5, +\infty[$ (*i.e.* $-\sigma''$ is a strictly decreasing function on $]1.5, +\infty[$) we can deduce that:

$$\sigma'(x - a) \leq \sigma'(x) - a \cdot \sigma''(x - a)$$
$$= \sigma'(x) - a \cdot \sigma'(x - a)\left(1 - 2\sigma(x - a)\right)$$
$$\leq \sigma'(x) + a \cdot \sigma'(x - a)$$
$$\leq \sigma'(x) + a \cdot e^{-(x-a)}$$

∎

We finish this subsection with a result about the convergence of a geometric series involving $\sigma'$. This lemma is very handy for the proof of Theorem 16 and Theorem 14.

**Lemma 20** *Let $a > 1$, $z = \frac{1}{a}$, $\zeta = \ln\left(\frac{1+\sqrt{1-4z}}{1-\sqrt{1-4z}}\right)$ and $\sigma : \mathbb{R} \ni x \mapsto \frac{1}{1+e^{-x}} \in [0,1]$ the sigmoidal function. Then for all $t > 0$ we have:*

$$\sigma'(\zeta + t)a < 1.$$

**Proof** Let $a > 1$, $z = \frac{1}{a}$, $\zeta = \ln\left(\frac{1+\sqrt{1-4z}}{1-\sqrt{1-4z}}\right)$ and $\sigma : \mathbb{R} \ni x \mapsto \frac{1}{1+e^{-x}} \in [0,1]$. By Lemma 17 we know that $\sigma'(\zeta) = z$, thus $\sigma'(\zeta)a = 1$. To conclude, we know that $\sigma'$ is a strictly decreasing function on the interval $[0, +\infty[$ Therefore, for any $t > 0$ we can assert that:

$$\sigma'(\zeta + t)a < 1.$$

∎

### A.2 Proof of Theorem 16

**Definition 21 (Recall of section 3)** *Let $\mathcal{R}_\theta$ be a SRN with $\mathbf{M}_h = [\mathbf{U}\ \mathbf{W}\ \mathbf{b}]$ $\mathbf{M}_o = [\mathbf{V}\ \mathbf{c}]$ and $T \geq 1$ an integer. For a parameter $m_{i,j}$ in the matrix $\mathbf{M}_h$ we define a path $\gamma = (\gamma_1, \ldots, \gamma_k)$ of length $k$ for $1 \leq k \leq T$ as a finite sequence of $(\mathbf{M}_h, \mathbf{M}_o)$ parameters, such that for $k = 1$ we have $\gamma_1 = \mathbf{M}_o[i]$, for $k = 2$ we have $\gamma_1 = \mathbf{M}_o[v]$ and $\gamma_2 = m_{v,i}$ for some $1 \leq v \leq h$. For the case $k \geq 3$ we define $\gamma$ such that $\gamma_k = m_{v,i}$ for some $1 \leq v \leq h$, for all $1 < s < k$ $\gamma_{s-1} = m_{l,t}$, $\gamma_s = m_{t,r}$ and $\gamma_{s+1} = m_{r,w}$. Finally, we define the set $\Gamma_{i,j}$ containing all such paths.*

This section is dedicated to the proof of the criterion for BPTT failure. To do this we are going to prove that, under the hypotheses of Theorem 16, the update generated by GD is rounded to zero for every parameter in the SRN. We position ourselves in the context of binary classification on sequences, $S = \{(\omega_1, y_1), \ldots, (\omega_N, y_N)\}$ is the labeled training dataset where $\omega_i = \{\mathbf{x}_k\}_{k=1}^T \subset \mathbb{G}^u$ *i.e.* the data is vector embedded in dimension $u \geq 1$. $\mathcal{R}_\theta$ is a FP-SRN with in hidden dimension $h \geq 1$, input dimension $u$ and output dimension $o = 1$. We fix a parameter $m_{i,j}$ in the encoder weight matrix $\mathbf{M}_h$, and compute recursively the gradient on the word $\omega$ with $|\omega| = T$. We already established in Equation 2 that:

$$\frac{\partial \mathcal{L}(\mathcal{R}^\omega, y)}{\partial m_{i,j}} = \frac{\partial \mathcal{L}(\mathcal{R}^\omega, y)}{\partial x} \cdot \sigma'_{\mathbf{M}_o}(\mathbf{h}_T) \sum_{\gamma \in \Gamma_{i,j}} \prod_{s=1}^{k} \gamma_s \cdot \sigma'_{\gamma_s}(\mathbf{x}_{T-s} \oplus \mathbf{h}_{T-s-1} \oplus 1) \tag{4}$$

By the BPTT technique $\frac{\partial \mathcal{L}(\mathcal{R}^\omega, y)}{\partial x} \cdot \sigma'_{\mathbf{M}_o}(\mathbf{h}_T) \sum_{\gamma \in \Gamma_{i,j}} \prod_{s=1}^{k} \gamma_s \cdot \sigma'_{\gamma_s}(\mathbf{x}_{T-s} \oplus \mathbf{h}_{T-s-1} \oplus 1)$ contains the update for a parameter $m_{i,j}$ in every layer of the unrolled neural network $\mathcal{R}^\omega$. The following lemma deals with the number of paths in $\Gamma_{i,j}$ depending on their length.

**Lemma 22** *Let $\mathcal{R}$ be a FP-SRN and $\omega$ a word of length $T \geq 1$. Let $m_{i,j}$ be a parameter in $\mathbf{M}_h$ and $\Gamma_{i,j}$ the set of all paths as in Definition 21. Then we have:*

$$\Gamma_{i,j} = \bigcup_{\varpi=1}^{T} \{\gamma \in \Gamma : |\gamma| = \varpi\}$$

$$Card(\Gamma_{i,j}) = \sum_{\varpi=1}^{T} Card(\{\gamma \in \Gamma : |\gamma| = \varpi\})$$

$$= \sum_{\varpi=1}^{T} h^\varpi$$

**Proof** Let $\mathcal{R}$ be a FP-SRN and $\omega$ a word of length $T \geq 1$. Let $m_{i,j}$ be a parameter in $\mathbf{M}_h$ and $\Gamma_{i,j}$ the set of all paths as in Definition 21. The identity

$$\Gamma_{i,j} = \bigcup_{\varpi=1}^{T} \{\gamma \in \Gamma : |\gamma| = \varpi\}$$

is trivially true. It is clear the for $\varpi \neq \tilde{\varpi}$, we have

$$\{\gamma \in \Gamma \,:\, |\gamma| = \varpi\} \cap \{\gamma \in \Gamma \,:\, |\gamma| = \tilde{\varpi}\} = \emptyset$$

because a path cannot have two distinct lengths. This proves that:

$$Card(\Gamma_{i,j}) = \sum_{\varpi=1}^{T} Card(\{\gamma \in \Gamma \,:\, |\gamma| = \varpi\}).$$

Now let $\gamma \in \Gamma_{i,j}$ be a path of length $\varpi$, we know that $\gamma_\varpi = m_{t,i}$ for some $1 \leq t \leq h$ meaning that we have $h$ possible choices for the index $t$. Once we have fixed $t$, we know that $\gamma_{\varpi-1} = m_{r,t}$ for some $1 \leq r \leq h$. Once again we have $h$ possible choices for the variable $r$. This argument is true for all $\gamma_s$ in $\gamma$, hence we have $h^\varpi$ possible choices for $\gamma$ of length $\varpi$. ∎

Now we proceed to prove the claim of Theorem 16, but first, for convenience, we recall the statement:

**Theorem 16** Let $\mathbb{G}$ be the set of all float numbers representable in a given finite precision configuration $(B, M, X)$. Let $S = \{(\omega_1, y_1), \dots (\omega_N, y_N)\}$ be a labeled training set with $N$ samples and where $\omega_l = \{\mathbf{x}_k\}_{k=1}^T \subset \mathbb{G}^u$ is a finite sequence of vectors such that all vector $\mathbf{x}_k$ respect $\|\mathbf{x}_k\|$ for $1 \leq l \leq N$ and $T \geq 1$ an integer representing the length of the word $\omega_l$. We set $\mathcal{L} : \mathbb{G} \times \mathbb{G} \to \mathbb{G}$ to be a differentiable loss function. Let $\alpha > 0$ the learning rate as used in Definition 10. Let $\mathcal{R}_\theta$ be an FP-SRN with hidden dimension $h$, input dimension $u$ and output dimension $o = 1$ *i.e.* $\mathcal{R}_\theta$ is shaped for binary classification on strings. We recall that the row vectors of the matrix $\mathbf{M}_h$ are denoted $\mathbf{m}_i$ for $1 \leq i \leq h$, and that since the output dimension $o = 1$ it means that the matrix $\mathbf{M}_o$ is of dimension $(1, h)$. We set

$$m^* := \max \{|m| \,:\, m \in \theta\}$$
$$\underline{m} := \min \{|m| \,:\, m \in \theta \text{ and } |m| > 0\},$$

*i.e.* $m^*$ is the largest parameter in absolute value and $\underline{m}$ is the smallest, in absolute value, non zero parameter. We set $\varepsilon = B^D$ with $D = \lfloor \log_B(\underline{m}) \rfloor - (M + 1)$. We suppose that:

1. For all $y \in \{0, 1\}$ the function $[0, 1] \ni x \mapsto \left| \frac{\partial}{\partial x} \mathcal{L}(x, y) \right|$ is bounded by a constant $\rho_L$.

2. $\exists\ \eta > 0$ such that $\mathcal{R}_\theta$ is $(\zeta + \eta)$-saturated where

$$\zeta = \ln \left( \frac{1 + \sqrt{1 - 4z}}{1 - \sqrt{1 - 4z}} \right) \,, z = \frac{4}{\psi h \rho_L} \frac{\varepsilon}{1 + \varepsilon} \,, \psi := \alpha \cdot \max\{(m^* + \eta/2), \|\mathbf{W}\|\}$$

Then with $\Delta := \sigma'(\zeta + \frac{\eta}{2})$ and $\mathbf{W}$ the transition kernel extracted from $\mathbf{M}_h$, for all $\vartheta \in \mathbb{R}^{dim(\theta)}$ such that $\|\vartheta_{Encoder}\| \leq \frac{\eta}{2} \cdot \frac{(1 - \Delta\|\mathbf{W}\|)}{(2 + \sqrt{h})}$ the FP-SRN $\mathcal{R}_{\theta+\vartheta}$ will experience a stationary gradient for all non zero parameters on the training data set $S$.

**Proof** [Theorem 16] We assume that $\mathcal{R}_\theta$ is a FP-SRN satisfying the assumptions 1 and 2 of Theorem 16. By Theorem 14 we know that for $\vartheta \in \mathbb{R}^{dim(\theta)}$ such that $\|\vartheta\| \leq \frac{\eta}{2} \frac{(1 - \Delta\|\mathbf{W}\|)}{(2 + \sqrt{h})}$, the FP-SRN $\mathcal{R}_{\theta+\vartheta}$ is $(\zeta + \eta/2)$-saturated. Indeed we are applying Theorem 14 in the case where $t = 1/2$. We select a vector $\vartheta \in \mathbb{R}^{dim(\theta)}$ such that $\|\vartheta\| \leq \frac{\eta}{2} \frac{(1 - \Delta\|\mathbf{W}\|)}{(2 + \sqrt{h})}$ and we will prove that $\mathcal{R}_{\theta+\vartheta}$ will experience a Stationary Gradient on all one-hot-encoded sequences. We recall that for a sequence $(\omega, y) \in S$, $|\omega| = T$, the gradient back propagated in order to update the parameter $\mathring{m}_{i,j}$ can be expressed as follows:

$$\frac{\partial \mathcal{L}(\mathcal{R}_{\theta+\vartheta}(\omega), y)}{\partial \mathring{m}_{i,j}} = \frac{\partial \mathcal{L}(\mathcal{R}_{\theta+\vartheta}(\omega), y)}{\partial x} \cdot \sigma'_{\mathring{\mathbf{M}}_o}(\mathring{\mathbf{h}}_T) \sum_{\gamma \in \Gamma_{i,j}} \prod_{s=1}^{|\gamma|} \gamma_s \cdot \sigma'_{\gamma_s}(\omega_{T-s} \oplus \mathring{\mathbf{h}}_{T-s-1} \oplus 1).$$

We set $F := \frac{\partial \mathcal{L}(\mathcal{R}_{\theta+\vartheta}(\omega),y)}{\partial x} \cdot \sigma'_{\mathring{\mathbf{M}}_o}(\mathring{\mathbf{h}}_T)$. We will discus the variable $F$ later. As stated above the FP-SRN $\mathcal{R}_{\theta=\vartheta}$ is $(\zeta + \eta/2)$-saturated. Therefore, meaning that for all $1 \leq s \leq h$ and for all $1 \leq k \leq T$ we have:

$$\left| \langle \mathring{\mathbf{m}}_s, \omega_k \oplus \mathring{\mathbf{h}}_{k-1} \rangle \right| \geq \zeta + \eta/2$$

implying by the properties of $\sigma'$ that:

$$\sigma'(\langle \mathring{\mathbf{m}}_s, \omega_k \oplus \mathring{\mathbf{h}}_{k-1} \rangle) \leq \sigma'(\zeta + \eta/2).$$

Moreover, by hypothesis on $\vartheta$, for all parameters $\mathring{m}_{r,t}$ in the matrix $\mathring{\mathbf{M}}_h$ we have

$$|\mathring{m}_{r,t}| \leq m^* + \frac{\eta}{2} \frac{(1 - \Delta \|\mathbf{W}\|)}{(2 + \sqrt{h})} \leq m^* + \frac{\eta}{2}.$$

Let $\gamma \in \Gamma_{i,j}$ be a path, from the $(\zeta + \eta/2)$-saturation of $\mathcal{R}_{\theta+\vartheta}$ we can deduce that:

$$\prod_{s=1}^{|\gamma|} \gamma_s \cdot \sigma'_{\gamma_s}(\mathbf{e}_{T-s} \oplus \mathring{\mathbf{h}}_{T-s-1} \oplus 1) \leq \prod_{s=1}^{|\gamma|}(m^* + \eta/2) \cdot \sigma'(\zeta + \eta/2)$$

$$= \left((m^* + \eta/2) \cdot \sigma'(\zeta + \eta/2)\right)^{|\gamma|}.$$

This allows us to derive a bound on $\frac{\partial \mathcal{L}(\mathcal{R}_{\theta+\vartheta}(\omega),y)}{\partial \mathring{m}_{i,j}}$:

$$\sum_{\gamma \in \Gamma_{i,j}} \prod_{s=1}^{|\gamma|} \gamma_s \cdot \sigma'_{\gamma_s}(\mathbf{e}_{T-s} \oplus \mathring{\mathbf{h}}_{T-s-1} \oplus 1) \leq \sum_{\gamma \in \Gamma_{i,j}} \left((m^* + \eta/2) \cdot \sigma'(\zeta + \eta/2)\right)^{|\gamma|}$$

By Lemma 22 we know the exact number of paths in $\Gamma_{i,j}$. We set $C := \left((m^* + \eta/2) \cdot \sigma'(\zeta)\right)$. The function $\sigma'$ is strictly decreasing on $[0, \infty[$ Therefore one can derive the following bound:

$$\sum_{\gamma \in \Gamma_{i,j}} \left((m^* + \eta/2) \cdot \sigma'(\zeta + \eta/2)\right)^{|\gamma|} < \sum_{\gamma \in \Gamma_{i,j}} \left((m^* + \eta/2) \cdot \sigma'(\zeta)\right)^{|\gamma|}$$

$$= \sum_{\varpi=1}^{T} \sum_{\gamma \in \Gamma_{i,j}, |\gamma|=\varpi} C^{\varpi}$$

$$= \sum_{\varpi=1}^{T} h^{\varpi} C^{\varpi}$$

$$= hC \left(\frac{1 - h^T C^T}{1 - hC}\right)$$

We know that $C \leq \max\{(m^* + \eta/2), \|\mathbf{W}\|\} \cdot \sigma'(\zeta)$ Therefore in virtue of Assumption **2** and with the application of Lemma 17 we can assert that:

$$hC < 1 \quad \text{and} \quad \frac{hC}{1 - hC} = \frac{\varepsilon}{\alpha \rho_L \rho_\sigma}.$$

This leads us to the fact that:

$$hC \left(\frac{1 - h^T C^T}{1 - hC}\right) < \frac{hC}{1 - hC} = \frac{\varepsilon}{\alpha \rho_L \rho_\sigma}.$$

By definitions of $F = \frac{\partial \mathcal{L}(\mathcal{R}_{\theta+\vartheta}(\omega),y)}{\partial x} \cdot \sigma'_{\mathring{\mathbf{M}}_o}(\mathring{\mathbf{h}}_T)$ we have on one side that :

$$\sigma'_{\mathring{\mathbf{M}}_o}(\mathring{\mathbf{h}}_T) \leq 1/4$$

because for all $x \in \mathbb{R}$ $\sigma'(x) \leq 1/4$. And on the other side by Assumption 1 we have:

$$\frac{\partial \mathcal{L}(\mathcal{R}_{\theta+\vartheta}(\omega), y)}{\partial x} \leq \rho_L \ .$$

We can conclude that:

$$\left| \frac{\partial L(\mathcal{R}_{\theta+\vartheta}(\omega), y)}{\partial \mathring{m}_{i,j}} \right| < \frac{\varepsilon}{\alpha} \ .$$

Hence $\mathcal{R}_{\theta+\vartheta}$ will experience a stationary gradient on all of its non zero parameters in the gradient descent with $\alpha$ as learning rate. ∎

# B    Stability of saturated SRNs

Our reasoning is based on certain facts concerning the saturation of the sigmoidal function. Indeed, a FP-SRN $\mathcal{R}$ respecting the assumptions of Theorem 16 is found to be stable to perturbations. These may well be perturbations in the data, or perturbations in the parameters. We are interested in parameter perturbations, because we want to exhibit an Euclidean ball in the parameter space centered on the parameters of $\mathcal{R}$ such that all FP-SRN $\tilde{\mathcal{R}}$ whose parameters come from this ball, will behave closely to $\mathcal{R}$. This stability is due to the fact that the tanh and sigmoidal functions have small variations near the edges of their target sets. Moreover, because of the exponential in the definition of these functions, they reach the edges of their target sets rapidly.

The aim of this section is to show that it is sufficient to noise only the bias of the parameters. This simplifies the proofs in the sequel, as fewer parameters need to be controlled.

## B.1    Propagation of the noise through the iterations.

**Definition 23 (Perturbed SRN)** *Given a SRN $\mathcal{R}_\theta$ as in Equation 5, and $\vartheta \in \mathbb{R}^{dim(\theta)}$, we define the perturbed SRN $\mathcal{R}_{\theta+\vartheta}$ by the following set of equations:*

$$\begin{cases} \mathring{\mathbf{h}}_k = \sigma\left( (\mathbf{U} + \vartheta_{\mathbf{U}})\mathbf{x}_k + (\mathbf{W} + \vartheta_{\mathbf{W}})\mathring{\mathbf{h}}_{k-1} + \mathbf{b} + \vartheta_{\mathbf{b}} \right) \\ \mathring{\mathbf{y}}_k = \sigma\left( (\mathbf{V} + \vartheta_{\mathbf{V}})\mathring{\mathbf{h}}_k + \mathbf{c} + \vartheta_{\mathbf{c}} \right) \end{cases} ,$$

*where $\vartheta = \mathbf{vec}(\vartheta_U, \vartheta_W, \vartheta_V, \vartheta_b, \vartheta_c)$, and $\mathring{\mathbf{h}}_0 = \mathbf{h}_0$.*

Perturbed SRNs are central to our analysis.

**Lemma 24 (Vector variant of the mean value theorem)** *Let $\mathbf{y}$, $\tau \in \mathbb{R}^d$ and let $f : \mathbb{R}^d \to \mathbb{R}^d$ be a differentiable function such that $f(\mathbf{y}) = (f_1(\mathbf{y}[1]), \dots, f_d(\mathbf{y}[d]))$ where every $f_i$ is differentiable. We define $\nabla f = (f_1', \dots, f_d')$. Then there exists $\mathbf{c} \in \mathbb{R}^d$ such that:*

$$f(\mathbf{y} + \tau) = f(\mathbf{y}) + f^\nabla(\mathbf{c}) \odot \tau$$

*and $\in \mathbf{y}[j] < \mathbf{c}[j] < (\mathbf{y} + \tau)[j]$ for all $1 \leq j \leq d$.*

**Proof** Let $\mathbf{y}$, $\tau \in \mathbb{R}^d$, and let $1 \leq j \leq d$. By the mean value theorem there exists $c_j$, $\mathbf{y}[j] < c_j < (\mathbf{y} + \tau)[j]$ such that:

$$f\big((\mathbf{y} + \tau)[j]\big) = f_j(\mathbf{y}[j]) + f_j'(c_j) \cdot \tau[j].$$

We can repeat this operation for any coordinate $1 \leq j \leq d$ and create a vector $\mathbf{c} = (c_1, \dots, c_d)$ such that:

$$f(\mathbf{y} + \tau) = f(\mathbf{y}) + f^\nabla(\mathbf{c}) \odot \tau.$$

∎

The following lemma is very handy in the expression of the hidden state vector of a fuzzy SRN.

**Lemma 25** *Let $\mathbf{x}, \mathbf{y} \in \mathbb{R}^d$ for $d \geq 1$, then:*

$$\mathrm{Diag}(\mathbf{x})\mathbf{y} = \mathbf{x} \odot \mathbf{y}$$

**Proof** Let $\mathbf{x}, \mathbf{y} \in \mathbb{R}^d$ for $d \geq 1$, and let $1 \leq j \leq d$. We define $\mathbf{D} = \mathrm{Diag}(\mathbf{x})$, we have:

$$
\begin{aligned}
\mathbf{D}\mathbf{y}[j] &= \sum_{i=1}^{d} \mathbf{D}_{j,i}\mathbf{y}[i] \\
&= \mathbf{D}_{j,j}\mathbf{y}[j] \text{ by definition of a diagonal matrix} \\
&= \mathbf{x}[j]\mathbf{y}[j] \\
&= (\mathbf{x} \odot \mathbf{y})[j]
\end{aligned}
$$

∎

**Lemma 26 (Expression of the perturbed hidden state vector)** *Let $\mathcal{R}_\theta$ be a SRN and $\mathcal{R}_{\mathbf{P}+\vartheta}$ a perturbed SRN with $\vartheta \in \mathbb{R}^{dim(\theta)}$. For all $\{\mathbf{x}_k\}_{k=1}^T \in \mathcal{X}$ there exists a sequence $\{\mathbf{c}_k\}_{k=1}^T \in \mathbb{R}^h$ such that for $1 \leq k \leq T$:*

$$
\mathring{\mathbf{h}}_k = \mathbf{h}_k + \sum_{s=1}^{k-1} \Lambda(\mathbf{c}_k) \left( \prod_{l=1}^{k-s} \mathbf{W}\Lambda(\mathbf{c}_{k-l}) \right) \mathbf{n}_s + \Lambda(\mathbf{c}_k)\mathbf{n}_k
$$

*where :*

- $\Lambda(\mathbf{c}_k) = \mathrm{Diag}\big(\sigma'(\mathbf{c}_k)\big)$ *with $\mathbf{c}_k$ provided by Lemma 24*

- $\prod_{l=1}^{k-s} \mathbf{W}\Lambda(\mathbf{c}_{k-l}) = \mathbf{W}\Lambda(\mathbf{c}_{k-1})\mathbf{W}\Lambda(\mathbf{c}_{k-2})\cdots\mathbf{W}\Lambda(\mathbf{c}_s)$.

- $\mathbf{n}_k = \left(\vartheta_\mathbf{U}\mathbf{x}_k + \vartheta_\mathbf{W}\mathring{\mathbf{h}}_{k-1} + \vartheta_\mathbf{b}\right)$

**Proof** Following the notations introduced in the statement of the lemma, we prove the result by induction on $k$. We analyze the execution of the perturbed SRN $\mathcal{R}_\theta^\varepsilon$. By Lemma 24, there exists $\mathbf{c}_1 \in \mathbb{R}^h$ such that:

$$
\begin{aligned}
\mathring{\mathbf{h}}_1 &= \sigma\left((\mathbf{U}\mathbf{x}_1 + \vartheta_\mathbf{U}\mathbf{x}_1) + (\mathbf{W}\mathbf{h}_0 + \vartheta_\mathbf{W}\mathbf{h}_0) + (\mathbf{b} + \vartheta_\mathbf{b})\right) \\
&= \sigma\left((\mathbf{U}\mathbf{x}_1 + \mathbf{W}\mathbf{h}_0 + \mathbf{b}) + (\vartheta_\mathbf{U}\mathbf{x}_1 + \vartheta_\mathbf{W}\mathbf{h}_0 + \vartheta_\mathbf{b})\right) \\
&= \sigma\left(\mathbf{U}\mathbf{x}_1 + \mathbf{W}\mathbf{h}_0 + \mathbf{b}\right) + \sigma^\nabla(\mathbf{c}_1) \odot (\vartheta_\mathbf{U}\mathbf{x}_1 + \vartheta_\mathbf{W}\mathbf{h}_0 + \vartheta_\mathbf{b}).
\end{aligned}
$$

By definition of the perturbed FP-SRN we have $\mathring{\mathbf{h}}_0 = \mathbf{h}_0$ which gives us the following equality:

$$
\left(\vartheta_\mathbf{U}\mathbf{x}_1 + \vartheta_\mathbf{W}\mathbf{h}_0 + \vartheta_\mathbf{b}\right) = \left(\vartheta_\mathbf{U}\mathbf{x}_1 + \vartheta_\mathbf{W}\mathring{\mathbf{h}}_0 + \vartheta_\mathbf{b}\right) =: \mathbf{n}_1.
$$

Hence we deduce the equality:

$$
\mathring{\mathbf{h}}_1 = \mathbf{h}_1 + \Lambda(\mathbf{c}_1)\mathbf{n}_1,
$$

where in the last line we set $\Lambda(\mathbf{c}) = \mathrm{Diag}(\sigma^\nabla(\mathbf{c}))$ and leverage the linearity of the Hadamard product proved by Lemma 25.

We proved the initialization step (*i.e.* $k = 1$) but for comprehensibility we prove the statement for $k = 2$. By reapplying the same argument we can find a vector $\mathbf{c}_2 \in \mathbb{R}^h$ such that:

$$
\begin{aligned}
\mathring{\mathbf{h}}_2 &= \sigma\left((\mathbf{U} + \vartheta_\mathbf{U})\mathbf{x}_2 + (\mathbf{W} + \vartheta_\mathbf{W})(\mathbf{h}_1 + \Lambda(\mathbf{c}_1)\mathbf{n}_1) + (\mathbf{b} + \vartheta_\mathbf{b})\right) \\
&= \sigma\left(\mathbf{U}\mathbf{x}_2 + \mathbf{W}(\mathbf{h}_1 + \Lambda(\mathbf{c}_1)\mathbf{n}_1) + \mathbf{b} + (\vartheta_\mathbf{U}\mathbf{x}_1 + \vartheta_\mathbf{W}\mathring{\mathbf{h}}_1 + \vartheta_\mathbf{b})\right) \\
&= \sigma\left(\mathbf{U}\mathbf{x}_2 + \mathbf{W}\mathbf{h}_1 + \mathbf{b}\right) + \sigma^\nabla(\mathbf{c}_2) \odot (\mathbf{W}\Lambda(c_1)\mathbf{n}_1) + \sigma^\nabla(\mathbf{c}_2) \odot (\vartheta_\mathbf{U}\mathbf{x}_1 + \vartheta_\mathbf{W}\mathring{\mathbf{h}}_1 + \vartheta_\mathbf{b}) \\
&= \mathbf{h}_2 + \Lambda(\mathbf{c}_2)\mathbf{W}\Lambda(\mathbf{c}_1)\mathbf{n}_1 + \Lambda(\mathbf{c}_2)\mathbf{n}_2 \\
&= \mathbf{h}_2 + \sum_{s=1}^{2-1} \Lambda(\mathbf{c}_2) \left( \prod_{l=1}^{2-s} \mathbf{W}\Lambda(\mathbf{c}_{2-l}) \right) \mathbf{n}_1 + \Lambda(\mathbf{c}_2)\mathbf{n}_2.
\end{aligned}
$$

Let us assume now that for $k \geq 1$ we have the following expression:

$$\mathring{\mathbf{h}}_k = \mathbf{h}_k + \sum_{s=1}^{k-1} \Lambda(\mathbf{c}_k) \left( \prod_{l=1}^{k-s} \boldsymbol{W}\Lambda(\mathbf{c}_{k-l}) \right) \mathbf{n}_s + \Lambda(\mathbf{c}_k)\mathbf{n}_k.$$

By Lemma 24 there exists a vector $\mathbf{c}_{k+1} \in \mathbb{R}^h$ such that:

$$\tilde{\mathbf{h}}_{k+1} = \sigma\left( (\mathbf{U} + \vartheta_{\mathbf{U}})\mathbf{x}_{k+1} + (\mathbf{W} + \vartheta_{\mathbf{W}})\Big(\mathbf{h}_k + \sum_{s=1}^{k-1} \Lambda(\mathbf{c}_k) \Big( \prod_{l=1}^{k-s} \mathbf{W}\Lambda(\mathbf{c}_{k-l}) \Big) \mathbf{n}_s + \Lambda(\mathbf{c}_k)\mathbf{n}_k \Big) + (\mathbf{b} + \vartheta_{\mathbf{b}}) \right)$$

$$= \sigma\left( \mathbf{U}\mathbf{x}_{k+1} + \mathbf{W}\Big(\mathbf{h}_k + \sum_{s=1}^{k-1} \Lambda(\mathbf{c}_k) \Big( \prod_{l=1}^{k-s} \mathbf{W}\Lambda(\mathbf{c}_{k-l}) \Big) \mathbf{n}_s + \Lambda(\mathbf{c}_k)\mathbf{n}_k \Big) + \mathbf{b} + \mathbf{n}_{k+1} \right),$$

where $\mathbf{n}_{k+1} := \vartheta_{\mathbf{U}}\mathbf{x}_{k+1} + \vartheta_{\mathbf{W}}\mathring{\mathbf{h}}_k + \vartheta_{\mathbf{b}}$. Now by applying Lemma 24 we obtain:

$$\mathring{\mathbf{h}}_{k+1} = \sigma\left( \mathbf{U}\mathbf{x}_{k+1} + \mathbf{W}\mathbf{h}_k + \mathbf{b} \right) + \sigma^{\nabla}(\mathbf{c}_{k+1}) \odot \left( \mathbf{W}\sum_{s=1}^{k-1} \Lambda(\mathbf{c}_k) \Big( \prod_{l=1}^{k-s} \mathbf{W}\Lambda(\mathbf{c}_{k-l}) \Big) \mathbf{n}_s + \mathbf{W}\Lambda(\mathbf{c}_k)\mathbf{n}_k + \mathbf{n}_{k+1} \right)$$

$$= \mathbf{h}_{k+1} + \Lambda(\mathbf{c}_{k+1}) \sum_{s=1}^{k-1} \left( \prod_{l=1}^{k-s+1} \mathbf{W}\Lambda(\mathbf{c}_{k-l+1}) \right) \mathbf{n}_s + \Lambda(\mathbf{c}_{k+1})\mathbf{W}\Lambda(\mathbf{c}_k)\mathbf{n}_k + \Lambda(\mathbf{c}_{k+1})\mathbf{n}_{k+1}$$

$$= \mathbf{h}_{k+1} + \sum_{s=1}^{k} \Lambda(\mathbf{c}_{k+1}) \left( \prod_{l=1}^{k-s+1} \mathbf{W}\Lambda(\mathbf{c}_{k-l+1}) \right) \mathbf{n}_s + \Lambda(\mathbf{c}_{k+1})\mathbf{n}_{k+1}.$$

Thus by induction Lemma 26 is true for all $\{\mathbf{x}_1, \ldots, \mathbf{x}_T\} \in \mathcal{X}$. ∎

The following result is on the resilience to noise that saturated FP-SRNs exhibit. This result is based on saturation, but the $\zeta$ border is different from the one defined in Theorem 16. To avoid confusion in this result, we have chosen to write $\tilde{\zeta}$ and $\tilde{\eta}$.

**Theorem 27** *Let $\mathcal{R}_\theta$ be a $\tilde{\zeta} + \tilde{\eta}$-saturated FP-SRN, with $z = \frac{1}{\|\mathbf{W}\|}$ and $\tilde{\zeta} = \ln\left( \frac{1+\sqrt{1-4z}}{1-\sqrt{1-4z}} \right)$, for some $\tilde{\eta} > 0$. We assume that $\mathcal{R}_\theta$ is operating on one-hot encoded data. We select any real number $t$ such that $0 < t < 1$, then for all $\vartheta := \mathbf{vec}(\vartheta_{\mathbf{U}}, \vartheta_{\mathbf{W}}, \vartheta_{\mathbf{V}}, \vartheta_{\mathbf{b}}, \vartheta_{\mathbf{c}}) \in \mathbb{R}^{dim(\theta)}$ such that $\|\vartheta\| \leq (1-t)\tilde{\eta} \cdot \frac{(1-\Delta\|\mathbf{W}\|)}{(2+\sqrt{h})}$, with $\Delta = \sigma'(\tilde{\zeta} + t\tilde{\eta})$ and for all $\{\mathbf{x}_1, \ldots, \mathbf{x}_T\} \in \mathcal{X}$ we have:*

1. *$\|\mathring{\mathbf{h}}_k - \mathbf{h}_k\|_\infty \leq \frac{(2+\sqrt{h})}{1-\Delta\|\mathbf{W}\|}\|\vartheta\|$ where $1 \leq k \leq T$.*

2. *$\mathcal{R}_{\theta+\vartheta}$ is $(\tilde{\zeta} + t\tilde{\eta})$-saturated.*

*The sequence of hidden state vectors $\mathring{\mathbf{h}}_k$ is produced by the perturbed FP-SRN $\mathcal{R}_{\mathbf{P}+\vartheta}$.*

This theorem proves, by induction, that saturated FP-SRNs can be stable to parameter perturbations.

**Proof** Let $\mathcal{R}_\theta$ be a $\tilde{\zeta} + \tilde{\eta}$-saturated FP-SRN, with $z = \frac{1}{\|\mathbf{W}\|}$ and $\tilde{\zeta} = \ln\left( \frac{1+\sqrt{1-4z}}{1-\sqrt{1-4z}} \right)$, $\tilde{\eta} > 0$. We assume that $\mathcal{R}_\theta$ operates on one-hot-encoded data. Let $\vartheta := \mathbf{vec}(\vartheta_{\mathbf{U}}, \vartheta_{\mathbf{W}}, \vartheta_{\mathbf{V}}, \vartheta_{\mathbf{b}}, \vartheta_{\mathbf{c}}) \in \mathbb{R}^{dim(\theta)}$ such that $\|\vartheta\| \leq (1-t)\tilde{\eta} \cdot \frac{(1-\Delta\|\mathbf{W}\|)}{(2+\sqrt{h})}$ with $\Delta := \sigma'(\tilde{\zeta} + t\tilde{\eta})$. We also fix a sequence $\{\mathbf{x}_1, \ldots, \mathbf{x}_T\} \in \mathcal{X}$. Before we start with the induction proof, we summon Lemma 20 in order to validate the assertion that $0 \leq \Delta\|\mathbf{W}\| < 1$. We start the proof by induction naturally with $k = 1$ i.e. $\|\mathring{\mathbf{h}}_1 - \mathbf{h}_1\| \leq \frac{(2+\sqrt{h})}{(1-\Delta\|\mathbf{W}\|)}\|\vartheta\|$. By Lemma 26 we know that one can find $\mathbf{c}_1 \in \mathbb{R}^h$ such that:

$$\mathring{\mathbf{h}}_1 - \mathbf{h}_1 = \Lambda(\mathbf{c}_1)\mathbf{n}_1, \tag{5}$$

with $\mathbf{n}_1 = \vartheta_{\mathbf{U}}\mathbf{x}_1 + \vartheta_{\mathbf{W}}\mathring{\mathbf{h}}_0 + \vartheta_{\mathbf{b}}$. We will prove that $\|\Lambda(\mathbf{c}_1)\| \leq \Delta$ and $\|\mathbf{n}_1\| \leq (1-t)\tilde{\eta}$, and we start with the second inequality. One can observe:

$$\|\mathbf{n}_1\| = \|\vartheta_{\mathbf{U}}\mathbf{x}_1 + \vartheta_{\mathbf{W}}\mathring{\mathbf{h}}_0 + \vartheta_{\mathbf{b}}\|$$
$$\leq \|\vartheta_{\mathbf{U}}\mathbf{x}_1\| + \|\vartheta_{\mathbf{W}}\mathring{\mathbf{h}}_0\| + \|\vartheta_{\mathbf{b}}\|$$
$$\leq \|\vartheta_{\mathbf{U}}\|\|\mathbf{x}_1\| + \|\vartheta_{\mathbf{W}}\|\|\mathring{\mathbf{h}}_0\| + \|\vartheta_{\mathbf{b}}\|.$$

We have assumed that $\mathcal{R}_\theta$ operates on one-hot-encoded data, thus $\|\mathbf{x}_k\| \leq 1$ for all elements of the alphabet. Moreover, by definition of FP-SRNs and SRNs the norm of the hidden state vector is always bounded by $\sqrt{h}$, leading us to the bound:

$$\|\mathbf{n}_1\| \leq \|\vartheta_{\mathbf{U}}\|\|\mathbf{x}_1\| + \|\vartheta_{\mathbf{W}}\|\|\mathring{\mathbf{h}}_0\| + \|\vartheta_{\mathbf{b}}\|$$
$$\leq \|\vartheta_{\mathbf{U}}\| + \sqrt{h}\|\vartheta_{\mathbf{W}}\| + \|\vartheta_{\mathbf{b}}\|$$
$$\leq \|\vartheta_{\mathbf{U}}\|_{\mathrm{Fro}} + \sqrt{h}\|\vartheta_{\mathbf{W}}\|_{\mathrm{Fro}} + \|\vartheta_{\mathbf{b}}\|_{\mathrm{Fro}}$$
$$\leq \|\vartheta\| + \sqrt{h}\|\vartheta\| + \|\vartheta\|$$
$$= (2 + \sqrt{h})\|\vartheta\|$$
$$\leq \frac{(2 + \sqrt{h})}{(1 - \Delta\|\mathbf{W}\|)}\|\vartheta\| \text{ because } 0 < (1 - \Delta\|\mathbf{W}\|) \leq 1$$

Since by definition $\|\vartheta\| \leq (1-t)\tilde{\eta}\frac{(1-\Delta\|\mathbf{W}\|)}{(2+\sqrt{h})}$ we can assert that $\|\mathbf{n}_1\| \leq (1-t)\tilde{\eta}$. Consequently for $1 \leq j \leq h$: $|\mathbf{n}_1[j]| \leq \frac{\tilde{\eta}}{2}$. We bring the Readers attention on the fact that the bound:

$$\|\mathbf{n}_1\| \leq \|\vartheta_{\mathbf{U}}\| + \sqrt{h}\|\vartheta_{\mathbf{W}}\| + \|\vartheta_{\mathbf{b}}\|$$

is independent of the iteration, *i.e.* for all $\{\mathbf{x}_1, \ldots, \mathbf{x}_T\} \in \mathcal{X}$ we know that for $1 \leq k \leq T$ $\|\mathbf{x}_k\| \leq 1$ because $\mathbf{x}_k$ are one-hot encodings of a finite alphabet, and $\mathring{\mathbf{h}}_k \in \left[-1, 1\right]^h$ thus $\|\mathring{\mathbf{h}}_k\| \leq \sqrt{h}$. By extension we are able to bound all $\|\mathbf{n}_k\|$ by $(2 + \sqrt{h})\|\vartheta\|$. Now we discus the inequality $\|\Lambda(\mathbf{c}_1)\| \leq \Delta$. Lemma 24 describes the location of the vector $\mathbf{c}_1$, which is for $1 \leq j \leq h$, $\mathbf{c}_1[j]$ is in the interval $\left]x, x + \mathbf{n}_1[j]\right[$, $x$ being defined by $x := \left(\mathbf{U}\mathbf{x}_1 + \mathbf{W}\mathbf{h}_0 + \mathbf{b}\right)[j]$. Therefore $|x + \mathbf{n}_1[j]| \geq \tilde{\zeta} + t\tilde{\eta}$ because $|\mathbf{n}_1[j]| \leq t\tilde{\eta}$, thus by extension:

$$\mathbf{c}_1[j] \in \left]-\tilde{\zeta} - \tilde{\eta}, -\tilde{\zeta} - t\tilde{\eta}\right[ \cup \left]\tilde{\zeta} + t\tilde{\eta}, \tilde{\zeta} + \tilde{\eta}\right[.$$

We draw the Reader's attention to the fact that

$$|\sigma^{-1}(\mathring{\mathbf{h}}_1[j])| \geq \tilde{\zeta} + t\tilde{\eta}$$

In order to bound $\|\Lambda(\mathbf{c}_1)\|$, the quantity of interest for us is:

$$\sup_{x \in S} \sigma'(x), \ \ S := \left]-\tilde{\zeta} - \tilde{\eta}, -\tilde{\zeta} - t\tilde{\eta}\right[ \cup \left]\tilde{\zeta} + t\tilde{\eta}, \tilde{\zeta} + \tilde{\eta}\right[$$

which is easy to determine because $\sigma'$ is symmetric and is strictly decreasing on the interval $[0, +\infty[$. We obtain, quite simply, that:

$$\sup_{x \in S} \sigma'(x) = \sigma'\left(\tilde{\zeta} + t\tilde{\eta}\right).$$

This reasoning does not depend on $1 \leq j \leq h$, Therefore we can assert that:

$$\|\Lambda(\mathbf{c}_1)\| = \|\sigma^\nabla(\mathbf{c}_1)\|_\infty \leq \sigma'(\tilde{\zeta} + t\tilde{\eta}).$$

To sum up, we have proved that:

- $\|\mathring{\mathbf{h}}_1 - \mathbf{h}_1\|_\infty \leq \frac{(2+\sqrt{h})}{(1+\Delta\|\mathbf{W}\|)}\|\vartheta\|,$

- the first iteration of $\mathcal{R}_{\theta+\vartheta}$ is $(\tilde{\zeta} + t\tilde{\eta})$-saturated,

- $\|\Lambda(\mathbf{c}_1)\| \leq \Delta,$

now we pursue with the induction hypothesis which is :
for all $l \in \{1\ldots,k\}$ we have:

- $\|\mathring{\mathbf{h}}_l - \mathbf{h}_l\|_\infty \leq \frac{(2+\sqrt{h})}{(1+\Delta\|\mathbf{W}\|)}\|\vartheta\|,$

- the $l^{th}$ iteration is $(\tilde{\zeta} + t\tilde{\eta})$-saturated,

- $\|\Lambda(\mathbf{c}_l)\| \leq \Delta.$

By Lemma 26 we have an expression of the difference between $\mathring{\mathbf{h}}_{k+1}$ and $\mathbf{h}_{k+1}$ which is:

$$\mathring{\mathbf{h}}_{k+1} - \mathbf{h}_{k+1} = \sum_{s=1}^{k} \Lambda(\mathbf{c}_{k+1}) \left( \prod_{l=1}^{k-s+1} \mathbf{W}\Lambda(\mathbf{c}_{k-l+1}) \right) \mathbf{n}_s + \Lambda(\mathbf{c}_{k+1})\mathbf{n}_{k+1}$$

$$= \Lambda(\mathbf{c}_{k+1}) \underbrace{\left[ \sum_{s=1}^{k} \left( \prod_{l=1}^{k-s+1} \mathbf{W}\Lambda(\mathbf{c}_{k-l+1}) \right) \mathbf{n}_s + \mathbf{n}_{k+1} \right]}_{=:\mathbf{g}}.$$

To simplify the notations we regrouped all the disturbances in the FP-SRN encoder, before activation, in $\mathbf{g}$. We have to prove that $\|\mathbf{g}\| \leq \frac{(2+\sqrt{h})}{(1+\Delta\|\mathbf{W}\|)}\|\vartheta\|$. We recall that $\Delta = \sigma'(\tilde{\zeta} + t\tilde{\eta})$, by induction hypothesis we know that $\|\Lambda(\mathbf{c}_l)\| \leq \Delta$ for all $l \in \{1,\ldots,k\}$ Therefore we can assert that :

$$\|\mathbf{g}\| = \left\| \sum_{s=1}^{k} \left( \prod_{l=1}^{k-s+1} \mathbf{W}\Lambda(\mathbf{c}_{k-l+1}) \right) \mathbf{n}_s + \mathbf{n}_{k+1} \right\|$$

$$\leq \sum_{s=1}^{k} \left( \prod_{l=1}^{k-s+1} \|\mathbf{W}\|\|\Lambda(\mathbf{c}_{k-l+1})\| \right) \|\mathbf{n}_s\| + \|\mathbf{n}_{k+1}\|$$

$$\leq \sum_{s=1}^{k} \left( \prod_{l=1}^{k-s+1} \Delta\|\mathbf{W}\| \right) \|\mathbf{n}_s\| + \|\mathbf{n}_{k+1}\|$$

$$= \sum_{s=1}^{k} (\Delta\|\mathbf{W}\|)^{k-s+1} \|\mathbf{n}_s\| + \|\mathbf{n}_{k+1}\|$$

Since summing from $s = k$ to $s = 1$ is exactly the same as summing from $s = 1$ to $s = k$, we can write:

$$\sum_{s=1}^{k} (\Delta\|\mathbf{W}\|)^{k-s+1} \|\mathbf{n}_s\| + \|\mathbf{n}_{k+1}\| = \sum_{s=1}^{k} (\Delta\|\mathbf{W}\|)^s \|\mathbf{n}_s\| + \|\mathbf{n}_{k+1}\|$$

$$= \sum_{s=1}^{k} (\Delta\|\mathbf{W}\|)^s \|\mathbf{n}_s\| + (\Delta\|\mathbf{W}\|)^0 \|\mathbf{n}_{k+1}\|$$

$$= \sum_{s=0}^{k} (\Delta\|\mathbf{W}\|)^s \|\mathbf{n}_s\|$$

$$\leq \sum_{s=0}^{k} (\Delta\|\mathbf{W}\|)^s \left((2 + \sqrt{h})\|\vartheta\|\right)$$

$$\leq \sum_{s=0}^{\infty} (\Delta\|\mathbf{W}\|)^s \left((2 + \sqrt{h})\|\vartheta\|\right)$$

$$= \left((2 + \sqrt{h})\|\vartheta\|\right) \frac{1}{1 - \Delta\|\mathbf{W}\|},$$

yet by hypothesis on $\vartheta$ we know that

$$\left((2 + \sqrt{h})\|\vartheta\|\right) \leq (1 - \Delta\|\mathbf{W}\|)(1 - t)\tilde{\eta}.$$

Hence one can deduce that:

$$\|\mathbf{g}\| \leq (1 - t)\tilde{\eta}.$$

For the location of $\mathbf{c}_{k+1}$ we apply the exact same argument as for $\mathbf{c}_1$, *i.e.* :

$$\tilde{\mathbf{h}}_{k+1} = \sigma\left(\mathbf{U}\mathbf{x}_{k+1} + \mathbf{W}\mathbf{h}_k + \mathbf{b} + \mathbf{m}\right)$$

$$= \sigma\left(\mathbf{U}\mathbf{x}_{k+1} + \mathbf{W}\mathbf{h}_k + \mathbf{b}\right) + \Lambda(\mathbf{c}_{k+1})\mathbf{m}.$$

We established that for all $1 \leq j \leq h$, $|\mathbf{g}[j]| \leq \|\mathbf{g}\| \leq (1 - t)\tilde{\eta}$, Therefore:

$$\mathbf{c}_{k+1}[j] \in \left] -\tilde{\zeta} - \tilde{\eta}, -\tilde{\zeta} - t\tilde{\eta}\right[ \cup \left]\tilde{\zeta} + t\tilde{\eta}, \tilde{\zeta} + \tilde{\eta}\right[.$$

Hence by the properties of $\sigma'$ function, and of the $\Lambda(\mathbf{c}_{k+1})$ matrix we can assert that $\|\Lambda(\mathbf{c}_{k+1})\| \leq \sigma'\left(\tilde{\zeta} + t\tilde{\eta}\right)$. We conclude that the proof by induction is valid. ∎

In Theorem 16 and Theorem 14 we use a notion of thresholds $\zeta, \tilde{\zeta}$ in the saturation of the activation function $\sigma$. Moreover, in Theorem 16 these two values are interacting, leaving ambiguity as to whether they can interact. The following lemme is there to clear this ambiguity.

**Lemma 28** *Let $\mathcal{R}_\theta$ by a $(\tilde{\zeta} + \tilde{\eta})$-saturated FP-SRN with $\zeta = \ln\left(\frac{1 + \sqrt{1 - \frac{4}{\|\mathbf{W}\|}}}{1 - \sqrt{1 - \frac{4}{\|\mathbf{W}\|}}}\right)$ and $\tilde{\eta} > 0$. Let $\zeta \geq 0$ and $\eta > 0$ be two real numbers such that $\zeta + \eta = \tilde{\zeta} + \tilde{\eta}$ and $\tilde{\zeta} \leq \zeta$, then for all $\vartheta \in \mathbb{R}^{dim(\theta)}$ verifying $\|\vartheta\| \leq \frac{\eta}{2}\frac{(1 - \Delta\|\mathbf{W}\|)}{2 + \sqrt{h}}$ with $\Delta = \sigma'(\zeta + \eta/2)$ the FP-SRN $\mathcal{R}_{\theta+\vartheta}$ is $(\zeta + \eta/2)$-saturated.*

**Proof** Let $\mathcal{R}_\theta$ by a $(\tilde{\zeta} + \tilde{\eta})$-saturated FP-SRN with $\tilde{\zeta} = \ln\left(\frac{1 + \sqrt{1 - \frac{4}{\|\mathbf{W}\|}}}{1 - \sqrt{1 - \frac{4}{\|\mathbf{W}\|}}}\right)$ and $\tilde{\eta} > 0$. Let $\zeta \geq 0$ and $\eta > 0$ be two real numbers such that $\zeta + \eta = \tilde{\zeta} + \tilde{\eta}$ and $\tilde{\zeta} \leq \zeta$. Let $\vartheta \in \mathbb{R}^{dim(\theta)}$ verifying $\|\vartheta\| \leq \frac{\eta}{2}\frac{(1 - \Delta\|\mathbf{W}\|)}{2 + \sqrt{h}}$. From the hypothesis $\zeta + \eta = \tilde{\zeta} + \tilde{\eta}$ one can deduce that

$$\zeta = \tilde{\zeta} + \tilde{\eta} - \eta$$

$$\eta/2 = \tfrac{1}{2}(\tilde{\zeta} + \tilde{\eta} - \zeta).$$

The hypothesis $\zeta \geq \tilde{\zeta}$ implies that $\tilde{\eta} \leq \eta$ Therefore, there exists $t \in [0,1]$ such that $\frac{\eta}{2} = (1-t) \cdot \tilde{\eta}$. In the same way, the quantity $\zeta + \eta/2$ can be transformed into $\tilde{\zeta} + t \cdot \tilde{\eta}$, indeed:

$$\zeta + \eta/2 = \tilde{\zeta} + \tilde{\eta} - \eta + \eta/2$$
$$= \tilde{\zeta} + \tilde{\eta} - \eta/2$$
$$= \tilde{\zeta} + \tilde{\eta} - (1-t) \cdot \tilde{\eta}$$
$$= \tilde{\zeta} + t \cdot \tilde{\eta}$$

Hence $\Delta = \sigma'(\zeta + \eta/2) = \sigma'(\tilde{\zeta} + t \cdot \tilde{\eta})$ and $\eta/2 = (1-t) \cdot \tilde{\eta}$ with $t \in ]0,1[$, allowing us to conclude by Theorem 14 that the claim of Lemma 28 is true. ∎

## C  DFA2SRN algorithm

In this appendix, we provide a proof that the algorithm presented in Section 5 is well defined. We start by recalling the definition of the Euclidean division, then the algorithm DFA2SRN, followed by the proof of its well definition.

**Definition 29 (Euclidean division)** *Let $(l, j)$ two positive integers greater than one. There exists a unique couple $(a, b)$ of positive integers such that:*

$$l = aj + b \text{ where } 0 \leq b < j.$$

*We call a the **quotient** and b the **rest**. We denote the **quotient** $l//j := a$.*

Now we recall the algorithm DFA2SRN.

Let $\mathcal{D} = (\Sigma, Q, q_1, \delta, F)$ be a DFA. We suppose that the alphabet $\Sigma = \{\mathbf{e}_1, \ldots, \mathbf{e}_u\}$ is a set of one-hot vectors. We recall that $Q = \{q_1, \ldots, q_{|Q|}\}$ is the set of states, $q_1$ is the initial state, $\delta : \Sigma \times Q \to Q$ is the transition function, and $F \subset Q$ is the set of final states.. In order to simulate the DFA, we have to simulate the transition function, to encode every state such that it can be read by the FP-SRN, and then the model has to simulate the function discriminating between elements of $F$ and $Q \setminus F$. The encoding of the states of the DFA follows the spirit of one-hot encoding. The transition function is simulated by the encoder and the discrimination function will be simulated by the decoder.

The main idea is to make the hidden states $\mathbf{h}_t$, $t \geq 1$, be an encoding of the state $q$ reached in the DFA while parsing a sequence up to its $t^{th}$ element. By construction, using the saturation float $J$, the FP-SRN $\mathcal{R}_\theta$ will produce a One-hot hidden vector at each iteration. In order to perform the simulation we need to set the dimension $h$ of the hidden vector to $h = |\Sigma| \cdot |Q|$, and groups of $|\Sigma|$ One-hot encoded vectors will represent a unique state $q \in Q$. For example, if $\Sigma = \{a, b\}$ an alphabet with two symbols, the vector $\begin{pmatrix} 1 & 0 & 0 & \ldots & 0 \end{pmatrix}$ will correspond to the state $q_1$, but so will $\begin{pmatrix} 0 & 1 & 0 & \ldots & 0 \end{pmatrix}$. More generally, vectors where only one of the first $|\Sigma|$ coordinates is equal to 1, and the others are nil, will represent the $q_1$ state. In this construction, $\mathbf{M}_h$ will encode the transition function, and $\mathbf{M}_o$ will encode the $q \in F$ relation. In the rest of this section we will refer to the FP-SRN definition version 2 and use the formalism of kernels $\mathbf{W}$ the transition kernel $\mathbf{U}$ the input kernel and $\mathbf{v}$ the output kernel (the output kernel is a vector because the output that we need is a scalar in $\{0, 1\}$).

We set $|Q| := Card(Q)$, $|\Sigma| := Card(\Sigma)$. We assume that $Q = \{q_1, q_2, \ldots, q_{|Q|}\}$ and we define $I_F \subset \{1, \ldots, |Q||\Sigma|\}$ with

$$(\{i, i+1, \ldots, (i + |\Sigma|) - 1\} \subset I_F \iff q_i \in F)$$

The linear parts of the SRN are defined by:

$$\mathbf{W} \in \mathbb{G}^{|Q||\Sigma| \times |Q||\Sigma|} \ ; \ \mathbf{U} \in \mathbb{G}^{|Q||\Sigma| \times |\Sigma|}$$

$$\mathbf{h}_0 \in \mathbb{G}^{|Q||\Sigma|} \text{ with } \mathbf{h}_0[1] = 1, \mathbf{h}_0[j] = 0 \text{ if } j \neq 1;$$

$$\mathbf{v} \in \mathbb{G}^{1 \times |Q||\Sigma|} \text{ with } \mathbf{v}[j] = J \text{ if } j \in I_F \text{ and } -J \text{ if } j \notin I_F$$

$J$ is the smallest float number saturating the activation function as defined in Lemma 3.

We define $\mathbf{U} := 2J \cdot \left.\begin{bmatrix} \mathbf{I}_{|\Sigma|} \\ \vdots \\ \mathbf{I}_{|\Sigma|} \end{bmatrix}\right\} |Q|$ identity matrices $\mathbf{I}_{|\Sigma|}$ piled up. For the construction of $\mathbf{W}$, we start by

defining for every state $q_k$, $1 \le k \le |Q|$, a column vector $\mathbf{w}^k \in \mathbb{G}^{|Q||\Sigma|}$ by the following rules:

R1: For $1 \le s \le |\Sigma|$ if $\delta(q_k, a_s) = q_j$ then $\mathbf{w}^k[r] = -1$ with $r = (j-1)|\Sigma| + s$

R2: All the other coordinates of $\mathbf{w}^k$ are set to $-3$

Note that by definition of a DFA every vector $\mathbf{w}^k$ has exactly $|\Sigma|$ coordinates set to $-1$ and $(|Q|-1)|\Sigma|$ coordinates set to $-3$. Also, by definition of a DFA, every state can be reached by $|\Sigma|$ transitions. The rule R1 defines an encoding for $(q_k, a_s, \delta(q_k, a_s))$ Therefore, the vector $\mathbf{w}^k$ contains all the transitions starting at $q_k$.

Afterward we generate, for every $q_k \in Q$, the matrix:

$$\mathbf{W}^k := \underbrace{\begin{bmatrix} \mathbf{w}^k & \mathbf{w}^k & \cdots & \mathbf{w}^k \end{bmatrix}}_{|\Sigma| \text{ times}} \in \mathbb{G}^{|Q||\Sigma| \times |Q||\Sigma|}$$

$$\mathbf{W} := J \cdot \begin{bmatrix} \mathbf{W}^1 & \mathbf{W}^2 & \cdots & \mathbf{W}^{|Q|} \end{bmatrix} \in \mathbb{G}^{|Q||\Sigma| \times |Q||\Sigma|}.$$

(6)

In the definition of $\mathbf{W}^k$ the vector $\mathbf{w}^k$ is repeated $|\Sigma|$ times.

**Proof** . In the following paragraph we prove that the constructed FP-SRN is simulating the functioning of the fixed DFA. For this purpose, in first place, we have to prove that, for any word $\omega = \mathbf{e}_1 \cdots \mathbf{e}_T \in \Sigma^*$ of length $T$, there exists a well defined correspondence between

$$\left[\sigma\Big(\mathbf{U}\mathbf{e}_1 + \mathbf{W}\mathbf{h}_0\Big), \ldots, \sigma\Big(\mathbf{U}\mathbf{e}_T + \mathbf{W}\mathbf{h}_{T-1}\Big)\Big)\right] \xmapsto{f} [\delta^*(q_1, \mathbf{e}_1), \delta^*(q_1, \mathbf{e}_1\mathbf{e}_2), \ldots, \delta^*(q_1, \omega)]$$

(7)

The correspondence $f$ will be explicitly defined later. Secondly we will prove that we have:

$$\left[\delta(q_1, \omega) \in F\right] \iff \left[\sigma\big(\mathbf{v}\mathbf{h}_T\big) = 1\right]$$

(8)

The proof is by induction on $k \in \{1, \ldots, T\}$, and we naturally start with $k = 1$. Let $\mathbf{e}_s \in \Sigma$ with $1 \le s \le |\Sigma|$, and $\mathbf{h}_0$ the initial hidden state vector as defined above. We are going to compute $\mathbf{h}_1 = \sigma(\mathbf{U}\mathbf{e}_s + \mathbf{W}\mathbf{h})$ and show that $\mathbf{h}_1$ will have a unique non zero coordinate whose position encodes the state $\delta^*(q_1, \mathbf{e}_s) = \delta(q_1, \mathbf{e}_s)$. We start with $\mathbf{U}\mathbf{e}_s$:

$$\mathbf{U}\mathbf{e}_s[k] = 2J \text{ if } (k \equiv s \text{ modulo } |\Sigma|) \text{ and } 0 \text{ otherwise.}$$

We obtain this because of the structure of $\mathbf{U}$ ($|\Sigma|$ identity matrices piled up and multiplied by $2J$) and $\mathbf{e}_s$ is a one-hot encoded vector. Then we proceed to the computation of $\mathbf{W}\mathbf{h}_0$:

$$\mathbf{W}\mathbf{h}_0 = J \cdot \mathbf{w}^1 \text{ by construction of } \mathbf{W} \text{ and definition } \mathbf{h}_0.$$

Indeed, we have $\mathbf{h}_0[1] = 1$ and for $2 \le i \le |Q||\Sigma|$ $\mathbf{h}_0[i] = 0$. There is a unique $q \in Q$ such that $\delta(q_1, \mathbf{e}_s) = q$. We assume that $\delta(q_1, \mathbf{e}_s) = q_j$ for $1 \le j \le |Q|$. By construction of $\mathbf{w}^1$, we have:

$$\mathbf{w}^1[r] = -1 \text{ with } r = (j-1)|\Sigma| + s$$

referring to the rule R1. By definition of a DFA, it is not possible to have $\tilde{j} \in \{1, \ldots, |Q|\}, \tilde{j} \ne j$ and $\delta(q_1, \mathbf{e}_s) = q_{\tilde{j}}$, Therefore for all $\tilde{r} \in \{1, \ldots, |Q||\Sigma|\}$ such that $\mathbf{w}^1[\tilde{r}] = -1$ we have:

$$(\tilde{r} \equiv s \text{ modulo } |\Sigma|) \iff r = \tilde{r}.$$

This last point is important for the computation of $\mathbf{U}\mathbf{e}_s + \mathbf{W}\mathbf{h}_0$, indeed we will have:

$$(\mathbf{U}\mathbf{e}_s + \mathbf{W}\mathbf{h}_0)[r] = 2J - J = J$$

$$(\mathbf{U}\mathbf{e}_s + \mathbf{W}\mathbf{h}_0)[t] = 2J - 3J \text{ or } 0 - 3J \text{ or } 0 - J$$

with $t \in \{1, \ldots, |Q|\}$ $t \neq r$. Most importantly, the vector $\mathbf{U}\mathbf{e}_s + \mathbf{W}\mathbf{h}_0$ has only one coordinate with the value $J$, and the other coordinates are less than $-J$. Hence when we apply $\sigma$ in finite precision to $\mathbf{U}\mathbf{e}_s + \mathbf{W}\mathbf{h}_0$ we obtain the vector $\mathbf{h}_1$ with the property:

$$\mathbf{h}_1[r] = 1 \text{ and for } t \neq r \ \mathbf{h}_1[t] = 0.$$

Since $r = (j-1)|\Sigma| + s$, by Definition 29 we get that $r // |\Sigma| + 1 = j$, corresponding to the state $q_j = \delta^*(q_1, \mathbf{e}_s)$. Hence we can explicitly define the correspondence $f$ as follows:

$$f : \mathbb{R}^{|Q||\Sigma|} \ni \mathbf{h} \mapsto g(\mathbf{h}) // |\Sigma| + 1 \ \in \{1, \ldots, |Q|\},$$

where $g$ is the function that returns the index of the greatest coordinate of the vector $\mathbf{h}$.

We proved that the FP-SRN $\mathcal{R}$ successfully simulates the DFA $\mathcal{D}$ on all words of length $k = 1$. We make the induction assumption that for all words $\omega \in \Sigma^*$ such that $|\omega| = T \leq k$ we have:

- We have $(\mathbf{h}_1, \ldots, \mathbf{h}_T) \overset{f}{\longmapsto} (\delta^*(q_1, \mathbf{e}_1), \ldots, \delta^*(q_1, \omega))$.

- The hidden state vectors $\mathbf{h}_1, \ldots, \mathbf{h}_T$ are all one-hot encoded.

We are going to show that induction hypothesis is true for $k + 1$.

Let $\omega \in \Sigma^*$, such that $|\omega| = k + 1$, let $(\mathbf{h}_1, \ldots, \mathbf{h}_{k+1})$ the sequence of hidden state vectors produced by $\mathcal{R}$ from the execution on $\omega$. We know that $\omega = \tilde{\omega} \cdot \mathbf{e}_{\tilde{s}}$, where $\tilde{\omega}$ is a prefix of $\omega$, of length $k$. We know for sure that $|\tilde{\omega}| \geq 1$ because by assumption $k \geq 1$. By the induction assumption we have that:

$$(\mathbf{h}_1, \ldots, \mathbf{h}_k) \overset{f}{\longmapsto} (\delta^*(q_1, \mathbf{e}_1), \ldots, \delta^*(q_1, \tilde{\omega})),$$

and all hidden vectors $\mathbf{h}_t$, for $1 \leq t \leq k$, are one-hot encoded. We also know by the induction hypothesis that for $j := f(\mathbf{h}_k)$ we have $q_j = \delta^*(q_1, \tilde{\omega})$. It is sufficient to apply the same arguments as in the case $k = 1$ in order to conclude the proof by induction. Indeed by construction of the matrices $\mathbf{U}$ and $\mathbf{W}$ we know that there exists a unique $\tilde{r} \in \{1, \ldots, |Q||\Sigma|\}$ such that:

$$(\mathbf{U}\mathbf{e}_{\tilde{s}} + \mathbf{W}\mathbf{h}_k)[\tilde{r}] = J$$
$$(\mathbf{U}\mathbf{e}_{\tilde{s}} + \mathbf{W}\mathbf{h}_k)[\tilde{t}] \leq -J \text{ for } t \neq \tilde{r}$$

and by the construction of $\mathbf{U}$ and $\mathbf{W}$, and more particularly by construction of the vector $\mathbf{w}^j$, a vector encoding all possible transitions from the state $q_j$, we know that $\tilde{j} := \tilde{r} // |\Sigma| + 1$ corresponds to $q_{\tilde{j}} = \delta^*(q_1, \omega)$. From the identity just above we can deduce that after the application of the activation function $\sigma$, the saturation will occur and the outputted vector $\mathbf{h}_{k+1}$ will be one-hot encoded, with a 1 at the position $\tilde{j}$. We can thus conclude that the proof by induction is valid.

■

## D    Finite precision arithmetic's results

**Lemma 30 (smallest increment)** *Let $(B, M, X)$ be a finite precision configuration and $a \in \mathbb{G}_{(B,M,X)}$. We define $L_M$ as the largest positive numbers in $\mathbb{G}_{(B,M,X)}$. If $0 \leq a < L_M$ there exists $r \in \mathbb{G}_{(B,M,X)}$ such that in finite precision arithmetic:*

$$a + r > a \text{ and } \forall \ 0 \leq s < r, \ a + s = a$$

**Proof**    Let $(B, M, X)$ be a finite precision configuration and $a \in \mathbb{G}_{(B,M,X)}$. We define $L_M$ as the largest positive number in $\mathbb{G}_{(B,M,X)}$. We assume that $0 \leq a < L_M$ and we write the float $a$ in its normal form:

$$a = a_1.a_2 \cdots a_M B^{\lambda_a}$$

with $a_i \in \{0, \dots, B-1\}$ for $1 \leq i \leq M$ and $a_1 > 0$. Now we define the minimal increment

$$r := B^{\lambda_a - M + 1} \, .$$

By hypothesis we have $a < L_M$. Therefore $a + r \leq L_M$ and:

$$a + r = a_1.a_2 \cdots a_M B^{\lambda_a} + B^{\lambda_a - M}$$

$$= \left( \sum_{i=1}^{M-1} a_i B^{(-i+1)} + (a_M + 1) B^{(-M+1)} \right) B^{\lambda_a}$$

$$> a \, .$$

Now if one considers a float number $0 \leq s < r$ with $s = s_1.s_2 \cdots s_M B^{\lambda_s}$, from the definition of $r$ one can deduce that $\lambda_s < \lambda_a - M + 1$ and Therefore $\lambda_s \leq \lambda_a - M$. Hence $a + s$ will be rounded to $a$. ∎

**Lemma 31 (Underflow boundary)** *Let $(B, M, X)$ the finite precision configuration. We define $u := \lfloor \frac{B^X}{2} \rfloor$ and $l := B^X - u$. If $M < \min\{u, l\}$ then there exists $J$ such that $\sigma(J) = 1$ and $\sigma(-J) = 0$ and for all positive float numbers $s$ such that $J - s < J$ we have $\sigma(-(J-s)) \neq 0$*

**Proof** The total number of exponents obtainable in this configuration is: The configuration $(B, M, X)$ allows to have $u := \lfloor \frac{B^X}{2} \rfloor$ positive exponents and $l := B^X - u$ negative exponents based on Definition 1. Let $K := \ln(B)l$ we have on one side: $\sigma(K) = \frac{1}{1 + e^{-\ln(B)l}} \leq \frac{1}{1 + B^{-l}} = 1$ because $1 + B^{-l}$ will be rounded to 1 in virtue of the Hypothesis 31 we know that $M < min\{u, l\}$. On the other side we get $\sigma(-K) = \frac{1}{1 + e^{\ln(B)l}} = \frac{1}{1 + B^l} = \frac{1}{B^l} = B^{-l}$ the smallest number we can have in the finite precision setup $(B, M, X)$. Consequently if we set $J := K + r$ we get:

$$e^{-K-r} = B^{-l} e^{-r} < B^{-l}$$

Hence we will have $\sigma(J) = 1$ and $\sigma(-J) < B^{-l}$ Therefore $\sigma(-J)$ will be inevitably rounded to zero.

Note that in the EEEI 754 norm the exponent of the 32 bit float numbers we have $l = 128$, $u = 127$, and the assumption $M < min\{u, l\}$ is satisfied. ∎

# E    Numerical experiments

This appendix is dedicated to more in-depth explanations of the numerical experiments described in Section 6.

For our vanishing gradient experiment we randomly generated a labeled training set of 458 words on the alphabet $\{a, b\}$. Words with label 1 are those containing an odd number of $a$'s, and the other words are labeled by 0. We defined a set of goal parameters $\theta_{Target}$ corresponding to the parameters of the example of Section 4.3, and we randomly defined initial parameters $\theta_{Initial}$. In order to simulate the best case scenario where the training of the FP-SRN would lead from $\theta_{Initial}$ to $\theta_{Target}$ in a straight line, we defined a sequence of parameter vectors $\{\theta_k\}_{k \geq 0}$ with

$$\theta_0 := \theta_{Initial}$$
$$\theta_{k+1} := \theta_k + \alpha \cdot (\theta_{Target} - \theta_{Initial})$$

where $0 < \alpha$ is a predefined learning rate. The goal of this setup is to calculate $\nabla \mathcal{L} \left( \mathcal{R}_{\theta_k}(\omega), y \right)$ over the whole training dataset, for $k = 0, 1, \dots$ until $\| \nabla \mathcal{L} \left( \mathcal{R}_{\theta_k}(\omega), y \right) \|_\infty < 10^{-14}$ for all $(\omega, y) \in S$. Once the stopping criterion was reached at $\theta_s$, we reproduce the same procedure with a more refined sequence of vector $\{\theta'_{k'}\}_{k' \geq 0}$ defined by:

$$\theta'_0 := \theta_{s-1}$$
$$\theta'_{k'+1} := \theta'_{k'} + \alpha' \cdot (\theta_{Target} - \theta_{Initial})$$

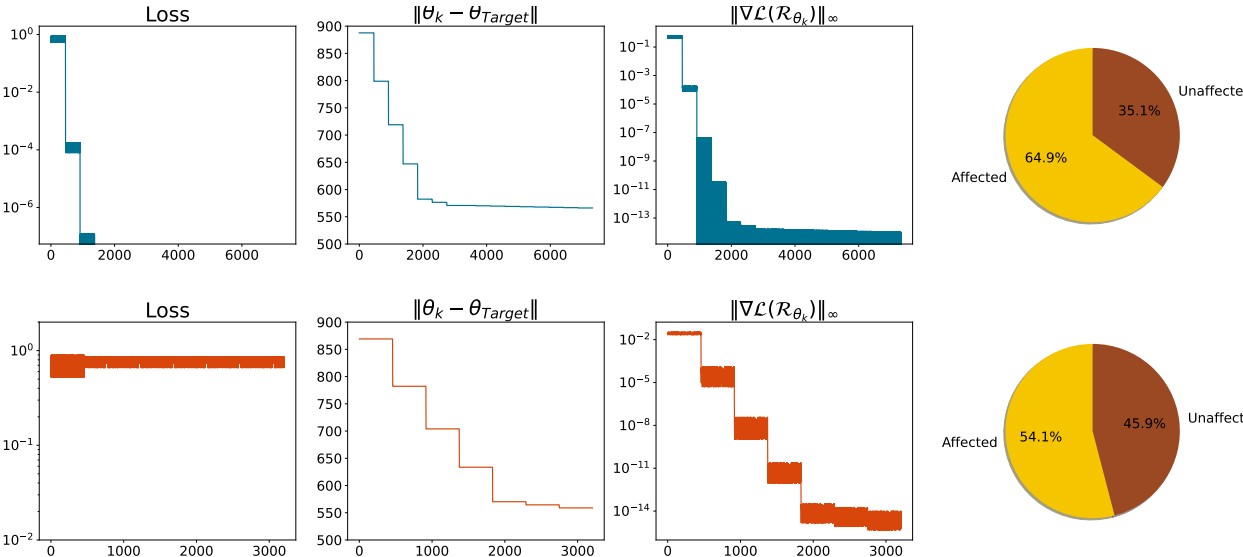

Figure 9: Two variants of the synthetic experiment, the first row represent the experiment where all the parameters are updated. The second row is the variant where the decoders parameters are frozen after one update.

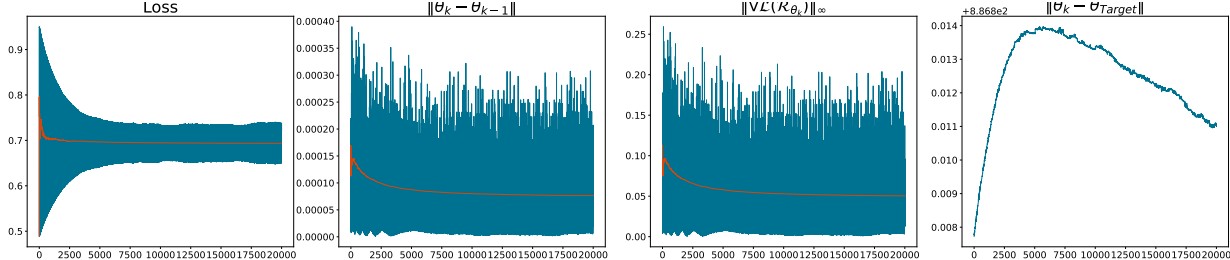

Figure 10: Regular SGD training where the four statistics of the learning experience of the odd $a$'s language. The curves in blue correspond to the statistics recorded during training, and the curves in orange are the rolling averages.

where $0 < \alpha' < \alpha$ is a strictly smaller learning rate. The refining, as we repeat it, allows us to find a better approximation of the point where the gradient becomes stationary. In our experiments we applied the straight line gradient evaluation with learning rates $\alpha \in \{0.1, 0.01, 0.001\}$.

Two version of the strait line gradient evaluation are displayed in Figure 9, one where all the parameters are updated represented by the blue graphs, and the second one where only the Encoders parameters are updated and the Decoders parameters are frozen after one update. Figure 9 displays, from left to right, the graphs of: the Loss, the distance to the target, the $\|Gradient\|_\infty$ infinite norm of the gradient, and the proportion of the parameters affected by the stationary gradient. We recall that a parameter $m_{i,j}$ experiences a stationary gradient if in finite precision

$$\forall \, (\omega, y) \in S \quad m_{i,j} - lr \cdot \frac{\partial \mathcal{L}}{\partial m_{i,j}}(\mathcal{R}(\omega), y) = m_{i,j}.$$

In other words, for the setup where all the parameters are updated, 64,9% of the parameters are experiencing the stationary gradient once the experiment is halted.

Figure 10 represents our attempt to learn a model with the same data by applying regular SGD. We used the same training data set as for the previous experiment, and we trained a SRN on 20,000 batches of size 32

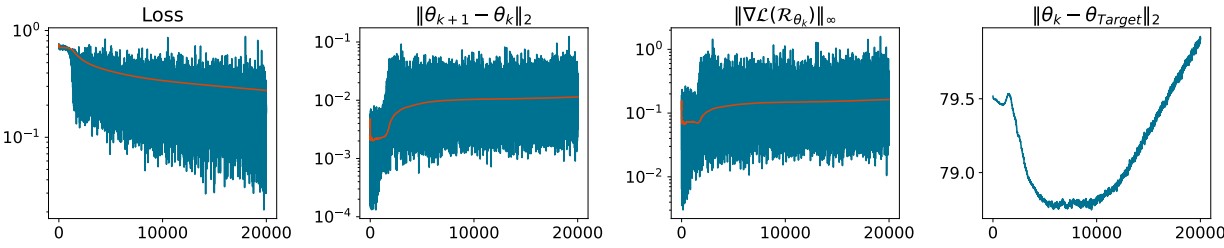

Figure 11: Training with SGD on the language **04.02.TLTT.2.3.0**

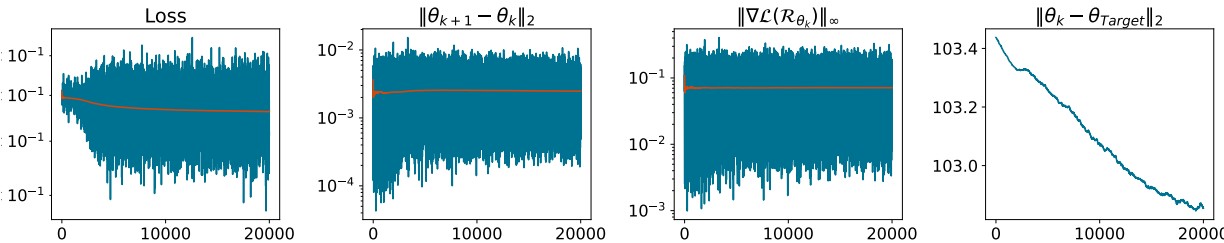

Figure 12: Training with SGD on the language **04.02.TLTT.4.2.3**

words. We adopted a slightly different method from that is usually used with pytorch or keras frameworks, since for each iteration between 1 and 20,000 we randomly selected 32 words from the 458-word training dataset. Usually the dataset is shuffled like a deck of cards, after which the batches are trained and fixed for the entire training run. The statistics we observed in this experiment are: the Loss, the Euclidean distance between two consecutive parameter vectors $\|\theta_t - \theta_{t-1}\|_2$, $\|\nabla\mathcal{L}(\mathcal{R}(\omega), y)\|_\infty$, and the euclidean distance to the target, the target being the same set of parameters as in the experiment described just above. In a gradient descent learning scenario, the nature of the data is not known a priori, so the distance to the target makes little sense. For this reason, we use the Euclidean distance between two consecutive parameter vectors, in order to track down some kind of convergence through the Cauchy criterion. Of course, in our experimental context, we know the data precisely, and thus the associated DFA. We therefore also displayed the distance to the goal parameters. With these experiences, we can conclude that SGD cannot even push FP-SRN parameters into stationary gradient regions.

As argued in Section 6, the language on the alphabet $\{a, b\}$ formed of all words containing an odd number of $a$'s falls into the highest complexity subclass of regular languages. To avoid any kind of bias linked to the complexity of the formal language, we have selected a set of 9 languages in the van der Poel et al. (2024) benchmark. The 9 selected languages cover the entire complexity spectrum of the benchmark. We reproduced the same learning framework, with learning samples for each 1000-word language and the same learning algorithm as for the experiment illustrated in Figure 10. The 9 figures 11,12,13,14,15, 16,17,18 and 19 are the reproduction of the experiment 6 on 9 MLRegTest dataset languages van der Poel et al. (2024). We have reproduced exactly the same experiment setup and collected the same statistics as in the 6 experiment. For these 9 experiments, we can draw the same conclusions as for the 6 experiment. We do not observe any convergence within the different statistics, after 20,000 mini batches. We conclude that the complexity of the regular language is not impact-full in this experimental paradigm. We recall that we randomly drew each mini-batch of the 20,000, unlike the usual setup where the training set is divided into mini-batches before training which will not change.

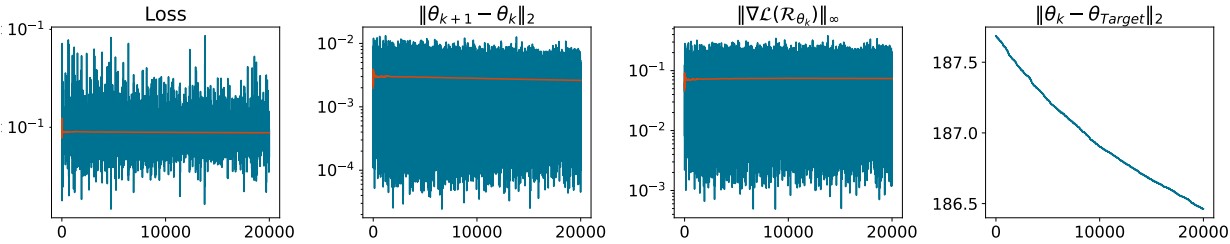

Figure 13: Training with SGD on the language **04.04.Reg.0.0.9**

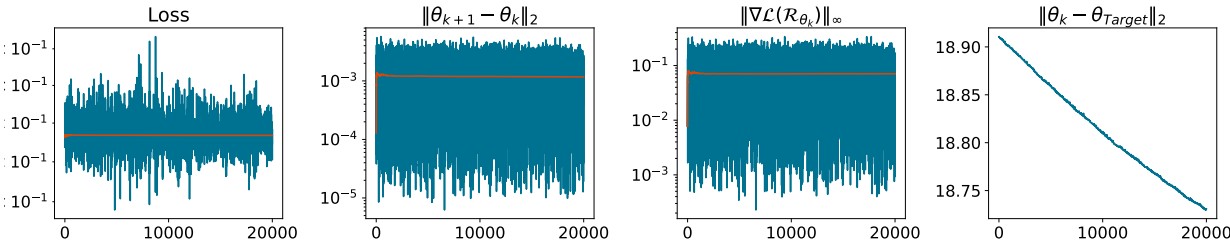

Figure 14: Training with SGD on the language **04.04.Zp.2.1.0**

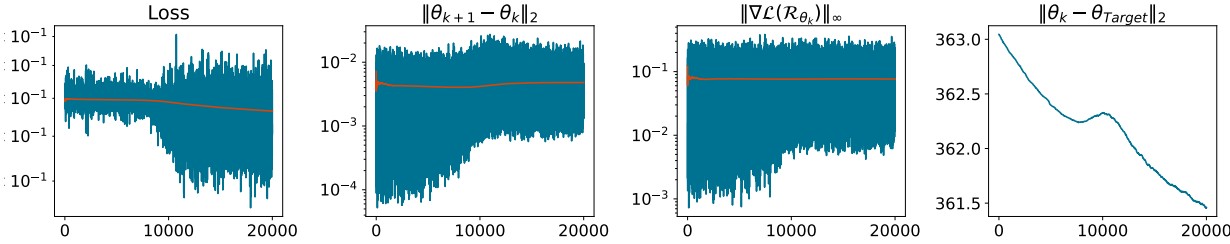

Figure 15: Training with SGD on the language **16.04.TLT.2.1.4**

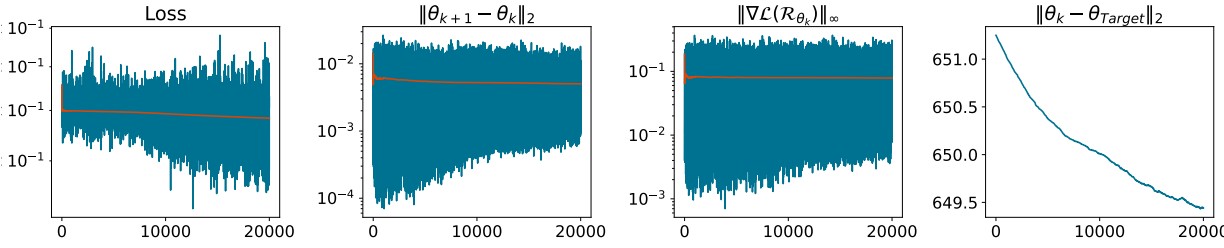

Figure 16: Training with SGD on the language **16.16.LT.4.1.5**

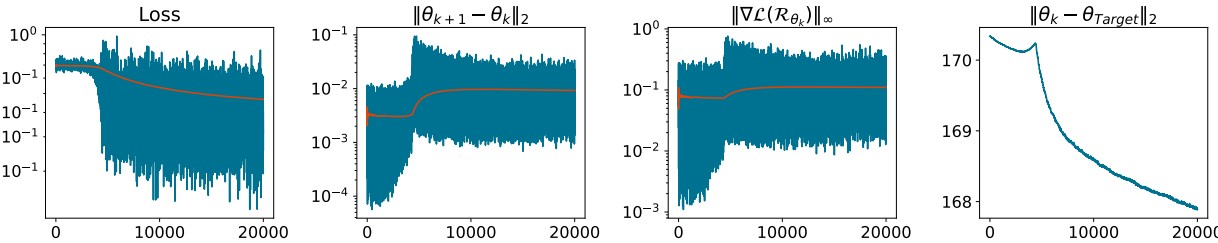

Figure 17: Training with SGD on the language **16.16.SP.2.1.0**

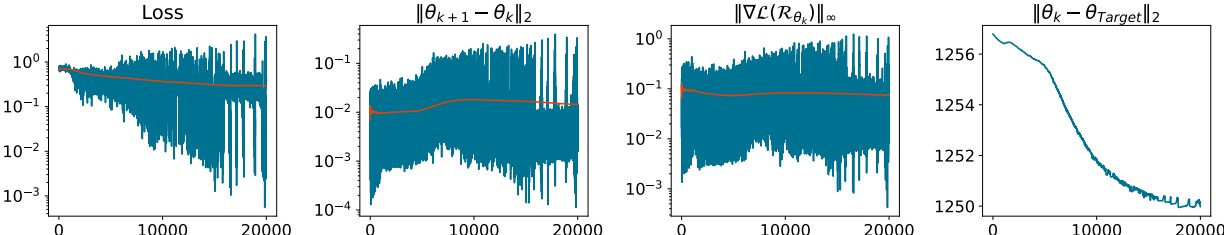

Figure 18: Training with SGD on the language **64.64.SF.0.0.0**

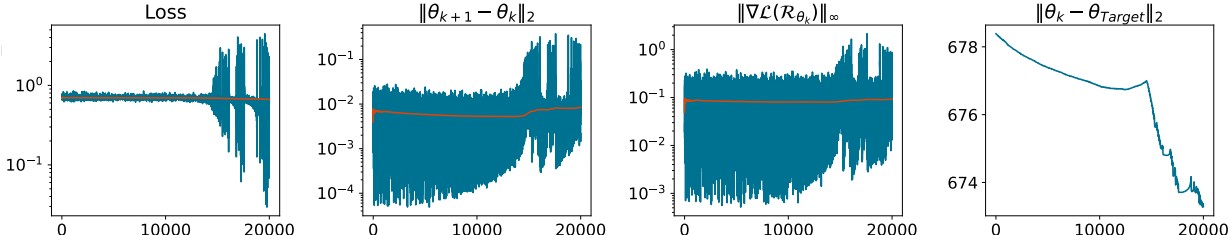

Figure 19: Training with SGD on the language **64.64.SL.4.1.0**

