# OpenReview forum: "On the theoretical limit of gradient descent for Simple Recurrent Neural Networks with finite precision"
_TMLR — Accepted by TMLR_

### Review · Reviewer_SuQh · 2024-09-05

**Summary Of Contributions:**

The authors make two significant contributions in this paper:

1. They demonstrate that there are regions in the parameter space of finite-precision simple recurrent neural networks that are (1) robust to noise and (2) exhibit a stationary gradient.
2. They devise an algorithm to convert any Deterministic Finite State Automaton into a finite-precision simple recurrent network.

**Audience:**

Yes

**Broader Impact Concerns:**

I do not think there are any broader impact concerns.

**Claims And Evidence:**

No

**Requested Changes:**

As it stands, I believe some audiences may find this paper interesting, but it does not yet meet the *claims and evidence* criterion. Specifically, I don't think Section 6 provides clear evidence to support the theory proposed by the authors. However, I am open to changing my perspective if the authors make some revisions.

Proposed Changes:
1. **Improvement in presentation**: The overall presentation of the paper needs enhancement. There are clear formatting issues throughout, figure descriptions lack sufficient detail, and there are several typographical errors. In my view, this detracts from the "clarity of evidence" component, which is part of the selection criteria for TMLR. While this is not an especially precise critique, I feel that the paper doesn't currently have a "professional" appearance. I would like to see it reworked to present a more polished and professional tone.

2. **Significant improvements to Section 6**: Currently, none of the results in Section 6 appear to support the theorem presented by the authors. The predicted stagnation distance should be 6.6, but the actual distance is 500. Moreover, the SGD experiments suggest that traditional learning fails which is interesting but seemingly orthogonal to the paper's main message. This section needs to better align with the theoretical claims or include significant discussion to address the discrepancy between the theory and the experiments. I would also be open to the authors removing it entirely and simply making the theoretical claim in theorem 16 although they would need to justify in rebuttal why we should ignore the discrepancy presented in section 6.

3. **Clearer contributions section or conclusion**: I would like to see a more structured and clear contributions section or conclusion to emphasise the key takeaways from the paper.

Additional Suggestions:
1. The use of $Ex$ for exponent seems sub-optimal. When I read this, I initially think you are referring to a matrix-vector product between $E$ and $x$. Perhaps you could use $q$ or some other single letter variable instead.

2. Could the authors adopt the more traditional $\theta$ for parameters, instead of $p$? This would align better with standard notation in the field.

3. The notation $p[Encoder]$ for the encoder's parameters is confusing. I think the notation would be clearer if it followed the typical convention of using $\theta_{\text{encoder}}$.

**Strengths And Weaknesses:**

This paper seems to be a decent theoretical contribution to the literature. Although I haven't checked the full proofs in the appendix, the sketch for Theorem 16 appears to make sense.

Strengths:
1. I like the use of diagrams in the text, though they require more detailed descriptions.
2. I appreciate that a concrete example is provided to illustrate how Theorem 16 can be applied.

Weaknesses:
1. The paper, overall, could benefit from greater clarity. For example, I am quite confused about what is being tested in the experimental section. What are the dataset, model, learning rate, etc.?

2. Most of the figures would benefit from more detailed descriptions. A single short sentence is not sufficient, in my opinion.

3. Do the authors have a solid explanation for why the theory and practice diverge in Section 6? There appears to be a significant discrepancy that isn't adequately discussed. Additionally, the SGD results seem to be entirely orthogonal to the main argument of the paper.

4. There should be a figure label for the image under Figure 3.

5. The take-home messages need to be clearer. Outside of Theorem 16, I'm unsure what other key points I'm meant to take away from the paper.

Specific weaknesses:
1. In Section 4.6, it is noted: "SRNs with more than $10^3$ neurons are rarely observed, so this last constraint on the number of neurons is realistic." Can the authors provide more justification for this? It seems to me that 1000 neurons might be quite small, though I'm not deeply familiar with this subfield.

2. The formatting on page 13 seems off.

3. The use of what appear to be Desmos figures makes the paper seem a bit unprofessional. This isn't a major reason for acceptance or rejection, but it's worth noting.

---

> ### Author Response · Authors · 2024-10-15
> **Our remarks and a discussion regarding the remarks.**
>
> We want first to thanks the Reviewer for noticing the potential impact of our work, our effort to illustrate our points with diagrams, as well as the interest of a broad study of an example to show how the constants of the theorem interact. We apologize for the lack of clarity in the experimental section. We have focused on the example and the interactions of the various constants in Theorem 16,
> and have not devoted enough attention to the experimental section. This error is of course corrected in the new version of this work. In particular, we precisely describe the data and the experimental setup used (the code will be made available once the anonymity constraint will be over).
>
> An important comment about experiments, that is linked to the take-home message, is that it does not diverge from the theory: on the contrary, it corresponds to an empirical validation of our theoretical result. Indeed, the main claim of our work is that it is impossible to learn saturated SRNs in finite precision with usual gradient descent algorithms. Given a vector of parameters θ that ensures the saturation of a SRN, Theorem 16 can be seen as a proof that there exists a buffer zone around θ that ensures that the gradient will be stationary in this zone. In the example we give, the theorem asserts that the buffer zone extends at least to distance 6.16 from the target parameters. What the experiments exhibit is that the task of learning saturated SRNs is even more difficult in practice than in theory by showing that, even in a best case scenario, the gradient will be stationary at distances greater than 500 to the target
> parameters, thus extending the buffer zone boundary from 6.16 up to over 500.
>
> This difference can be explained by the use of loose upper bounds in the proofs of Theorems 14 and 16. Just to put it in context, our approach is to exhibit a set of SRN parameters θ that we are sure have a stationary gradient (Def. 15) and such that Rθ is β-saturated. Then we show that it is possible to construct an open ball centered at θ that contains parameters ˚θ that will have similar characteristics (stationary gradient and β′-saturated with β′ ≈ β). The way we prove that neighboring parameters will generate β′- saturated SRNs is by showing that for a given sequence, the execution of the neighbors R˚θ will not deviate from the execution of Rθ (i.e. their respective hidden vectors $h_k$ and ˚$h_k$ will be close to one another at each iteration). The first loose upper bounding is in the estimation of these deviations where a triangle inequality is applied. A second non-optimal upper bounding is applied when considering the Lipschitz constant of the Loss function. This constant, denoted $\rho_L$ in the article, plays an important role in Theorem 16. It represents the worst possible case in the deviations between Rθ and its neighbors R˚θ. Note that the β-saturation can provide us with guarantees on the deviation: it is conceivable to remove $\rho_L$ and to replace it with a constant that will provide a much more realistic result. But we chose to go with the Lipschitz constant because it simplified the result while validating the message.
>
> The experimental section is roughly divided into two parts. In one of them – unfortunately presented in second in the submitted version – we try to learn SRNs in the classical way from a dataset by applying the SGD algorithm. The learning task is to learn an automaton from labelled words: we observe that it does not converge. However, as pointed out by the Reviewer, it does not suffice to validate experimentally the theorem since the reason of failure might be different than gradient vanishing. However, we decided to keep this part since we think it’s of interest to observe GD failure on this dataset – though we moved it at the beginning of the section because of its relative interest toward the goal the paper. The other part of the experiments is a synthetic experiment. What we are interested in is swallowing the gradient along the straight line between the initialization parameters and the parameters of the target FP-SRN provided by our algorithm. Doing so, we force the best case scenario where the gradient is constantly pointing in the best direction, towards the ideal parameters. We monitor the gradient norm as it is the easiest way to link the practical learning with Theorem 16. The reported results validate the theorem, showing that, even in this best case scenario, stationary gradient happens, farther from the target in practice than what the theorem predicts (see discussion above on that point). We agree with the Reviewer that the order in which the two sets of experiments were presented in the submitted version, and the way they were described and
> discusses was not clear, which diminished the interest of this section.

---

> > ### Author Response · Authors · 2024-10-15
> >
> > As a consequence, we completely rewrite the experimental section. It now discusses all these elements, hoping it is enforcing its interest and its clarity - and thus its impact.
> >
> > We would also like to address our claim that SRNs with more than $10^3$ neurons are rarely observed. We indeed have stated this fact without justifying it. Based on the survey of Lara-Benitez et al. [3] published in 2021, we found 3 works published between 2012 and 2018 that employ SRNs ([1], [6] and [5]). The information about the number of neurons is missing in [5], while the number of hidden neurons is less than 103 in the two other papers. The most recent use of SRNs, dating from 2023, is in the TAYSIR competition [2], where a benchmark of sequential models was presented to competitors. The benchmark is composed
> > of different RNN architectures and in particular, SRNs were proposed whose number of hidden neurons did not exceed 512. We added a discussion on that matter in the current version.
> >
> > To finish this already long answer, we want to add that we carefully follow all remarks on form. This includes longer captions for figures, the weird formatting of page 13, better drawn figures, a thorough typo search, a change of notations to follow more closely the usual ones in the field. We are grateful for the corresponding remarks since we are confident that these changes enforce
> > the clarity of the article and allow an easier understanding of the take-home messages.
> >
> > [1] Rohitash Chandra and Mengjie Zhang. Cooperative coevolution of elman recurrent neural networks for chaotic time series prediction. Neurocomputing, 86:116–123, 2012.
> >
> > [2] Rémi Eyraud, Dakotah Lambert, Badr Tahri Joutei, Aidar Gaffarov, Mathias Cabanne, Jeffrey Heinz, and Chihiro Shibata. Taysir competition: Transformer+\textscrnn: Algorithms to yield simple and interpretable representations. In International Conference on Grammatical Inference, pages 275–290. PMLR, 2023.
> >
> > [3] Pedro Lara-Benitez, Manuel Carranza-Garcia, and Jose C Riquelme. An experimental review on deep learning architectures for time series forecasting. International journal of neural systems, 31(03):2130001, 2021.
> >
> > [4] Volodimir Mitarchuk, Clara Lacroce, Rémi Eyraud, Rémi Emonet, Amaury Habrard, and Guillaume Rabusseau. Length independent pac-bayes bounds for simple rnns. In International Conference on Artificial Intelligence and
> > Statistics, pages 3547–3555. PMLR, 2024.
> >
> > [5] Mohsen Mohammadi, Faraz Talebpour, Esmaeil Safaee, Noradin Ghadimi, and Oveis Abedinia. Small-scale building load forecast based on hybrid forecast engine. Neural Processing Letters, 48:329–351, 2018.
> >
> > [6] R Rueda and MC Pegalajar. Energy consumption forecasting based on elman neural networks with evolutive optimization. Expert Systems with Applications, 92:380–389, 2018.
> >
> > [7] Chihiro Shibata, Kei Uchiumi, and Daichi Mochihashi. How lstm encodes syntax: Exploring context vectors and semi-quantization on natural text. arXiv preprint arXiv:2010.00363, 2020.

---

### Review · Reviewer_bmSq · 2024-09-17

**Summary Of Contributions:**

The paper focuses on understanding neural network behavior regarding the functions learnable in the context of a finite precision configuration. It investigates the limitations of gradient descent in Simple Recurrent Networks (SRN), identifying specific conditions where gradient descent encounters failures. Additionally, the authors introduce a class of SRNs based on Deterministic Finite State Automata (DFA) that illustrate these failure scenarios. They also present a constructive algorithm that converts any DFA into an SRN that computes exactly the same function, contributing valuable insights to the field.

**Audience:**

Yes

**Claims And Evidence:**

Yes

**Requested Changes:**

The results seem to solid and novel. My main concern revolves around the practical implications of the findings and how practitioners can leverage these theoretical insights to enhance their work. Could you please provide more detailed explanations?

**Strengths And Weaknesses:**

Strengths:

1. The work addresses the theoretical gap by exploring the limitations of training RNNs via Gradient Descent (GD) in finite precision.

2. The study explicitly identifies regions of the parameter space that are unreachable by classical GD with a bounded learning rate, providing a clearer understanding of the limitations faced in training saturated SRNs.

3. The authors prove a theorem that establishes the impossibility of learning saturated SRNs through GD.

4. The paper introduces an algorithm that constructs a saturated SRN from a Deterministic Finite State Automaton (DFA), extending Minsky's earlier work and offering practical implications for simulating finite state machines.


Weaknesses:

1. The study's focus on SRN may limit the generalizability of the findings to other types of neural networks.

2. The proposed algorithm may be complex to implement in practice.

3. The reliance on the saturation assumption may not hold in real-world scenarios, which could affect the applicability of the conclusions drawn.

---

> ### Author Response · Authors · 2024-10-15
> **Our thanks, and responses to comments on the SRN study and the saturation hypothesis.**
>
> First of all, we would like to thank the reviewer for noticing our contributions to the establishment of a learning impossibility theorem by providing explicit regions that cannot be reached by gradient descent, our theoretical contribution to the learning theory by incorporating finite precision in the analysis, and the value of the extension of M. Minsky's algorithm to SRN.
>
> To answer the main concern, a first answer is the state of the deep learning scientific field: the impressive practical results are way ahead of our theoretical understanding. In this context, theoretical works often seem disconnected to practical advancements, and this one is not an exception.
>
> In our work, we focus on SRNs because even if these models appear simple, they already present some challenges to address. Their specific properties allowed us to derive proofs and provide interesting insights about their learning capabilities. Nonetheless, our work paves the way to theoretical continuations, in particular by adapting the proofs to more complex - and thus more related to practical RNNs - architectures.
>
> Concerning the saturation assumption, it is indeed a strong constraint empirically.
> Nevertheless, it is an element that could be of great use for practitioners. Indeed, behaviors closely related to saturation have been observed in practice (see for instance [4,7]).
> Moreover, saturation is interesting given the stability that it offers and the long term dependencies it allows (as in DFAs). Since our work shows that gradient descent is not well-suited to achieve saturation, an advice that practitioners who are interested in learning long term dependencies may get from our work is to look either for new regularization term forcing saturation in the loss or even for other learning algorithm than GD. We added a remark describing these elements at the end of our conclusion.
>
> Finally, concerning the remark about the complexity to implement the algorithm in practice, we are not sure what this remark is targeting. To show that it can easily be coded, we will provide an implementation of the algorithm and we added its pseudo-code in the article.
>
> [4] Volodimir Mitarchuk, Clara Lacroce, Rémi Eyraud, Rémi Emonet, Amaury
> Habrard, and Guillaume Rabusseau. Length independent pac-bayes bounds
> for simple rnns. In International Conference on Artificial Intelligence and
> Statistics, pages 3547–3555. PMLR, 2024
>
> [7] Chihiro Shibata, Kei Uchiumi, and Daichi Mochihashi. How lstm encodes
> syntax: Exploring context vectors and semi-quantization on natural text.
> arXiv preprint arXiv:2010.00363, 2020

---

### Review · Reviewer_TtPt · 2024-10-04

**Summary Of Contributions:**

The paper provides a class of recurrent neural networks that cannot be learned. More precisely, these networks are assumed to have finite precision, and it's assumed that they are saturated. The paper shows that then there is a space of parameters on which the gradient is "stationary," meaning that applying a step of gradient descent doesn't update the network's parameters.

**Audience:**

Yes

**Broader Impact Concerns:**

Not applicable.

**Claims And Evidence:**

Yes

**Requested Changes:**

I understand that Section 2 contains the necessary definitions for the paper, but it's quite a dry, long section. Perhaps some of the less crucial definitions can be moved to the appendix?

There are a few spelling and grammatical issues that need to be fixed. I found these ones, but a careful proofreading is appreciated:
1. Page 1, Line -2: "decent" -> "descent"
2. Page 4, Line 6: "lost" -> "loss"
3. Page 4, Line before Section 2.3: Period is missing. Also, "Appendix 31" -> "Appendix D"
4. Definition 8, did you mean 0 <= beta <= 1?

**Strengths And Weaknesses:**

Strength:
This is a solid theoretical contribution, proving that the celebrated gradient descent algorithm fails in some simple cases. It is important for the deep learning community to know about the failure cases of gradient descent so as to avoid them in practice.

---

> ### Author Response · Authors · 2024-10-15
> **Our thanks, and our response to the comment on the number of definitions in Section 2.**
>
> We thanks the Reviewer for noticing the solidity of our theoretical work.
>
> We agree that Section 2 contains a lot of definitions. However, since our work stands between three different fields -- namely language theory, the computation theory, and the deep learning -- we think that researchers in one of these fields might find it useful to introduce concepts from the other fields. For this reason, we think that moving some definitions into the appendix might degrade the reading experience.
>
> Finally, we carefully went through the paper searching for typos, spelling and grammatical issues, fixing everything we found.

---

### Author Response · Authors · 2024-10-15
**General response**

We wish to thank all reviewers for their valuable comments. We believe your feedback allowed us to elevate the quality of this work. We responded individually to each review, taking care to address every remark/question that has been raised. A modified version the article is uploaded which is in line with the comments on; the typos, the quality of the figures and their captions, and the structure of the entire experimental section. In order to be easy to locate, the text modifications are in blue.

---

### Decision · Action_Editor_z7WW · 2024-11-18

**Recommendation:** Accept as is

**Comment:**

This paper tackles the highly practical issue of gradient saturation in neural networks with a rigorous theoretical approach. However, its current presentation may be daunting for practitioners. The main result, which highlights a critical failure mode of gradient descent in finite-precision training, is not presented until page 10 after extensive technical groundwork.  To enhance accessibility, I recommend providing a simplified explanation of the core findings and their implications earlier in the paper. This would allow practitioners to quickly grasp the key insights and motivate them to delve deeper into the technical details.


I also recommend the authors explore the growing body of literature on low-precision training of language models (see e.g., https://arxiv.org/abs/2407.17465. ) This could uncover interesting theoretical questions and reveal potential issues with low-precision training that practitioners may not be aware of.




Typo:  4.1 Back Propagation `Trough` Time

**Audience:**

Yes. The paper studies learnability and limitation of gradient descent (on simple RNN) which is an important topic in machine learning.

**Claims And Evidence:**

This paper presents interesting theoretical work on the limitations of gradient descent in finite-precision recurrent neural networks. It demonstrates that when these networks are `saturated,` meaning their parameters cannot be updated by gradient descent, there exists a class of functions they cannot learn. This highlights a potential failure mode of gradient descent (for simple RNN), especially relevant as low-precision training becomes increasingly popular and relevant for training large language models.

All reviewers agree this is a solid/decent theoretical contribution, with two recommending acceptance and one leaning towards acceptance. I concur with their assessment and recommend accepting this paper.